# Murine models of *IDH*-wild-type glioblastoma exhibit spatial segregation of tumor initiation and manifestation during evolution

Yinghua Li [1,2,3,13], Bo Li[4,13], Wei Li[5,13], Yuan Wang [1,2,3], Seçkin Akgül [1,2,3,6,12], Daniel M. Treisman [1,2,3,6], Kevin A. Heist[7], Brianna R. Pierce[1,2,3], Benjamin Hoff[7], Cheng-Ying Ho[5], David O. Ferguson[8], Alnawaz Rehemtulla[9], Siyuan Zheng [10], Brian D. Ross[7], Jun Z. Li [4] & Yuan Zhu [1,2,3,6,11 ✉]

Recent characterization of spatiotemporal genomic architecture of *IDH*-wild-type multifocal glioblastomas (M-GBMs) suggests a clinically unobserved common-ancestor (CA) with a less aggressive phenotype, generating highly genetically divergent malignant gliomas/GBMs in distant brain regions. Using serial MRI/3D-reconstruction, whole-genome sequencing and spectral karyotyping-based single-cell phylogenetic tree building, we show two distinct types of tumor evolution in p53-mutant driven mouse models. Malignant gliomas/GBMs grow as a single mass (Type 1) and multifocal masses (Type 2), respectively, despite both exhibiting loss of *Pten*/chromosome 19 (chr19) and PI3K/Akt activation with sub-tetraploid/4N genomes. Analysis of early biopsied and multi-segment tumor tissues reveals no evidence of less proliferative diploid/2N lesions in Type 1 tumors. Strikingly, CA-derived relatively quiescent tumor precursors with ancestral diploid/2N genomes and normal *Pten*/chr19 are observed in the subventricular zone (SVZ), but are distantly segregated from multi focal Type 2 tumors. Importantly, PI3K/Akt inhibition by *Rictor*/mTORC2 deletion blocks distant dispersal, restricting glioma growth in the SVZ.

[1] Gilbert Family Neurofibromatosis Institute, Children's National Hospital, Washington, DC 20010, USA. [2] Center for Cancer and Immunology Research, Children's National Hospital, Washington, DC 20010, USA. [3] Center for Neuroscience Research, Children's National Hospital, Washington, DC 20010, USA. [4] Department of Human Genetics, University of Michigan Medical School, Ann Arbor, MI 48109, USA. [5] Center for Genetic Medicine Research, Children's National Hospital, Washington, DC 20010, USA. [6] Cellular and Molecular Biology Graduate Program, University of Michigan Medical School, Ann Arbor, MI 48109, USA. [7] Department of Radiology, University of Michigan Medical School, Ann Arbor, MI 48109, USA. [8] Department of Pathology, University of Michigan Medical School, Ann Arbor, MI 48109, USA. [9] Department of Radiation Oncology, University of Michigan Medical School, Ann Arbor, MI 48109, USA. [10] Greehey Children's Cancer Research Institute, The University of Texas Health Science Center at San Antonio, San Antonio, TX 78229, USA. [11] GW Cancer Center, The George Washington University, Washington, DC 20052, USA. [12] Present address: Sid Faithfull Brain Cancer Laboratory, Department of Cell and Molecular Biology, QIMR Berghofer Medical Research Institute, Brisbane, QLD 4006, Australia. [13] These authors contributed equally: Yinghua Li, Bo Li, Wei Li. ✉email: yzhu@childrensnational.org

Glioblastoma (GBM or Grade IV malignant glioma) is the most common and lethal primary brain tumor in adults with a median survival of 15 months, despite recent advance in conventional and molecularly targeted therapies[1–3]. The vast majority of GBMs (~90%) in adults are primary (de novo) GBMs that arise rapidly without evidence of preexisting lower-grade gliomas (Grade II or Grade III)[2]. Primary GBMs are typically wild type (WT) for isocitrate dehydrogenase 1/2 (*IDH*), with no *IDH*-mutant associated hypermethylated phenotype (termed glioma CpG island methylated phenotype (G-CIMP)), while secondary (progressive) GBMs often arise from *IDH*-mutant lower-grade gliomas[1,3–7]. It remains speculative whether rapidly progressing *IDH*-WT primary GBMs evolve from an early lesion with a less aggressive and proliferative phenotype. More importantly, it remains unknown whether a therapeutic strategy can be developed to treat primary GBMs at early stages.

Recent characterization of the spatiotemporal architecture of GBM genomic profiles, based on matched diagnostic and relapsed GBMs as well as GBMs from different regions of the same brains, provided important insights into early stages of primary GBM evolution[8–15]. Recurrent GBMs at sites distant from their initial tumors tend to be highly genetically divergent from their therapy-naive initial tumors, and in some cases, nearly 90% of genetic events are not shared[8–10]. This observation supports a model wherein the dominant clone(s) at diagnostic and relapse stages diverged from a "common ancestor" (CA) with fewer driver mutations and a less aggressive phenotype[13]. Similarly, multi-focal/multicentric GBMs (M-GBMs) from the same brain at diagnosis often share very few mutational events[10]. Importantly, recurrent M-GBMs frequently arise at sites distant from the diagnostic lesion(s)[9,16–18]. These observations suggest the existence of a unique evolutionary pattern, particularly for pre-therapy M-GBMs and post-therapy distally recurrent GBMs. This mode of GBM evolution suggests spatial segregation between CA-derived early precursor lesions at the tumor initiation site and disease-causing GBMs in distant regions. A critical question is whether a specific brain region(s) is particularly more susceptible to generating a CA cell and undergoing local expansion, while remaining clinically silent during GBM evolution. A recent study demonstrated that the subventricular zone (SVZ)—the largest source of neural stem cells in the adult brain—contained cells with low-level driver mutations that were shared with GBMs clinically manifested in distant brain regions[14,19]. This phenomenon was observed in over half of individuals with *IDH*-WT GBMs, but not in other types of brain tumors[14]. This study further showed that mouse SVZ cells simultaneously transformed by three driver mutations (*p53*, *Pten*, and *Egfr*) migrated out into distant brain regions and formed malignant gliomas and GBMs[14]. Critically, histopathologically normal cells with the targeted mutations were maintained in the SVZ[14]. However, it is not clear that the SVZ cells and distally located GBMs, despite being targeted by the same three cancer drivers, are clonally related[14].

Here we present a series of genetically engineered mouse (GEM) models of *IDH*-WT GBMs that arise from a susceptible cell population(s) targeted only by mutation(s) in *p53* tumor suppressor. Only after accumulating additional cancer driver alterations, a single *p53*-mutant cell is selected to become a CA cell that evolves into malignant gliomas and GBMs. Importantly, these single-cell-derived models recapitulate spatial segregation of tumor initiation in the SVZ and distant manifestation sites observed in human *IDH*-WT GBMs.

## Results

### GEM models for human primary *TP53*-mutant *IDH*-WT GBM.
To develop GBM models with an evolutionary pattern similar to

human cancer, we constructed a series of *p53* conditional knockout (CKO) models for GBM driven by a neural-specific Cre driver under control of the human GFAP promoter (hGFAP-cre) (Fig. 1a and Supplementary Fig. 1a). The *p53*CKO model ($p53^{R172H/\Delta E2-10}$CKO or $p53^{R172H}$CKO) produced targeted cells with *p53* compound heterozygous mutations harboring one $p53^{\Delta E2-10}$ null mutation (deleting the entire *p53* coding region) and one hotspot missense point mutation $p53^{R172H}$ (homologous to $TP53^{R175H}$ in human cancers) (Fig. 1a and Supplementary Fig. 1b)[1]. Given that a recent study showed no evidence of gain-of-functions of mutant *p53* alleles in human myeloid cancers[20], we generated a second model ($p53^{\Delta E2-10/\Delta E2-10}$CKO or $p53^{null}$CKO) carrying homozygous *p53*-null mutations to investigate additional oncogenic activity(s) of mutant *p53* alleles in $p53^{R172H}$CKO littermates (Fig. 1a and Supplementary Fig. 1b). In comparison, we included a previously published *p53*CKO GBM model ($p53^{\Delta E5-6}$CKO) in which the hGFAP-cre driver induced an in-frame *p53* deletion mutant lacking exons 5 and 6[21]. No significant difference was observed among the three *p53*CKO models, including survival, tumor latency, histopathology, and tumor penetrance for gliomas (Grade III/IV, 75–80% of *p53*CKO mice) and GBMs (Grade IV, 50%) (Fig. 1b, c and Supplementary Fig. 1c, d). We validated that malignant gliomas and GBMs in the two new *p53*CKO models arose from $p53^{R172H/\Delta E2-10}$-mutant and $p53^{\Delta E2-10/\Delta E2-10}$-null cells in the brain, respectively (Fig. 1c, d and Supplementary Fig. 1e–i). Transcriptomic analysis between *p53*CKO malignant gliomas/tumor-derived cell lines and human GBM subtypes (see "Methods") revealed that GEM malignant gliomas and GBMs strongly expressed a dominant proneural signature (Fig. 1e, f and Supplementary Fig. 1j)[1,22,23]. Furthermore, we found no somatic mutation in *Idh1, Idh2*, or *H3f3a* (mutated in pediatric GBMs) in malignant gliomas and GBMs from all three models, which were more similar to the human *IDH*-WT GBMs without exhibiting G-CIMP (Fig. 1g and Supplementary Fig. 1k, l)[22,24]. Unlike human *TP53*-mutant *IDH*-WT proneural GBMs that frequently exhibited genetic alterations of *PDGFRα* (Supplementary Fig. 1m)[1], no evidence of genetic *Pdgfrα* abnormality was found in malignant gliomas and GBMs from all three *p53*CKO models. Instead, we observed aberrantly high levels of *Pdgfrα*/Pdgfrα expression at both mRNA and protein levels in ~50% of the tumors analyzed, suggesting a nongenetic mechanism of activating Pdgfrα signaling (Fig. 1d and Supplementary Fig. 1n). In summary, all three *p53*-mutant genotypes equally and sufficiently induce malignant gliomas and GBMs, resembling proneural *IDH*-WT GBMs with both high and low levels of Pdgfrα signaling[21,22].

### The two growth patterns versus two clonal nonreciprocal translocation (cNRT) acquisition patterns.
To determine the in vivo growth patterns, we performed serial magnetic resonance imaging (MRI) screens once a week from 5.5 to 12.5 months of age, detecting early glioma lesions (0.2–10 μl) in vivo (Fig. 2a, b). The initial lesions were detected after 6–12 months but underwent rapid tumor growth, leading to mortality within 1–2 months of initial detection (Fig. 2a, b). Three-dimensional (3D) reconstruction of the serial MRI data revealed two distinct patterns in these rapidly growing tumors (Fig. 2b and Supplementary Movies 1–4). The Type 1 pattern, growing as a single mass throughout the entire screening process, was observed in ~30% of 43 tumor-bearing brains analyzed by this approach (Fig. 2b, c and Supplementary Movie 1). In contrast, the Type 2 pattern was characterized by rapid growth of multiple tumors at spatially segregated sites (Fig. 2b, c and Supplementary Movies 2–4). Of note, we observed spatially segregated tumors with different degrees of merging in 13 of the 30 Type 2 cases, either partially

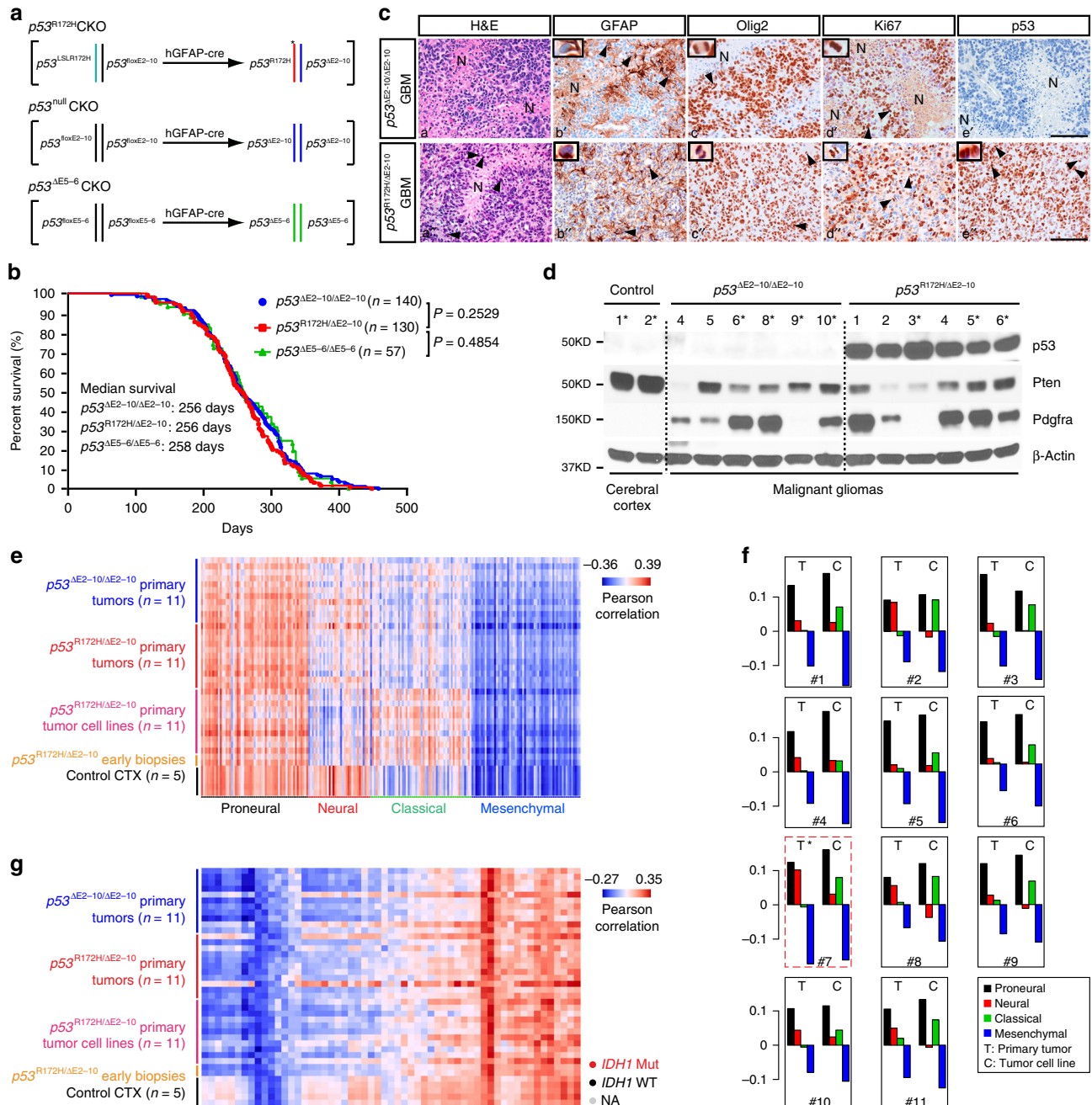

**Fig. 1 *p53*CKO GEM models for human primary *TP53*-mutant *IDH*-WT GBM. a** Genetic configurations of hGFAP-cre-mediated recombination in the generation of three *p53*CKO models. **b** Kaplan–Meier survival curves for three *p53*CKO models. The Mantel–Cox (log-rank) test was used to statistically compare the overall survival. **c** Adjacent sections from representative brains with end-stage GBMs from *p53*null CKO and *p53*R172H CKO mice were stained for H&E, GFAP, Olig2, Ki67, and p53. Pseudopalisading cells were observed adjacent to necrotic areas (a′–e′, a″). The arrows and inset show the tumor cell with perinuclear GFAP staining (b′, b″). Arrows and insets show abnormal mitotic figures at anaphase and/or metaphase in tumors (c′, d′, c″–e″). N necrosis. Scale bars: 50 μm. **d** Western blotting analyzed p53, Pten, and Pdgfrα levels in parenchymal gliomas/GBMs from *p53*null CKO (*n* = 6) and *p53*R172H CKO (*n* = 6) mice. Normal adult cortical tissues from hGFAP-cre+; *p53*floxE2-10/+ (*n* = 2) mice were used as controls (Ctr). "*" marks the samples with both western blotting (**d**) and microarray (Supplementary Fig. 1n, red points) analysis. Source data are provided as a Source data file. **e** Gene expression profiles of primary tumor tissues of *p53*null CKO mice (*n* = 11), paired primary tumor tissues and primary tumor-derived cell lines of *p53*R172H CKO mice (*n* = 11), primary tumor-derived cell lines from early stage *p53*R172H CKO biopsies (*n* = 2), and normal cortical tissues (CTX, *n* = 5) were compared to human GBMs (*n* = 202) using the classification system established by Verhaak et al.[23]. Red indicates strong sample-wise Pearson correlation; blue indicates the reverse. See methods for details. **f** Gene expression profiles of paired *p53*R172H CKO primary tumor and primary tumor-derived cell lines were scored based on their gene expression similarity to four human GBM subtypes. Positive values indicate similarity. Primary tumor #7 (red box, *) was documented by histology with normal tissue contamination prior to microarray analysis. **g** Gene expression profiles of the same set of samples from (**e**) were compared to human proneural subtype GBMs with or without *IDH1* mutations (red or black dots). See methods for details.

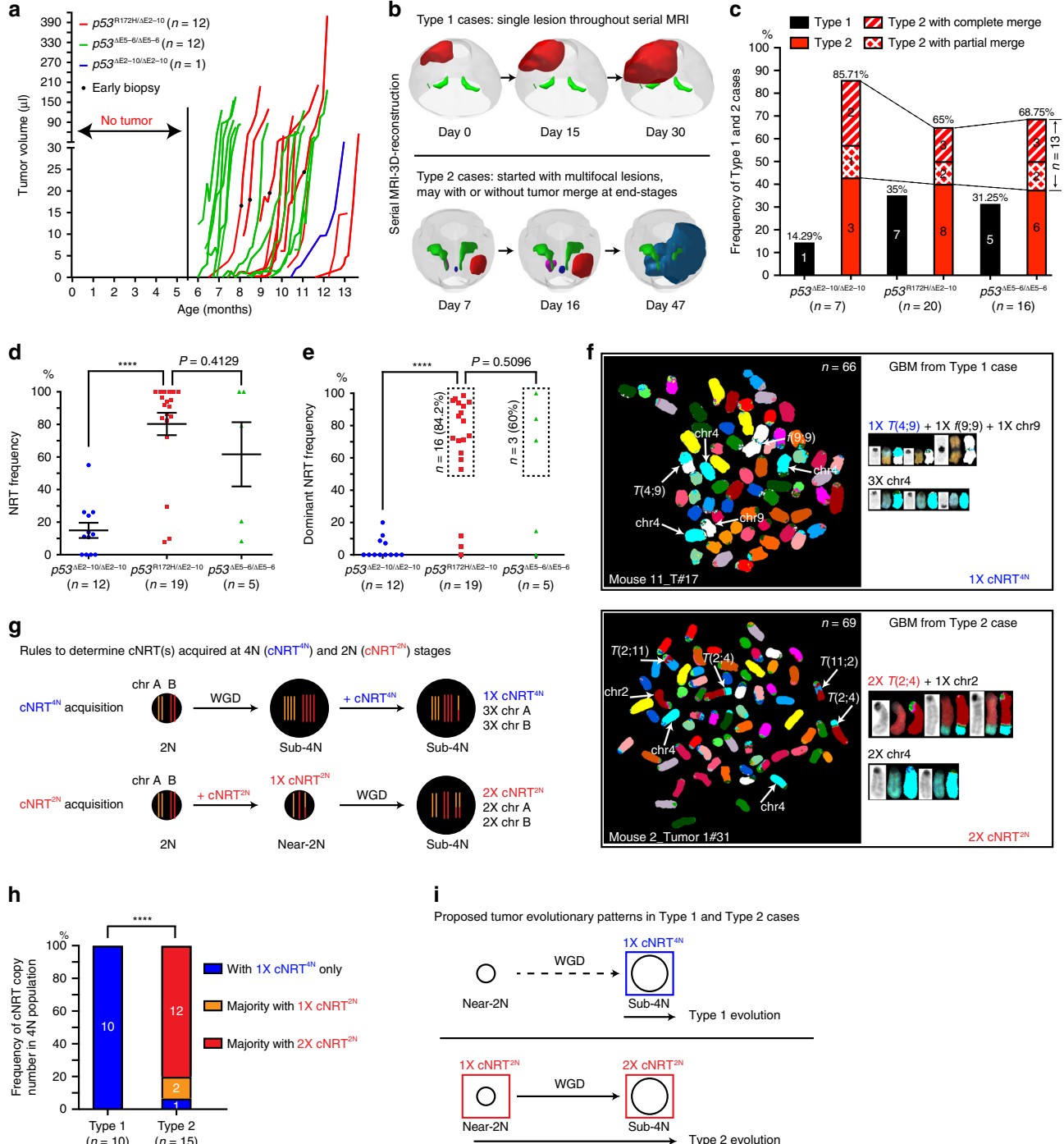

**Fig. 2 Two growth patterns are associated with two cNRT acquisition patterns. a** In vivo tumor growth curves of three *p53*CKO models were determined by serial MRI. In mice with multifocal masses, the fastest-growing tumor is shown. **b** Representative 3D-reconstruction of MRI images shows the single-mass Type 1 (upper panel) and multifocal-mass Type 2 (bottom panel) growth patterns. **c** Frequency of Type 1 and Type 2 growth patterns in three *p53*CKO models was determined by serial MRI. The single-mass Type 1 pattern was marked by black. The multifocal-mass Type 2 pattern was presented by three types: (1) no merge (red), (2) partial merge (red with diamonds), and (3) complete merge (could be misdiagnosed as Type 1, red with stripes). The percentage of cells from each tumor sample carrying a nonreciprocal translocation (NRT) (**d**) and the percentage of cells carrying the dominant NRT (the NRT with the highest frequency in a tumor sample) (**e**) were compared. Each dot represents a tumor sample. Data are presented as a dot plot with mean ± SEM. **f** Representative spectral karyotyping (SKY) images identify clonal NRT(s) (cNRTs, observed in over 50% of the metaphases analyzed in a given sample). Analysis of SKY images demonstrates that a sub-4N Type 1 tumor cell at end-stage contained only one copy of its cNRT (upper panel), while a sub-4N Type 2 tumor cell typically contained two copies of its cNRT (bottom panel). **g** Schematic illustration of a cNRT acquired in a cell with a 4N genome (cNRT$^{4N}$, blue) or acquired in a cell with a 2N genome (cNRT$^{2N}$, red). **h** The percentage of sub-4N tumors in Type 1 and Type 2 cases carrying cNRT$^{4N}$(s) and/or cNRT$^{2N}$(s) was shown. **i** Proposed tumor evolution patterns in Type 1 and Type 2 cases. Briefly, Type 1 tumors could only be traced back to a cNRT$^{4N}$-bearing cell with a sub-4N genome (blue square), while Type 2 tumors could be traced back to a cNRT$^{2N}$-bearing cell with a near-2N genome (red squares). Unpaired, two-tailed Student's *t* test was used for statistical analysis in (**d**, **e**, **h**). ****$p < 0.0001$.

(38%) or completely (62%) (labeled by colored dashed lines, Fig. 2c). To determine whether these GEM GBMs exhibit chromosomal abnormalities frequently seen in human cancers[25,26], we employed spectral karyotyping (SKY) analysis. Malignant gliomas and GBMs isolated from the brain parenchyma of all three p53CKO models were predominantly comprised of cells harboring sub-tetraploid (sub-4N) genomes, typically 60–80 chromosomes, as compared to normal neural stem cells with diploid (2N) or near-2N genomes (Supplementary Fig. 2a–c). This phenotype is consistent with whole-genome duplication (WGD) followed by limited chromosome loss, which is more frequently observed in TP53-mutant human cancers, including GBMs, than those with WT TP53 (Supplementary Fig. 2d, e)[1,27,28]. Many chromosomal abnormalities, including chromosomal fusions, were present at similar rates in malignant gliomas/GBMs from all p53CKO models (Supplementary Fig. 2f–j). Importantly, cNRTs, the NRTs observed in >50% of metaphase cells per tumor analyzed, were specifically identified in most malignant gliomas and GBMs driven by p53[R172H] and p53[ΔE5–6], but not p53[null] mutations (Fig. 2d–f and Supplementary Fig. 2f–h). Together, we identified additional oncogenic activity exerted by p53-mutant alleles beyond loss of functions, which promotes malignant gliomas and GBMs with higher levels of chromosomal abnormalities characterized by cNRT(s). We thus focused on the p53[R172H] model for further investigation.

Two patterns of cNRTs were observed in sub-4N tumor cells, featuring either one copy of the NRT with three copies of the translocated chromosomes or two copies of the NRT with two copies of the translocated chromosomes (Fig. 2f–h). These two patterns represent cNRT acquisition in cells with a 4N or 2N genome, termed cNRT[4N] and cNRT[2N], respectively (Fig. 2g). We analyzed a cohort of 13 cases with 7 single-mass Type 1 cases and 6 multifocal-mass Type 2 cases determined by histological analysis, of which 6 cases (three for each type) were validated by serial MRI/3D reconstruction (Fig. 2h, Table 1, and Supplementary Fig. 3a,4). All Type 1 tumors carried at least one cNRT[4N](s), but no cNRT[2N] (Fig. 2h and Supplementary Fig. 3a). In contrast, the presence of at least one cNRT[2N](s) was observed in 14 of 15 Type 2 tumors (Fig. 2h and Supplementary Fig. 4). Together, the single-mass Type 1 versus multifocal-mass Type 2 pattern correlates with the acquisition pattern of cNRTs in a founder cell/clone (FC) with a 4N versus 2N genome, respectively (Fig. 2i).

**Spatial segregation in tumor initiation and manifestation**. To determine the existence of 2N tumor precursor cells in Type 1 tumors, we analyzed chromosomal profiles of tumor cells isolated from multiple regions of individual tumors, including tumor-free SVZ regions as a control (Supplementary Fig. 3a). Whereas tumor-free SVZ-derived cells predominantly had normal 2N genomes and normal chr19, all Type 1 tumors exhibited loss of Pten/chr19 with sub-4N genomes (eight segments from seven tumors) (Supplementary Fig. 3b). Furthermore, MRI-guided early biopsies exhibited the critical histopathological and genetic features of end-stage high-grade gliomas, featuring sub-4N genomes and loss of Pten/chr19 (Supplementary Fig. 3c–e). Thus, similar to most human primary GBMs, radiographically detectable Type 1 tumors have no evidence of tumor precursor populations, exhibiting a rapid one-phase evolutionary pattern driven by a sub-4N/Pten/chr19-loss FC and growing as a highly proliferative single mass. In contrast, tumor precursor populations with near-2N genomes were observed in the SVZ, which was spatially segregated from parenchymal tumors, in five of the six Type 2 cases analyzed (Fig. 3a and Supplementary Fig. 4). Strikingly, three of the Type 2 cases (Mice 2, 3, and 6) maintained tumor

precursor populations with near-2N genomes and normal Pten/chr19 in the SVZ, which carried one or more cNRT[2N](s) identical to those observed in autologous sub-4N/Pten/chr19-loss tumors (Fig. 3a). In one Type 2 case (Mouse 3), Pik3ca[H1047R], a hotspot mutation known to activate PI3K/Akt/mTORC1 signaling in human cancers[29], was exclusively found in more advanced tumors, but not the relatively tumor-free SVZ (SVZ[L]). In the other three Type 2 cases, Mouse 5 and Mouse 10 maintained a small number of tumor cells with near-2N genomes, while all four segments from three sites of Mouse 4 were entirely comprised of sub-4N/Pten/chr19-loss tumor cells (similar to Type 1 tumors). Of note, all tumor cells from those three Type 2 cases with near-2N or sub-4N genomes exhibited loss of Pten/chr19. Importantly, the near-2N/Pten/chr19-loss tumor precursor cells in Mice 5 and 10 also carried one copy of the cNRT[2N](s) (cNRT[2N-1×]) identical to the one(s) presented with two copies observed in autologous sub-4N/Pten/chr19-loss tumors (cNRT[2N-2×]) (Fig. 3a and Supplementary Fig. 4). Together, these observations support the model wherein Type 2 tumors arise from a FC with a 2N genome that acquires one copy of cNRT[2N](s) before undergoing WGD.

Consistent with intact Pten/chr19, the expression of Pten protein was significantly higher in SVZ-derived cells than those derived from autologous parenchymal tumors with Pten/chr19 loss (Fig. 3b, c and Supplementary Fig. 5a). By comparison, loss of Pten protein expression was accompanied by the activation of PI3K/Akt/mTORC1 signaling in autologous tumors isolated from the brain parenchyma (Fig. 3c, d and Supplementary Fig. 5b–e). Although Mouse 3 tumors exhibited high levels of Pten expression, the acquired Pik3ca[H1047R] mutation appeared to activate PI3K/Akt/mTORC1 signaling (Fig. 3b and Supplementary Fig. 5d)[1,29]. Importantly, the SVZ of the three Type 2 cases with near-2N/Pten/chr19-normal tumor cells was free of high-grade gliomas, and some cases, except for increased numbers of Olig2[+] cells, resembled a normal SVZ stem-cell niche (Fig. 3e and Supplementary Fig. 5f)[19]. Overall, the SVZ in these three Type 2 cases exhibited considerably lower cellularity and mitotic index than that observed in malignant gliomas and GBMs spatially segregated in the parenchyma of the same brains (Fig. 3f). Despite no significant difference in cNRTs observed among all tumor samples analyzed, both Type 1 and Type 2 tumors observed in the brain parenchyma showed a significantly more diverse population of non-cNRTs and consequently total NRTs than those from SVZ-derived tumors with near-2N genomes in Type 2 cases (Fig. 3g, h and Table 1). Thus, the Type 2 pattern exhibits a two-phase tumor evolution—an "early" phase with normal Pten/chr19 and relative stable near-2N genomes in the SVZ and a "late" phase with Pten/chr19 loss and highly unstable sub-4N genomes in the spatially segregated distant brain parenchyma, while the Type 1 pattern only has a one "late" phase during tumor evolution (Fig. 3i).

**Building single-cell phylogenetic trees by the SKY method**. The maintenance of cNRT[2N](s) throughout tumor evolution, together with WGD, allowed us to build a phylogenetic tree to delineate clonal origin and dispersal from the SVZ to distant locations in multifocal-mass Type 2 cases. We integrated SKY-based single-cell analysis with the whole-genome sequencing (WGS) data of bulk tumor samples. The average chromosome counts of the cells analyzed by SKY showed excellent agreement with the read depth profile in the WGS data at the genome-wide level, which correlates with copy-number variations (CNVs) (Fig. 4a, b). For example, in the Type 2 case with the most complex chromosomal profile (Mouse 2), we validated three chromosomal events determined by WGS and SKY data: (1) 50% loss of chr16 in all tumor cells, (2) normal and loss of Pten/chr19 in SVZ- and

**Table 1 Overview of SKY results of parenchymal tumors and SVZ samples from seven Type 1 and six Type 2 cases from *p53*[R172H] CKO models.**

| Sample name | WGD | cNRTs | Non-cNRTs and RTs | Fusions; ring chromosomes |
|---|---|---|---|---|
| *Type 1 cases (n = 7)* | | | | |
| Mouse 8 | | | | |
| 130620#1 | | | | |
| **T_Late P2** | Yes | t(11;7) t(10;4) | t(17;7;14) t(4;7;16) t(Y;2) t(13;Y) t(11;14) t(10;16)? t(1;7;12) t(5;7;5) t(1;7;1) t(15;19) t(1;4;3) t(15;4) t(1;2) t(1;7) t(15;7) | f(3;3) f(12;12) |
| SVZ-L P2 | No | – | – | – |
| SVZ-R P2 | No | – | – | – |
| Mouse 15 | | | | |
| 121211#1 | | | | |
| **T P5** | Yes | t(2;17) t(7;14) t(8;11) | t(X;13) t(5;3) t(4;1) t(8;2) t(8;15) t(2;7)& t(7;2) t(8;14) t(4;11) t(12;17) | – |
| SVZ-L P1 | No | – | – | – |
| Mouse 9 | | | | |
| 130504#1 | | | | |
| **T P2** | Yes | t(14;X) | t(8;15) t(6;11) t(16;16) t(11;6) t(4;18) t(17;19) t(13;3) t(2;10) t(1;7) t(9;X) t(15;14) t(15;5) t(5;16) t(10;11) t(4;7) t(2;18) t(15;6) t(14;15) t(1;14) t(X;12) t(4;8) | ring(3;3) ring(7;17) f(5;15) ring(1;1) f(5;5) f(13;13) ring(16;15) ring(12;12) ring(15;15) ring(14;11) f(3;1) f(9;7) |
| SVZ-L P4 | No | – | t(8;8) t(14;11) t(17;12) t(1;8) t(5;11) t(15;3) t(14;4) t(5;15) t(13;19)? t(5;Y) t(15;Y) | f(8;3) ring(10;10) ring(14;11) |
| Mouse 16 | | | | |
| 130125#3 | | | | |
| **T P4** | Yes | t(9;11) | t(4;4) t(4;9)? t(4;X) t(18;1)? t(X;10)? t(X;9) t(11;X) t(1;2) t(X;3) t(16;X)? t(4;6) t(11;X)& t(X;11) t(9;1) t(14;3) t(2;10)? t(14;4) t(14;10) t(16;5) t(X;12) t(2;3) t(4;6)? | f(8;8) f(7;7) f(16;16) ring(4;4) ring(9;4) |
| **SVZ-L P4** | Yes | t(11;8) t(6;6) | t(5;3;5) t(3;16) t(6;18) t(1;7) t(6;6;8) t(8;6) t(4;14) t(4;3;11) t(3;15) t(9;17) t(4;3) t(6;6;9) t(3;9) t(8;11) t(4;5) t(3;8) t(9;17) t(12;5;3;5) t(15;10) t(3;17) t(3;7) t(8;18) t(3;12) t(8;6)&t(6;8) t(1;2) | f(8;8) f(10;14) ring(4;6) ring(6;15) ring(3;3) |
| Mouse 13 | | | | |
| 121029#1 | | | | |
| **T_Late P1** | Yes | – | t(X;8) t(8;2) | f(7;11) ring(8;8) ring(3;8;8) ring(1;1) |
| **T_Early P3** | Yes | t(8;2) | t(2;6) t(5;17) t(4;4;16) t(14;16) t(12;17) t(14;1) t(14;10) t(19;15) t(1;5;1) t(12;X) t(1;2) t(8;1) t(X;14) t(10;15) t(4;2) t(14;2) t(4;19) t(3;14) t(17;3) t(14;3) t(12;18)&t(18;12) | – |
| **SVZ-L P2** | Yes | – | t(1;8) t(8;7) t(8;X) t(6;1) t(X;3) t(X;17) t(X;15) t(X;1) t(8;14) t(6;5) t(13;16) t(1;5) t(6;12) t(3;8) t(3;2) t(8;15) t(2;2) t(15;6) | f(13;1) f(17;19) f(19;5) f(19;19) f(9;9) f(6;12) f(19;4;19)? f(5;9)? f(3;9) |
| Mouse 11 | | | | |
| 130312#10 | | | | |
| **Tumor_A P2** | Yes | t(4;9) | t(13;6) t(11;19) t(15;13) t(14;8) t(14;5) t(1;17) t(4;6) t(1;7) t(4;10) t(2;5) t(2;5) t(12;13) t(11;6) t(X;10) t(11;2) t(12;17) t(4;8) t(X;2) | f(X;19) f(3;9) f(X;9) f(15;12) f(13;13) f(7;9) f(8;10) ring(12;6) ring(7;7) ring(1;X) |
| **Tumor_P P2** | Yes | – | t(2;18;6) t(7;1) t(4;6) t(13;11) t(3;X) t(14;8) t(13;5) | f(19;19) ring(X;4) |
| **SVZ-L P1** | Yes | – | t(19;17) t(10;17) t(4;19) t(X;9) t(4;X) t(4;5) | f(7;7) ring(12;12) |
| Mouse 1 | | | | |
| 120123#1 | | | | |
| **T P2** | Yes | t(7;17) | t(2;11) t(7;4) t(11;2) t(11;1) t(10;2) t(1;8) t(2;10) t(11;3) t(14;1) t(1;7;17) | ring(11;11) ring(4;4) ring(14;14) ring(13;13) |
| **SVZ-L P1** | Yes | t(7;17) | t(17;7) t(5;14) t(14;11) t(8;18) t(1;10) t(12;4) t(11;14) | ring(11;11) ring(4;4) ring(13;13) ring(1;1) ring(10;10) ring(2;2) |
| *Type 2 cases (n = 6)* | | | | |
| Mouse 2 | | | | |
| 120301#1 | | | | |
| **T1 P1** | Yes | **t(2;4)** | t(2;4)[a] t(14;15) t(3;11) t(11;17) t(17;X) t(9;2) t(9;3) t(3;3) t(7;12) t(2;11)&t(11;2) | f(18;14) |
| **T2 P2** | Yes | **t(2;4)** | t(3;4) t(14;5) t(5;14) t(7;8) t(10;11)[a] | f(18;14) f(11;18) |
| **SVZ-L P2** | No | **t(2;4) t(3;8) t(10;11)** | t(8;Y) | – |
| **SVZ-R P2** | No | – | t(2;4) t(1;12) t(12;2) t(10;19) t(11;13) t(15;1;Y) t(4;7) t(8;11) | – |
| Mouse 3 | | | | |
| 120417#1 | | | | |
| **TR P2** | Yes | **t(11;11)** | t(9;11) t(1;9) t(3;1) t(17;7) t(4;7) | – |
| **TL P3** | No | – | t(11;11) t(X;1)&t(1;X) t(14;4) t(7;14) t(9;16) t(X;16) | f(X;11) f(8;14) ring(11;11) |
| **SVZ-R P2** | No | **t(11;11)** | t(2;5) t(3;11) t(3;16) | – |
| **SVZ-L P3** | No | – | t(11;11) | f(18;2) |
| Mouse 6 | | | | |
| 120717#1 | | | | |
| **T_Late P4** | Yes | **t(11;11)** | t(4;2) t(7;17) t(17;2) t(2;13) t(9;11) t(4;7) | f(11;11) ring(14;14) ring(13;19) ring(2;19) ring(5;7) ring(11;4) |
| **T_Early P4** | Yes | **t(11;11)** | t(11;19) t(13;19) t(X;7) t(7;3) t(7;11) t(16;14) t(3;16) t(7;X;7) t(15;13) t(X;15) t(3;1) t(16;11) t(X;3) t(4;3) t(12;19) t(9;11) | ring(14;14) |
| SVZ-L P4 | No | **t(11;11)** | t(14;17) | ring(14;14) ring(1;15) |
| SVZ-R P2 | No | – | – | – |
| Mouse 5 | | | | |
| 140412#1 | | | | |
| **T P3** | Yes | **t(7;7)** | t(4;2) t(17;3) t(3;2) t(2;8) t(3;5) | – |
| SVZ-L P1 | No | – | – | – |
| **SVZ-R P1** | Yes | **t(7;7)** | – | – |

**Table 1 (continued)**

| Sample name | WGD | cNRTs | Non-cNRTs and RTs | Fusions; ring chromosomes |
|---|---|---|---|---|
| Mouse 10 121109#1 | | | | |
| **T1 P3** | Yes | *t*(9;11) *t*(11;2) | *t*(16;11) *t*(8;4;8) *t*(6;2) *t*(5;4)&*t*(4;5) *t*(4;7;4) *t*(4;7) *t*(4;6) *t*(16;7)&*t*(7;16) *t*(15;13) *t*(12;13) *t*(10;17) *t*(14;X) | *f*(X;18) *f*(15;12) *f*(X;X) *f*(15;15) ring(3;8) |
| **T2 P4** | Yes | – | *t*(10;17) *t*(3;X) *t*(1;X) *t*(X;6) *t*(5;1) *t*(15;13) *t*(10;8;17) *t*(10;8;15) *t*(10;8;1) *t*(X;8) *t*(18;4) *t*(9;8) *t*(1;14) | *f*(9;18) *f*(4;5) |
| **SVZ-R P3** | Yes | ***t*(2;16) *t*(7;4) *t*(10;7)** | *t*(4;15) *t*(15;5); *t*(5;15;5) *t*(4;7)? *t*(14;4) *t*(15;X) *t*(15;19) *t*(15;12) *t*(15;19) | *f*(15;15) ring(2;15) ring(15;19) ring(18;18); ring(15;15) |
| Mouse 4 120911#2 | | | | |
| **T1 P3** | Yes | ***t*(12;6)** | *t*(11;4;3) *t*(19;11) *t*(19;12) *t*(12;6)[a] *t*(2;9) | – |
| **T2 P3** | Yes | ***t*(12;6)** | *t*(11;4;3) *t*(8;11) *t*(15;2) *t*(11;4) *t*(13;19) *t*(3;2) *t*(16;3) *t*(15;11) *t*(1;12) *t*(8;3) *t*(12;13) *t*(11;18) *t*(11;X) *t*(12;5) | – |
| **SVZ-L P3** | Yes | ***t*(12;6)** | *t*(11;4;3) *t*(12;19) *t*(8;3) *t*(X;5) *t*(15;16) *t*(15;11) *t*(17;11) *t*(1;10) *t*(2;6) *t*(18;5) | *f*(15;15) ring(14;14) ring(4;4) |
| **SVZ-R P2** | Yes | ***t*(12;6)** | *t*(11;4;3) *t*(15;11) *t*(4;11) *t*(3;7) *t*(9;1) | – |

Bolded "Sample Names" represent tumor tissue. Bolded "cNRTs" represent cNRT[2N]
[a]These NRTs potentially have different breakpoints compared to the cNRTs in that sample.

parenchyma-derived tumors, respectively, and (3) acquisition of the founding cNRT[2N], *t*(2;4), accompanied by loss of chr4, in all tumor cells except for 62.5% of SVZ[R]-T-derived cells. Thus, these results demonstrate that the small number of metaphase cells analyzed by the SKY method is representative of the diversity of the overall tumor cell populations, suggesting no significant sampling bias from the SKY method. More importantly, we used WGS data to confirm that cNRTs from autologous SVZ- and parenchyma-derived tumors shared identical breakpoint sequences. All clonal or sub-clonal NRTs used for phylogenetic tree building were likely confirmed by the WGS data ($n = 26$). For further verification, we cloned the breakpoint sequences of one cNRT[2N], *t*(2;4), and performed Sanger sequencing and quantitative PCR, confirming that the *t*(2;4) breakpoint location was identical in all four tissue samples from Mouse 2 (Fig. 4c–e). The identical DNA sequence at the breakpoint location of NRTs further supports the model wherein malignant gliomas and GBMs arises from a single FC that acquires the same genetic event, *t*(2;4). Moreover, Monte Carlo simulations predict that the same "hotspot" NRT is unlikely to recur in tumors from two different mice, indicating that almost no NRT(s) is independently acquired in two different tumors (Fig. 4f, see "Methods" for details). Together, these results demonstrate that cNRT can serve as a lineage-specific marker to clonally link tumor cells from different regions of the same brain during tumor evolution.

**The mode of the two-phase evolutionary pattern.** We next used a neighbor-joining (NJ) computational algorithm to build a single-cell phylogenetic tree (Supplementary Fig. 6a)[30]. We converted chromosomal events into a distance matrix under the minimum evolution principle for each sample in all Type 2 cases (see "Methods" for details). Given that tumor precursor cells at the "early" phase of tumor evolution were maintained in the SVZ, we first analyzed the NJ tree from 40 tumor cells of the SVZ[R] of Mouse 2 (Fig. 5a). When cells are arranged based on genetic similarity, we expect that cells with a shared FC are grouped together, forming identifiable clusters, each of which could be traced back to a single FC. In the SVZ[R] of Mouse 2, we identified four clonally related clusters which followed an evolutionary path based on the gradual accumulation of somatic chromosomal events (Fig. 5b, c). Tumor evolution began with chr16 loss (shared by all 40 cells) that defined the oldest FC with fewest chromosomal abnormalities (FC0). Subsequent FC(s) developed by accumulating additional chromosomal events: FC1 acquired one copy of cNRT[2N-1×][*t*(2;4)]; the predicted FC2 lost chr18; FC3

underwent WGD, including duplication of *t*(2;4) (cNRT[2N-2×]); and FC4 acquired a cNRT[4N] [*t*(1;12)] (Fig. 5b, c). Next, we analyzed the NJ tree from tumor cells of the SVZ[L] of Mouse 2. All SVZ[L]-T-derived tumor cells were traced back to a single FC carrying three cNRT[2N-1×]s, including *t*(2;4), suggesting that SVZ[R]-FC1 acquires two more cNRT[2N]s and generates the entire cell population of SVZ[L]-T (Supplementary Fig. 6b, c). Compared to the linear evolutionary path of the SVZ tumors, both parenchyma-derived tumors were seeded by multiple clonally related FCs and lineages. Tumor 1 and Tumor 2 could be traced back to a total of seven and nine FCs, respectively (Supplementary Fig. 6d–g). Two of 16 FCs (both observed in Tumor 1) had near-2N genomes, and only one had normal *Pten*/chr19. However, this near-2N/*Pten*/chr19-normal FC was not the direct precursor for the other 15 FCs, as it acquired a distinct chromosomal event: chr3 gain. Instead, all 16 FCs could be traced back to the FC located in the SVZ[R]-FC1. Of note, immediate precursors of the two subpopulations in Tumor 2 were observed in Tumor 1 (Fig. 5c and Supplementary Fig. 6e, g). Together, these results reveal a two-phase evolutionary pattern starting from the SVZ[R] with a cNRT[2N-1×]-bearing FC, SVZ[R]-FC1, which undergoes local evolution while independently seeding three distant locations (SVZ[L], Tumor 1, and Tumor 2) (Fig. 5c).

To further establish a clonal relationship between SVZ[R]-derived cells before and after acquiring *t*(2;4), we cloned individual tumor cells into a 96-well plate, amplified single cells and performed WGS analysis of single-cell-derived bulk tumor samples (Supplementary Fig. 7). Using the seven single-cell-derived samples' CNV profiles, we built a NJ tree, which revealed that three clones without *t*(2;4) (Clones 1–3 with black circles) emerged earlier than four clones with *t*(2;4) (Clones 4–7 with red circles) during evolution (Fig. 5d). Moreover, regardless of having *t*(2;4), near-2N clones always emerged earlier than their respective sub-4N counterparts. These results not only validate the evolutionary direction of FCs by the SKY-based NJ tree, but also identify a precursor (SVZ[R]-FC0) of the cNRT[2N-1×]-bearing FC1, SVZ[R]-FC1, which harbors a cancer driver alteration: a specific homozygous deletion at *Nf1* tumor suppressor gene (Fig. 5d). Importantly, the NJ trees from two other Type 2 cases (Mouse 3 and Mouse 6) revealed a two-phase evolutionary pattern similar to that observed in Mouse 2 (Fig. 5e, f). Together, all three Type 2 cases show that cNRT[2N-1×]-bearing FC-derived tumor precursor cells with near-2N genomes and normal *Pten*/chr19 (Mouse 3 has normal *Pik3ca*[H1047] allele) drive local evolution in the SVZ, while multiple clonally related progeny of the CA cell seed one distant site ("polyclonal seeding") and/or

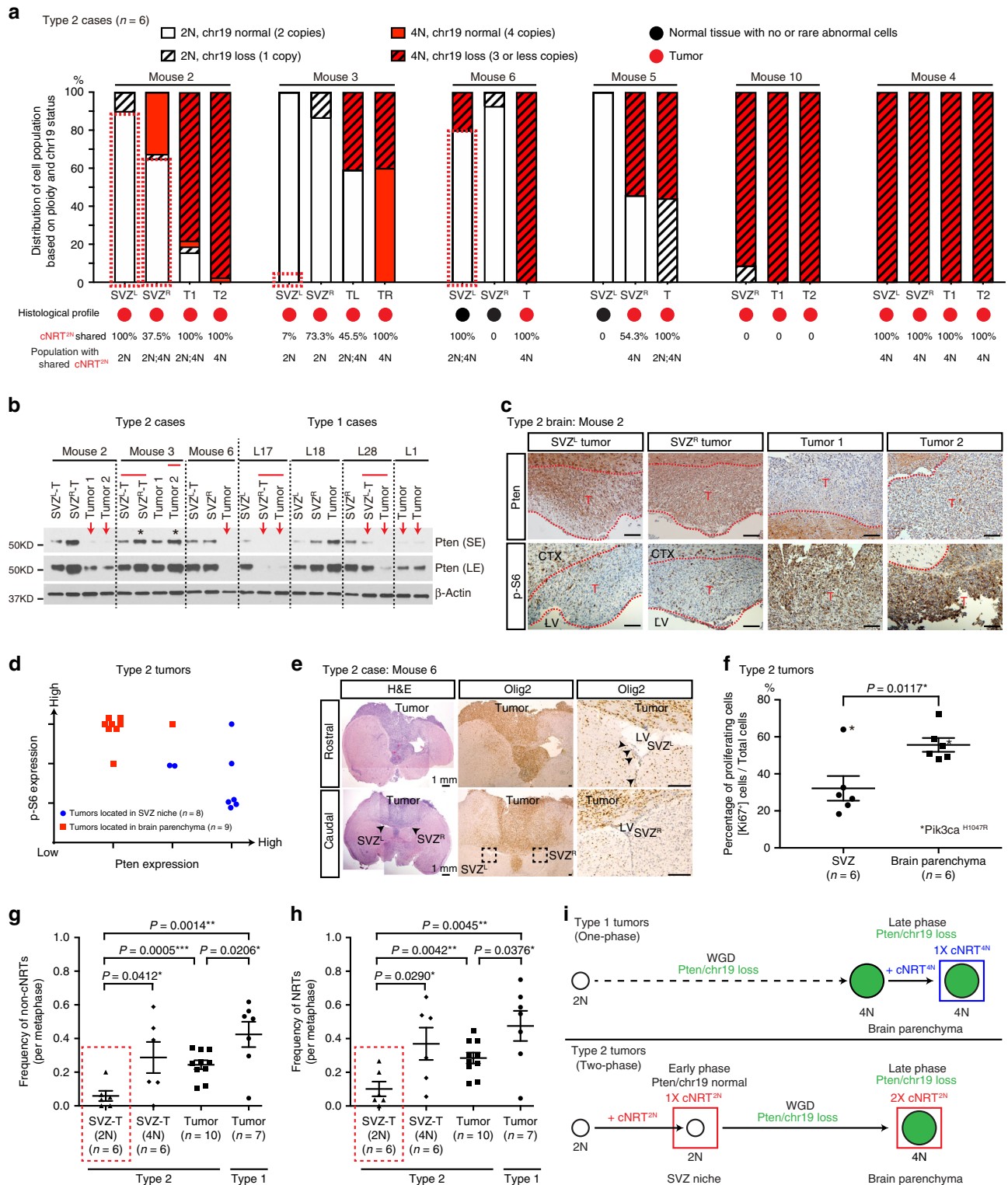

one such progeny seeds multiple distant sites ("parallel seeding"), generating autologous cNRT[2N-2×]-bearing with sub-4N genomes and loss of Pten/chr19 (Mouse 3 has mutant Pik3ca[H1047R] allele).

**Distant versus local dispersal from the SVZ to brain parenchyma.** The NJ trees of the other three Type 2 cases did not exhibit the complete two-phase evolutionary pattern described above. In Mouse 5, despite a lack of near-2N tumor precursors in

the SVZ, the observed cNRT[2N-2×]-bearing sub-4N cells had no additional chromosomal events. In contrast, parenchyma tumor-derived cells, including one near-2N population, all acquired additional chromosomal events, including one lineage with three events (one fusion and two chromosomal gains) and a second lineage with one cNRT[4N] (Fig. 6a). Thus, the two SVZ-derived FCs, one observed cNRT[2N-2×]-bearing sub-4N cell and one predicted cNRT[2N-1×] near-2N cell, distantly seeded the tumor in the parenchyma. Mouse 10 is a unique Type 2 case that had three

**Fig. 3 Spatial segregation in tumor initiation and manifestation in Type 2 cases. a** Ploidy and chr19 status at the single-cell level for early passage cNRT$^{2N}$-bearing cells derived from paired SVZ tissues and parenchymal Type 2 tumors are shown. Dashed red boxes indicate cNRT$^{2N}$-bearing cells with near-2N genomes and normal *Pten*/chr19. **b** Western blot reveals Pten in early passages of cell lines derived from paired SVZ tissues and parenchymal tumors. * indicates samples harboring Pik3ca$^{H1047R}$ mutation. Red lines indicate samples isolated from the same tumor masses. Red arrows indicate tumors identified with chr19 loss by SKY and/or chr19 FISH assay. Case L18 contained four copies of chr19. Source data are provided as a Source data file. **c** Matched Pten and p-S6 labeling in four spatially segregated tumors in Mouse 2. **d** Immunohistochemical staining of Pten and p-S6 in the SVZ niche and parenchymal tumors of Type 2 cases from the *p53*$^{R172H}$ CKO model were qualitatively scored. **e** Representative images show H&E (left) and Olig2 (middle) staining of parenchymal tumors and SVZ regions in Mouse 6. The boxed areas in SVZ regions are shown at high magnification (right). **f** Proliferation index measured by the percentage of Ki67$^{+}$ cells in the SVZ regions and brain parenchymal tumors were compared. * indicates samples harboring *Pik3ca*$^{H1047R}$ mutation. The frequency of non-clonal NRTs (non-cNRTs) (**g**) and NRTs (**h**) was compared in spatially segregated SVZ- and parenchyma-derived Type 2 tumors and Type 1 tumors. Dashed red lines marked Type 2 SVZ tumors with relatively stable genomes (2N). **i** Proposed tumor evolution patterns in Type 1 (top) and Type 2 (bottom) cases with defined *Pten*/chr19 and ploidy status. Data are presented as a dot plot with mean ± SEM. Unpaired, two-tailed Student's *t* test was used for statistical analysis in (**f**, **g**, **h**). *$p < 0.05$, **$p < 0.01$, ***$p < 0.001$. SE short exposure, LE long exposure, CTX cortex, SVZ subventricular zone, LV lateral ventricle, T tumor. Scale bars: 100 μm (**c**, **e**).

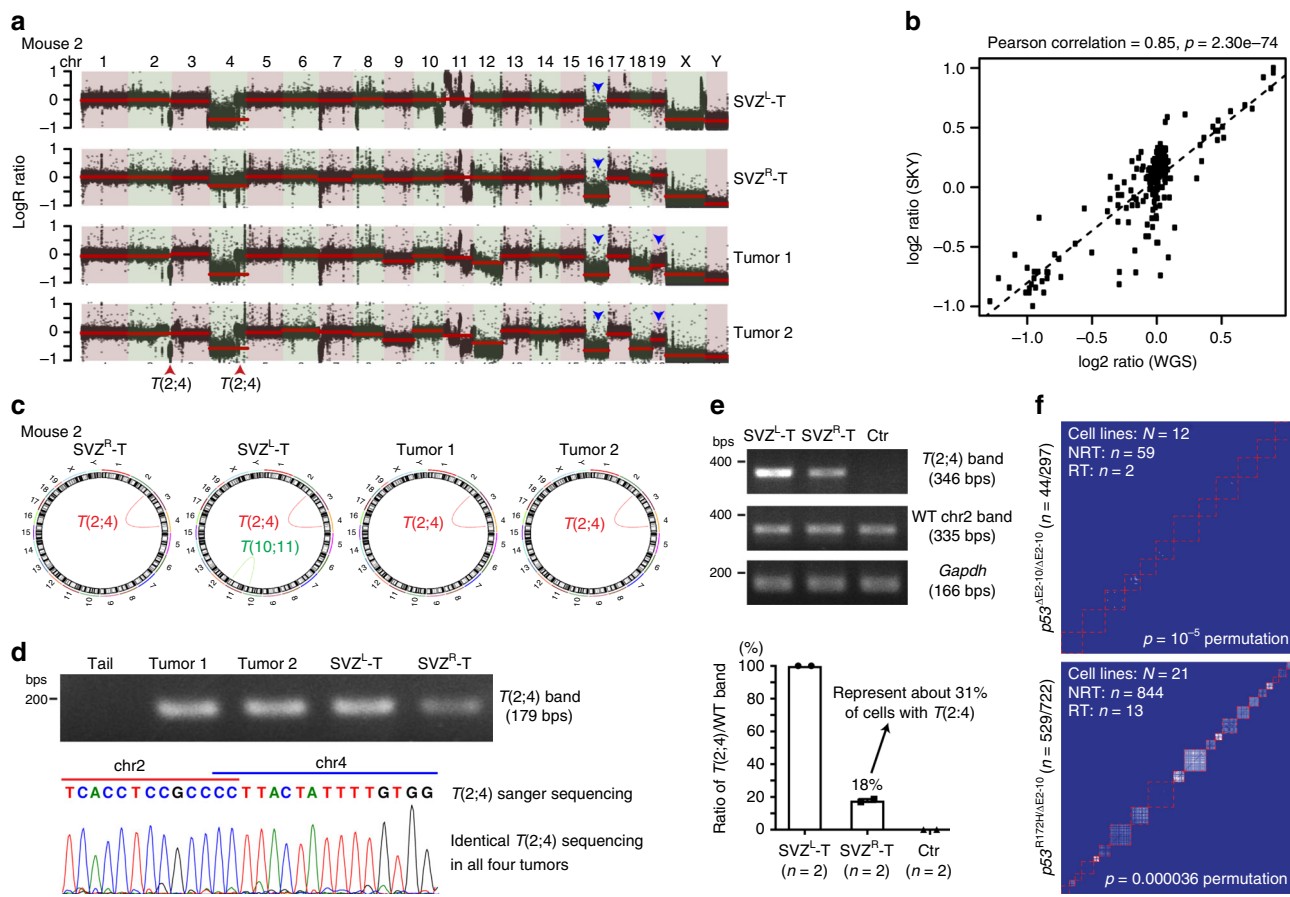

**Fig. 4 WGS validates SKY-based cytogenetic markers used for phylogenetic trees. a** The WGS-based copy number (CN) ratios of four tumor samples over tail (black points) from Mouse 2 were compared to the average of chromosome copy numbers obtained from SKY analysis (solid red segments). Arrowheads indicate the t(2;4) acquisition (red) and chromosome loss (blue). **b** Comparison of whole-chromosome CNV estimation (log2 ratio) between WGS and SKY using Pearson correlation coefficient (r). Dots represent each chromosome in each sample ($n = 13$). Dotted line represents a linear regression line between WGS and SKY estimations. **c** WGS analysis of primary cells from four tumors in Mouse 2 shows the interchromosomal rearrangement t(2;4) (red line) and t(10;11) (green line). **d**, **e** Sanger sequencing of t(2;4) and wild-type chr2 in Mouse 2 (**d**), with malignant glioma from other animal as control. q-PCR analysis shows the relative ratio of t(2;4) to wild-type chr2 between SVZ$^{L}$-T (100%) and SVZ$^{R}$-T (40%), exhibiting an excellent agreement with the SKY data (**e**). Source data are provided as a Source data file. **f** Sample-wise NRT sharing matrix was built for *p53*$^{null}$CKO and *p53*$^{R172H}$CKO gliomas. Shared NRTs between metaphases are displayed using a heatmap. Blue indicates no shared NRTs and red indicates shared NRTs. Red dashed squares indicate metaphases from a single mouse.

spatially segregated tumors, including one in the SVZ (Fig. 6b). However, two of the three tumors carrying at least one cNRT(s) not only exhibited distinct cNRTs, but also different patterns—the SVZ tumor had three distinct cNRTs$^{2N}$ and the parenchymal tumor had two distinct cNRTs$^{4N}$. Despite exhibiting no lineage relationship between SVZ- and parenchyma-derived tumors, SVZ-tumor-derived cells were comprised of both near-2N and sub-4N cells, carrying the same three cNRTs$^{2N}$ with one and two copies, respectively (Fig. 6b and Supplementary Fig. 4). These results suggest that Type 2 tumors could evolve locally from the

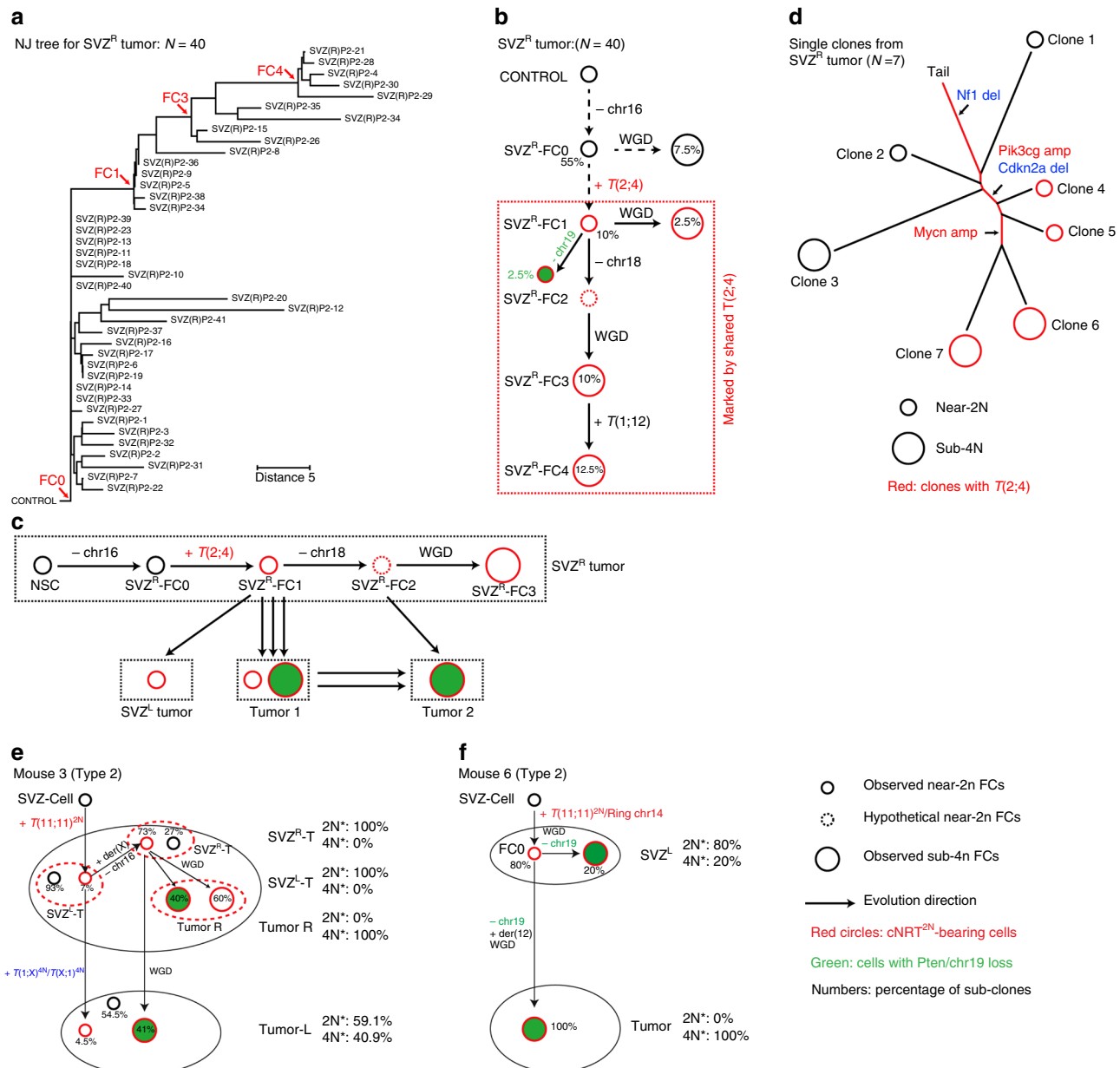

**Fig. 5 Single-cell phylogenetic trees reveal the two-phase evolution pattern in Type 2 cases. a** SKY-based single-cell phylogenetic tree was generated with all metaphases in Mouse 2 SVZ$^R$-T using a neighbor-joining algorithm with customized feature and distance measure selection (see "Methods" for details). Red arrows indicate major clusters. **b** The genotype of each founder cell/clone (FC) of the major clusters on the phylogenetic trees was extrapolated based on common genetic features, and their clonal relationship was illustrated. Major chromosomal events shared by cells in each branch are shown. **c** A schematic model shows the mode of distant dispersal in Mouse 2 from the SVZ$^R$ to the three distant sites via parallel seeding and polyclonal seeding based on interpretation of SKY-based single-cell analysis. **d** WGS data of seven single-cell-derived bulk tumor samples isolated from Mouse 2 SVZ$^R$-T were used to generate genome-wide CNV profiles, and a neighbor-joining tree was constructed to predict the evolutionary trajectory of these clones. Line length corresponds to difference in CNV profiles. Shared genetic events of known oncogenes or tumor suppressors are marked. See "Methods" for details. Models show a similar two-phase evolutionary pattern from the SVZ to distant brain regions in two additional Type 2 cases, Mouse 3 (**e**) and Mouse 6 (**f**).

SVZ to brain parenchyma. Finally, despite having no near-2N tumor cells in Mouse 4, the cNRT$^{2N-1\times}$, $t(12;6)$, was shared in all four tumor segments, suggesting the existence of an unobserved cNRT$^{2N-1\times}$-bearing/near-2N FC1 cell at tumor initiation, which were most frequently observed in the SVZ$^L$, from which we predict they were derived (Fig. 6c). Together, we demonstrate that in all Type 2 cases, the SVZ either exclusively or more abundantly harbors the closest descendant of the CA cell with a near-2N genome and fewer chromosomal events, generating malignant gliomas/GBMs with much more complex sub-4N genomes in the

brain parenchyma via distant dispersal, or less frequently, local expansion (Fig. 6d, e and Table 1).

To investigate the molecular mechanism underlying the two-phase evolutionary pattern in Type 2 cases, we preformed WGS analysis on early passage cell lines isolated from the SVZ and autologous tumors in distant brain regions from all six Type 2 cases. We categorized genetic alterations (e.g., focal deletions, amplification, and mutations) of known cancer drivers into two classes, shared and private events between SVZ- and parenchymal tumor-derived cells, which were manually placed on the trunks

and branches of each tree, respectively (see "Methods"). For the specificity of the selected alterations, the whole chromosomal loss (e.g., *Pten*/chr19 loss) or gain was not listed. We identified two types of cancer driver alterations in SVZ-derived cells from all six Type 2 cases. The three cases (Mice 2, 3, and 6) that maintained tumor precursor cells with normal *Pten*/chr19 and near-2N genomes all exhibited homozygous *Nf1* loss in early phases of tumor evolution (Fig. 6f–h). However, the other three Type 2 cases with no directly observed tumor cells with normal *Pten*/chr19 exhibited a distinct pattern of cancer driver alterations, characterized by amplification of oncogenes, including *Hras*, *Ccnd1*, and *c-Myc* in Mouse 5 tumors; *Fgfr2*, *Olig2*, *Foxo1*, and *Cdk4* in Mouse 10 tumors; and *Met*, *Ret*, *Ccnd2*, and *N-Myc* in Mouse 4 tumors (Fig. 6i–k). Thus, activation of receptor tyrosine kinase(RTK)/Ras-mediated Erk/MAPK signaling pathways is universally observed in both SVZ- and autologous parenchyma-derived tumors, suggesting an early event in the SVZ during the two-phase tumor evolution.

**Olig2⁺ progenitors underlie clonal expansion in the SVZ**. We investigated the role of loss of *Nf1* and/or activation of Erk/MAPK signaling during early evolution in the SVZ. Consistent with the WGS data of single-cell-derived tumors from SVZ[R]-T of Mouse 2 (Fig. 5d), homozygous deletion in the *Nf1* region (determined in the earliest FC, SVZ[R]-FC0) was shared among tumors from all four sites, accompanied by the complete absence of Nf1 protein expression (Fig. 7a, b). Moreover, WGS and protein expression analysis of bulk tumor samples was remarkably consistent with the SKY data from the other two Type 2 cases analyzed, including the SVZ[R] of Mouse 6 used as a positive control for the normal tissue (red arrows, Fig. 7a, b). The SVZ areas with fewer abnormal cells, such as the SVZ[L] and its closest tumor segment (TumorL) of Mouse 3 as well as the SVZ[L] of Mouse 6, retained relatively higher levels of Nf1 protein expression than their autologous more advanced tumors (e.g., SVZ[R] and TumorR of Mouse 3; Tumor of Mouse 6) (Fig. 7a, b). To investigate the phenotypic effects of *Nf1* loss in vivo, we used the expression of phosphorylated Erk (p-Erk) as a readout for activation of Ras-mediated Erk/MAPK signaling during early stages of tumor evolution in vivo[21,31,32]. We consistently identified p-Erk⁺ cells, presented as either individual cells in the SVZ or small clusters in the SVZ-associated areas, in age-matched *p53*[R172H]CKO and *p53*[ΔE5−6]CKO mice, but not in the SVZ of control mice (Fig. 7c, d). Importantly, the p-Erk⁺ cells were proliferating within the clusters of p53[Mutant]-expressing glioma precursors, as identified in our previous studies (Fig. 7e)[21,22]. While BrdU⁺ proliferating cells in the control SVZ were neuronal-restricted progenitors with no p-Erk or Olig2 expression, BrdU⁺ proliferating cells in the mutant SVZ and in glioma precursor clusters from age-matched *p53*[R172H]CKO and *p53*[ΔE5−6]CKO mice expressed Olig2 (Fig. 7f and Supplementary Fig. 8a)[19]. Consistent with our previous observation that most p53[Mutant]-expressing glioma precursors were positive for Olig2[21,22], mutant-specific p-Erk⁺ cells frequently expressed Olig2 with abnormal mitoses in the SVZ (Fig. 7g–i and Supplementary Fig. 8b, c). Thus, these observations are most consistent with a model wherein a single p53-mutant neural stem cell spontaneously loses *Nf1* or acquires other oncogenic mutations, activating Ras-mediated Erk/MAPK signaling pathway and promoting clonal expansion of p-Erk⁺Olig2⁺p53[Mutant]-expressing glioma precursors. Consistently, p-Erk⁺Olig2⁺p53[Mutant]-expressing glioma precursors expressed the markers for neural stem cells and transit-amplifying progenitors (TAPs), including Ascl1, Ezh2, and BLBP, but not more differentiated oligodendrocyte precursor cells (Fig. 7j)[19,21,31].

**Rictor/mTORC2 loss inhibits distant dispersal**. Since the status of *Pten*/chr19 distinguished "early" from "late" phase of the two-phase evolutionary pattern, we inactivated Rictor/mTORC2 to investigate the role of activation of PI3K/Akt/mTORC signaling in distant dispersal of glioma from the SVZ[22]. Rictor is an essential component of mTORC2, which phosphorylates serine 473 of AKT and is essential for full activation of AKT/Akt in both human and mouse tumor cells[33]. Similar to the previous studies showing that *Rictor* is dispensable for normal cells in the prostate and blood[34,35], conditional *Rictor* deletion (*Rictor*[Δ/Δ]) in the brain did not impact the lifespan of the conditional *Rictor*[Δ/Δ]CKO mice[22]. Importantly, *Rictor* deletion prolonged nearly 50% of the lifespan of *p53*[ΔE5−6]*Rictor*[Δ/Δ]CKO mice with malignant gliomas and GBMs, compared to *p53*[ΔE5−6]CKO mice. Importantly, *Rictor* deletion almost completely eliminated both phosphorylation of Akt[Ser473] and Akt[Thr308], and thus inhibited Akt activity in malignant gliomas from *p53*[ΔE5−6]*Rictor*[Δ/Δ]CKO mice[22]. To evaluate whether *Rictor* deletion has an impact on distant dispersal from the SVZ niche at early stages, we analyzed 14 brains of *p53*[ΔE5−6]CKO mice and 9 *p53*[ΔE5−6]*Rictor*[Δ/Δ]CKO mice at the age of 5–7 months. Using the expression of mutant p53[ΔE5−6] protein as a marker, we identified early glioma-like clusters comprised of BrdU⁺Olig2⁺ cells in 8/14 (57.1%) brains from *p53*[ΔE5−6]CKO mice compared to only 1/9 (11.1%) from *p53*[ΔE5−6]*Rictor*[Δ/Δ]CKO mice (Fig. 8a–e). Importantly, five out of the eight *p53*[ΔE5−6]CKO brains contained early glioma-like clusters, which were distributed at multiple sites distant from the SVZ/LV niche (Fig. 8a, c, e). In contrast, few BrdU⁺ cells with p53[ΔE5−6] expression were identified in the SVZ areas from eight out of nine *p53*[ΔE5−6]*Rictor*[Δ/Δ] CKO brains (Fig. 8b, d, e). Consistently, *Rictor* deletion completely eliminated multifocal tumors in *p53*[ΔE5−6]*Rictor*[Δ/Δ]CKO mice, compared to all three *p53*CKO models that exhibited a similar incidence of brains with multifocal tumors (Fig. 8f–h and Supplementary Fig. 8d). Moreover, almost all malignant gliomas and GBMs were found directly associated with the SVZ in the *Rictor*-deficient background (Fig. 8i). Together, these results demonstrate that PI3K/Akt pathway activation is required for distant dispersal of malignant gliomas/GBMs from the SVZ to brain parenchyma.

## Discussion

Using serial MRI and 3D-reconstruction analysis, we show that our GEM models recapitulate the two critical features of primary/de novo GBMs or *IDH*-WT GBMs—adult onset and rapid progression[1–3]. We integrated SKY-based single-cell analysis with WGS-based analysis of bulk tumor cells and single-cell-derived bulk tumor cells to investigate cancer evolution in the p53[R172H]-mutant driven GBM model. This model is the first demonstration of clonally related tumor cells in the SVZ undergoing distant dispersal (and less frequently, local expansion) via polyclonal and/or parallel seeding into distant brain regions by sequentially activating cancer drivers in Ras/Erk/MAPK and PI3K/Akt/mTORC signaling pathways (Fig. 8j)[14,36–38].

Two distinct patterns of tumor growth are observed in our GEM models, a single-mass Type 1 pattern, which arise de novo in the brain parenchyma, and a multifocal-mass Type 2 pattern, which in some cases retain a population of "early" tumor precursor cells in the SVZ stem-cell niche. Single-mass Type 1 tumors only show one phase of tumor evolution, exhibiting sub-4N genomes and loss of *Pten*/chr19. Together with the cNRT[4N] acquisition pattern, these observations suggest that the CA cell of Type 1 tumors become malignant after WGD and *Pten*/chr19 loss. Thus, Type 1 tumors resemble the growth pattern of the majority of *IDH*-WT GBMs, developing as a single mass and likely possessing only the rapid phase of tumor evolution[8–10]. In

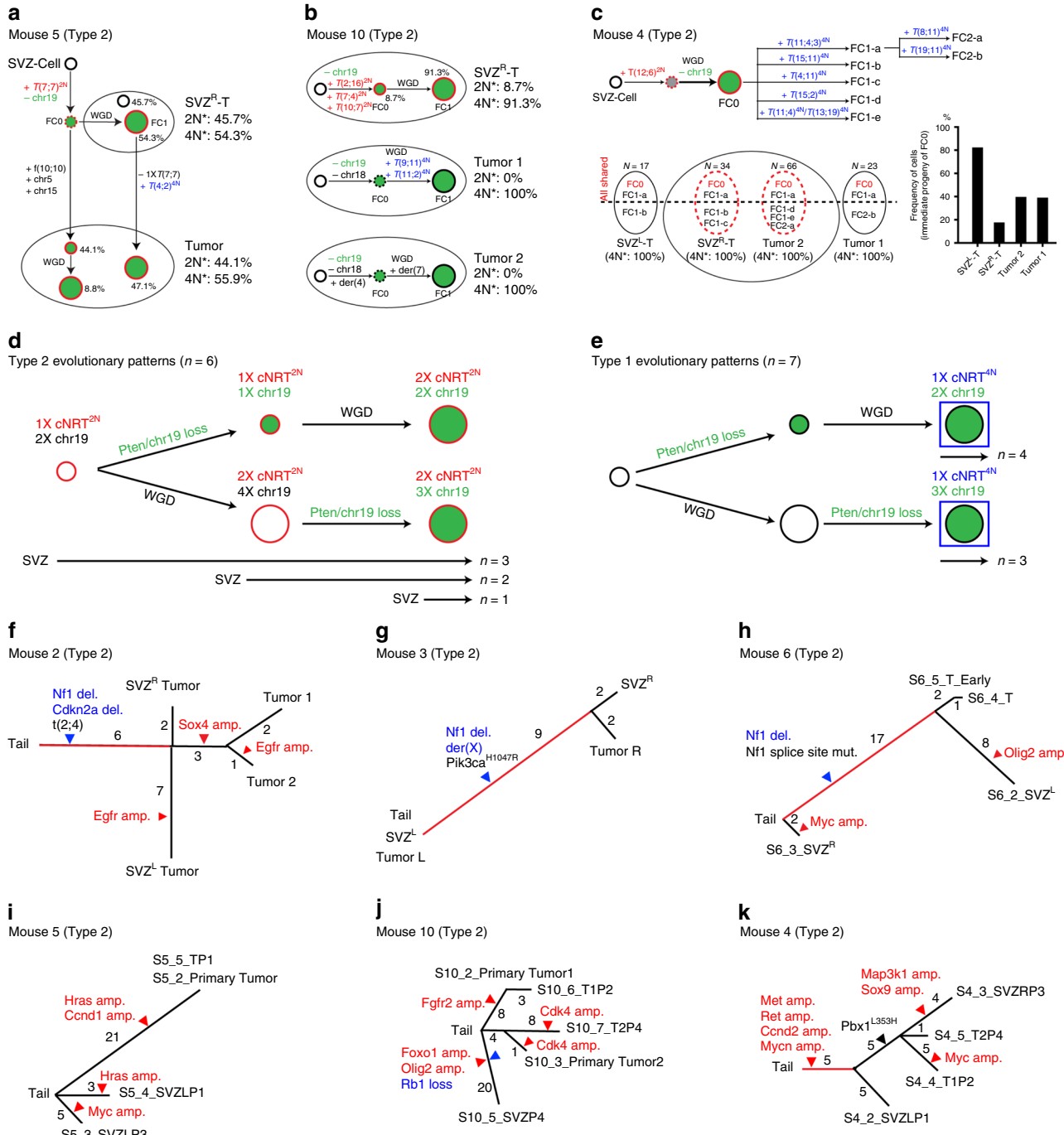

**Fig. 6 Distant versus local dispersal from the SVZ to brain parenchyma in additional Type 2 cases.** Models show the clonal evolution from the founder cell/clone (FC) to GBMs in three additional Type 2 cases, Mouse 5 (**a**), Mouse 10 (**b**), and Mouse 4 (**c**). Models were constructed using data from SKY-based single-cell analysis. The relative frequency of 2N and 4N cells in each site was shown. Note that the distant dispersal from the SVZ to parenchyma were observed in Mouse 5 (**a**) and Mouse 4 (**c**) based on the genetic complexity of FCs. In Mouse 10 (**b**), local evolution of clonally related near-2N and sub-4N cells carrying three different cNRT$^{2N}$s (labeled with red color and superscript "2N") in the SVZ tumor was illustrated. In Mouse 4, total metaphases (cells) of SVZ$^{L}$-T ($n = 17$), SVZ$^{R}$-T ($n = 36$), Tumor 1 ($n = 23$), and Tumor 2 ($n = 64$) were analyzed and the percentage of cells derived from FC0 in each sample was compared (**c**). Summary of tumor evolutionary patterns in all six Type 2 (**d**) and seven Type 1 cases (**e**) was analyzed. The number of cases that match each of these predicted growth patterns are noted below. The trees based on shared and private events in cancer drivers from WGS data were manually reconstructed for all Type 2 cases, Mice 2 (**f**), 3 (**g**), 6 (**h**), 5 (**i**), 10 (**j**), and 4 (**k**). The shared and private events between SVZ- and parenchymal tumor-derived cells were placed on the trunks and branches of each tree, respectively. Shared events determined the phylogenetic order of samples. Length is proportional to the number of events (number indicated). Arrowheads indicate genetic events (red, amplification; blue, deletion; and black, mutation). The source data underlying (**f**–**k**) are provided as a Source data file.

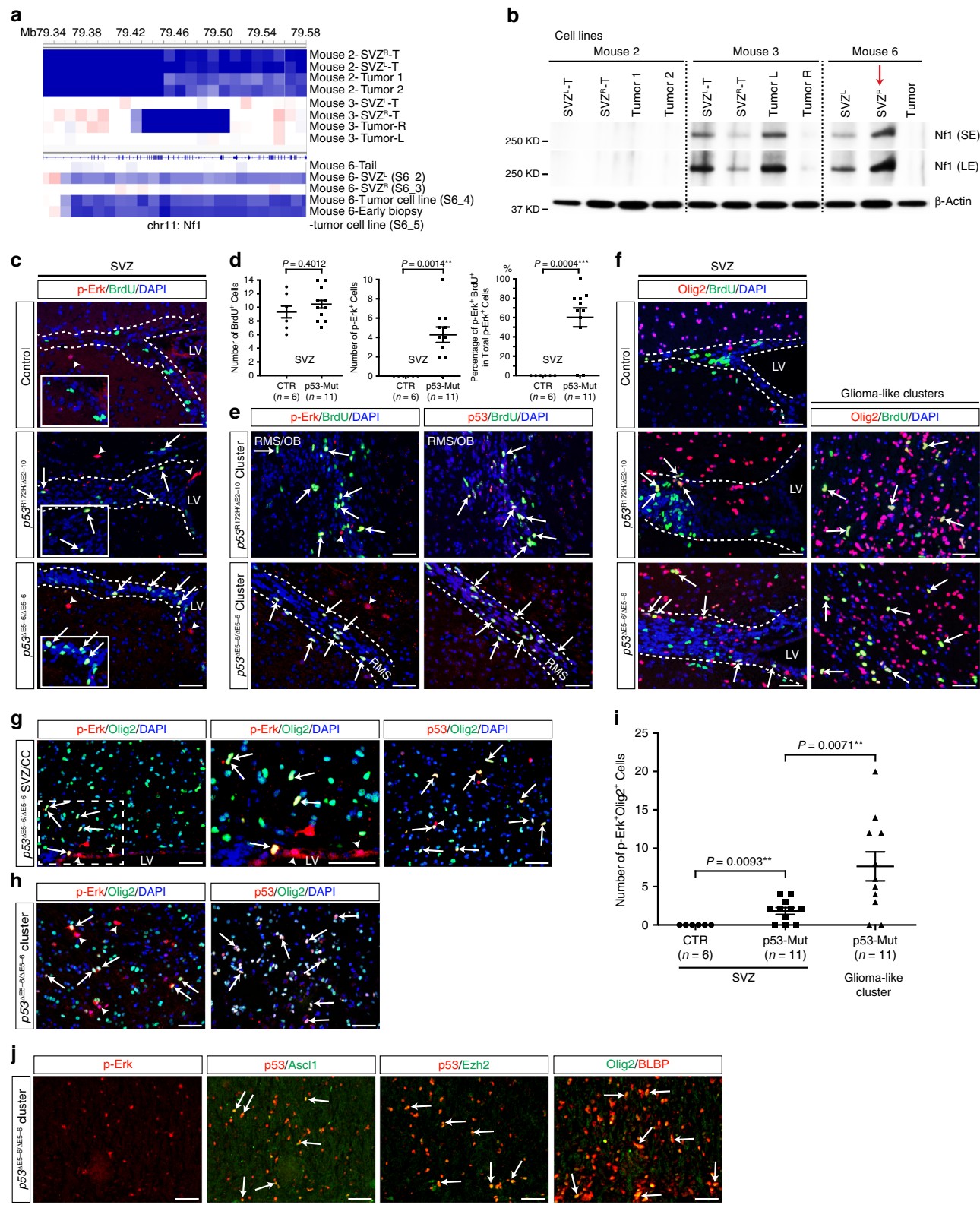

contrast, the Type 2 pattern was characterized by early acquisition of cNRTs in a CA cell with a near-2N genome or the $cNRT^{2N}$ acquisition pattern. The "early" versus "late" phases of tumor evolution in Type 2 tumors were distinguished by four critical features: (1) the ploidy of tumor cells, $cNRT^{2N-1\times}$-bearing near-2N versus $cNRT^{2N-2\times}$-bearing sub-4N cells; (2) the status of *Pten*/

chr19-normal versus *Pten*/chr19 loss (or *Pik3ca*[H1047R] mutation), and associated low versus high levels of PI3K/Akt/mTORC signaling; (3) the relatively quiescent versus high proliferative state of glioma cells; and (4) the localization of tumors to the SVZ versus spatially segregated distant brain parenchyma. Together, these findings suggest that the lack of a less proliferative "early"

**Fig. 7 Olig2⁺ progenitors with Erk/MAPK activation underlie clonal expansion in the SVZ. a** Copy-number variations (CNVs) of *Nf1* gene were shown in the three Type 2 cases. Blue represents copy loss/deletion, while red represents copy gain/amplification. **b** Western blotting of cell lines from the samples described above showed Nf1 protein levels. Red arrow indicates the histologically normal Mouse 6 SVZ$^R$: a control with normal levels of Nf1 expression. Source data are provided as a Source data file. **c–i** Control (n = 6), p53$^{R172H}$CKO (n = 3) and p53$^{ΔE5–6}$CKO (n = 14) mice at age 5–7 months were injected with BrdU to detect proliferating cells in tumor-free brains. White dashed lines in (**c**, **e**, **f**) mark the areas of SVZ and/or RMS in the stem-cell niche. **c**, **d** Co-labeling of p-Erk and BrdU in the SVZ. Arrows indicate p-Erk⁺BrdU⁺ cells (yellow), only observed in the mutant. Arrowheads indicate p-Erk⁺BrdU⁻ cells (green), observed in the control. Insets show high magnification in the SVZ. Quantifications are shown in (**d**). **e** Co-labeling of: p-Erk and BrdU; BrdU and p53 in early glioma-like clusters. Arrows indicate p-Erk⁺BrdU⁺, p53$^{R172H}$BrdU⁺, or p53$^{ΔE5–6}$BrdU⁺ cells. Arrowheads indicate p-Erk⁺BrdU⁻ cells. **f** Co-labeling of Olig2 and BrdU in the SVZ (left) and early glioma-like clusters (right). Arrows indicate Olig2⁺BrdU⁺ cells. **g–i** Co-labeling of p-Erk/Olig2 low- (left) and high- (middle, boxed areas in left) magnification, and p53/Olig2 (right) in the SVZ (**g**); early glioma-like cluster (**h**). Arrows indicate p-Erk⁺Olig2⁺ or p53$^{ΔE5–6}$Olig2⁺ cells. Arrowheads indicate p-Erk⁺Olig2⁻ cells. The quantification of p-Erk⁺Olig2⁺ cells is shown in (**i**). **j** Representative early glioma-like clusters in the olfactory bulb from *p53$^{ΔE5–6}$CKO* brains (n = 7) were labeled with: p-Erk; p53/Ascl1; p53/Ezh2; and Olig2/BLBP. Arrows indicate p53$^{ΔE5–6}$Ascl1⁺ or p53$^{ΔE5–6}$Ezh2⁺ or Olig2⁺BLBP⁺ cells. Data are presented as a dot plot with mean ± SEM. Unpaired, two-tailed Student's *t* test was used for statistical analysis in (**d**, **i**). **p < 0.01, ***p < 0.001. SVZ subventricular zone, LV lateral ventricle, OB olfactory bulb, RMS rostral migratory stream, CC corpus callosum, SE short exposure, LE long exposure. DAPI is a nuclear counterstain. Scale bars: 50 μm.

phase in the Type 1 pattern and the spatial segregation of the "early" and "late" phase of tumor evolution in the Type 2 pattern, contribute to the challenge associated with detecting early lesions in single-mass and multifocal-mass human primary GBMs, respectively[8–15].

At the single-cell level, the p53$^{R172H}$-mutant driven GEM model recapitulate the unique two-phase evolutionary pattern with spatial segregation of tumor initiation and manifestation recently observed in human *IDH*-WT GBMs[14]. Importantly, low versus high levels of genomic instability observed in the SVZ-versus autologous parenchyma-derived tumor cells in Type 2 cases could confer different sensitivity to drug treatment as previously described in human GBMs[11]. In a proof-of-principle experiment, we demonstrated that *Rictor*/mTORC2 deletion and associated PI3K/Akt inhibition eliminate distant dispersal of malignant gliomas/GBMs, leading to less proliferative SVZ niche-associated tumors[22]. Although treatment with these inhibitors has not been shown to be clinically effective for GBMs at the end stage[29], our study suggests a potential therapeutic strategy using PI3K/AKT/mTORC pathway inhibitors to prevent distant recurrence after standard treatment. Furthermore, this approach could prevent distant dispersal of *IDH*-WT primary GBMs if early detection of circulating primary GBM cells with chr10/*PTEN* loss becomes feasible.

## Methods

**Animal studies**. The floxed *p53* alleles (*p53$^{floxE2–10}$*) and (*p53$^{LSLR172H}$*) have been described previously[26,39]. Most of the *p53$^{null}$*CKO (*p53$^{ΔE2–10}$CKO, hGFAP-cre+; p53$^{floxE2–10/floxE2–10}$*) and *p53$^{R172H}$*CKO (hGFAP-cre+; *p53$^{LSLR172H/floxE2–10}$*) mice analyzed in this study were littermates from the breeding cross of hGFAP-cre+; *p53$^{LSLR172H/floxE2–10}$* (♂) × *p53$^{floxE2–10/floxE2–10}$* (♀). All mice were maintained in the mixed backgrounds of C57Bl6 and 129S1/Svj. *p53$^{ΔE5–6}$*CKO (hGFAP-cre+; *p53$^{floxE5–6/floxE5–6}$*); *p53$^{ΔE5–6}$Rictor$^{Δ/+}$*CKO (hGFAP-cre+; *p53$^{floxE5–6/floxE5–6}$*; *Rictor$^{flox/+}$*) and *p53$^{ΔE5–6}$Rictor$^{Δ/Δ}$*CKO (hGFAP-cre+; *p53$^{floxE5–6/floxE5–6}$*;*Rictor$^{flox/flox}$*) mice have been described previously[21,22]. All mice in this study were cared for according to the guidelines approved by the Animal Care and Use Committees of the University of Michigan at Ann Arbor and Children's National Hospital at Washington DC.

**MRI scans, image-guided biopsy, and image analysis**. MRI scans were performed on a 9.4 T, 16 cm horizontal bore (Agilent Technologies, Inc, Santa Clara, CA) Direct Drive system with a mouse head quadrature volume coil (m2m Imaging, Corp., Cleveland, OH) or mouse surface receive coil actively decoupled to a whole-body volume transmit coil (Rapid MR International, LLC., Columbus, OH). Throughout MRI experiments, animals were anesthetized with 1–2% of isoflurane/air mixture, and body temperature was maintained using a heated air system (Air-Therm Heater, World Precision Instruments, Sarasota, FL). MR images were acquired bi-monthly prior to tumor formation, weekly until biopsy and weekly thereafter until the animals were sacrificed or became moribund.

Delineation of tumor growth from healthy brain tissues was determined using T2-weighted fast-spin-echo images with the following parameters: repetition time

(TR)/echo time = 3010/12 ms, field of view = 20 × 20 mm², matrix size = 256 × 128, slice thickness = 0.5 mm, 25 slices, and 2 averages. Total acquisition time was 1 min and 48 s.

Biopsies were performed using pre-biopsy MRI to target tumor location, as well as post-biopsy MRI to verify accuracy of tissue collection. A fiducial marker filled with contrast agent was fixed to the bed adjacent to the head. A pre-biopsy T2-weighted image was acquired using a surface receive/volume transmit coil (Rapid MR International, LLC., Columbus, OH). X and Z coordinates relative to the fiducial marker, as well as the depth from the skull (Y coordinate) of the tumor, were determined using VnmrJ software (Agilent Technologies, Inc, Santa Clara, CA, version 3.2). The Rapid MRI bed was modified to enable attachment to the base of a bench-top stereotaxic device (David Kopf Instruments, Tujunga, CA). Mice were anesthetized just prior to biopsy with an IP injection of ketamine/xylazine. The head was shaved and eye lubricant was applied. Seventy percent of EtOH was used to sterilize the head prior to cutting. A high-speed surgical drill (The Foredom Electric Co., Bethel, CT) was mounted on the stereotaxic device and moved to the position of the fiducial marker. The drill was moved to the intended biopsy location, using the coordinates determined earlier, and a hole was drilled to allow needle insertion. The drill was removed and replaced with a 22-gauge, 3 7/8 in Samplemaster Chiba biopsy needle (Inrad, Kentwood, MI). The needle assembly was moved to the location of the hole and inserted to the top of the bevel. The needle was inserted to the previously determined depth and the needle insert was removed. A 10 ml syringe with valve attachment was fastened to the needle assembly with a section of infusion line. With the syringe plunger at 2.5 ml the valve was closed and the syringe pulled to 5 ml and locked to create a vacuum. The valve was opened for 3 s to allow the biopsy sample to be pulled up into the needle. The infusion line was removed from the syringe valve and pinched off. The needle was slowly retracted and a 10 ml syringe full of air was used to inject the sample into an Eppendorf tube for subsequent tissue culture studies or frozen for further genetic analysis.

Volumes of interest were manually contoured around the hyper-intense portion of the tumors on the T2-weighted images for tumor volume measurements using in-house software (The Center for Molecular Imaging, University of Michigan, Ann Arbor) running in MATLAB (MathWorks, Natick, MA, version 2016b). In brief, tumor volumes were determined by multiplying the number of tumor voxels by the voxel volume. In each time point, the tumor size was calculated and the tumor location was recorded. 3D representations of the segmentation volume (whole brain, ventricles, or tumor) were generated using Matlab scripts, including generation of an isosurface from the binary mask, followed by curvature flow smoothing of the surface mesh[40].

**PCRs and targeted Sanger sequencing**. For Genotyping PCR of the *p53$^{floxE2–10}$* and *p53$^{LSLR172H}$* alleles: Taq 2X MeanGreen Master Mix (Empirical Bioscience) was used in PCR experiments along with the primer sets for tail and tumor tissue genotyping.

The following primers were used for the genotyping PCR for *p53$^{floxE2–10}$* allele:
*p53*_1F: 5′-CACAAAAAACAGGTTAAACCCAG-3′
*p53*_1R: 5′-AGCACATAGGAGGCAGAGAC-3′
*p53*_10R: 5′-GAAGACAGAAAAGGGGAGGG-3′
The following primers were used for the genotyping PCR for *p53$^{LSLR172H}$* allele:
TO34: 5′-AGCCTGCCTAGCTTCCTCAGG-3′
TO35: 5′-CTTGGAGACATAGCCACACTG-3′
TO36: 5′-AGCTAGCCACCATGGCTTGAGTAAGTCTGCA-3′
For targeted Sanger sequencing of *p53$^{R172H}$*, *Idh1*, *Idh2*, and *H3f3a* alleles in tumors. Total DNA and RNA were extracted from snap-frozen brain tumor samples by using Qiagen AllPrep DNA/RNA Mini Kit (80204). Then the first-strand cDNA was synthesized from RNA by using Invitrogen SuperScript® III

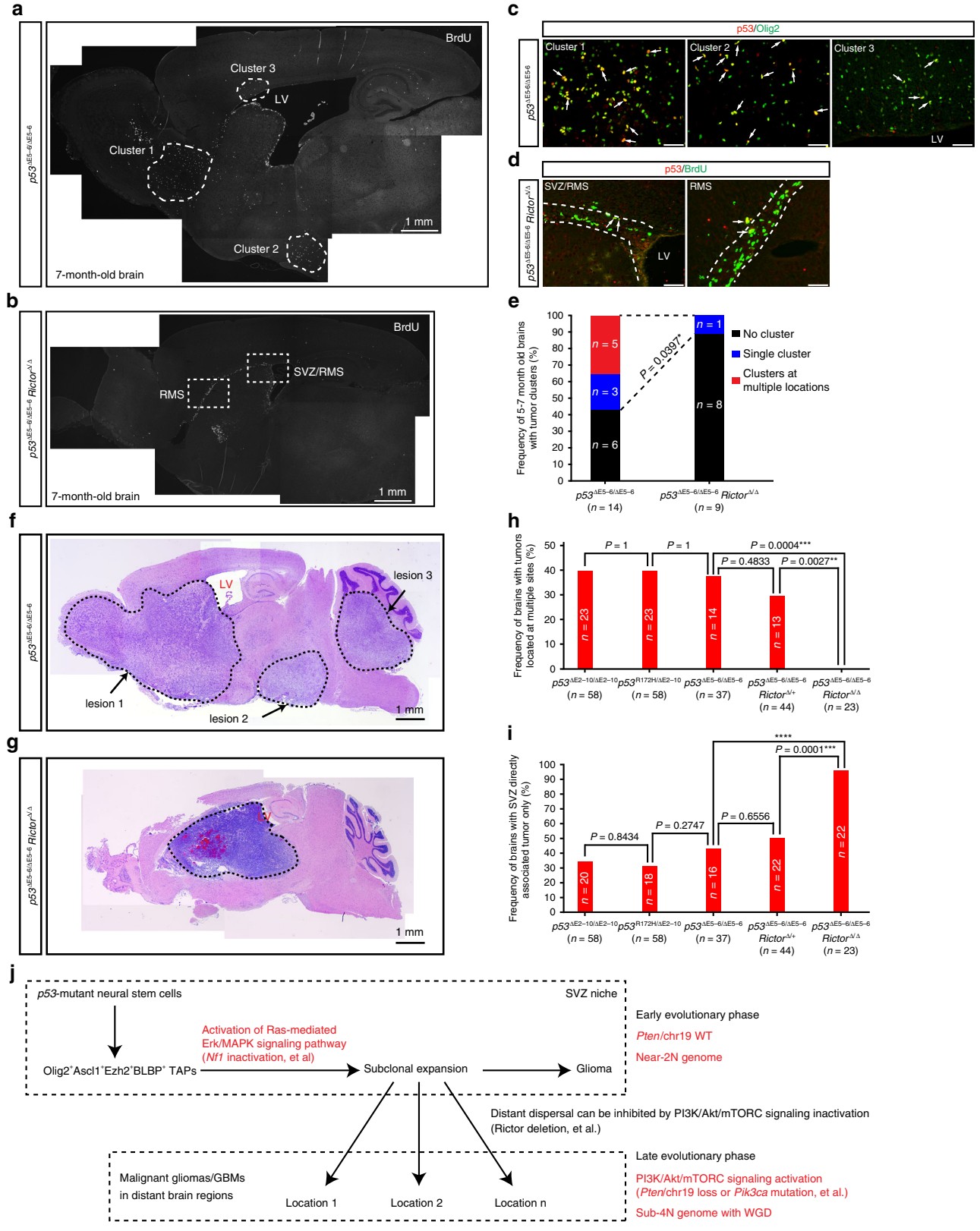

First-Strand Synthesis SuperMix (18080-400). Regular PCR was performed to amplify the coding regions of *p53* (NM_011640.3) by using primers *p53*_P1; *p53*_P2; *p53*_P3; *p53*_P4; *p53*_P7; and *p53*_P8. *Idh1* (NM_001111320.1) by *Idh1*_S and *Idh1*_AS. *Idh2* (NM_173011.2) by *Idh2*_S and *Idh2*_AS and *H3f3a* (NM_008210.4) by *H3f3a*_S and *H3f3a*_AS. Then the gel-recovered PCR fragments for *p53* (the fragment amplified by P1 and P2), *Idh1*, *Idh2*, and *H3f3a* were sequenced in the DNA sequencing core at University of Michigan by using

the same primers used in PCRs (For *p53* sequencing, additional two primers *p53*_P3 and *p53*_P7 were used). Primers used here are:

*p53*_P1: 5′-AGGTAGCGACTACAGTTAGGGG-3′
*p53*_P2: 5′-TGAAGTCATAAGACAGCAAGGAG-3′
*p53*_P3: 5′-CCTGCCATCACCTCAC-3′
*p53*_P4: 5′-GGAAGCCATAGTTGCC-3′
*p53*_P7: 5′-GCCGGCTCTGAGTATA-3′

**Fig. 8 Rictor/mTORC2 loss inhibits distant dispersal. a–e** $p53^{\Delta E5-6}$CKO ($n=14$) and $p53^{\Delta E5-6}Rictor^{\Delta/\Delta}$CKO ($n=9$) mice at age of 5–7 months were injected with BrdU to detect proliferating cells in tumor-free brains. Adjacent brain sections were stained for BrdU (**a, b**), co-labeled for p53 and Olig2 (**c**), or co-labeled for BrdU and p53 (**d**). **a, c** A representative image with BrdU+ proliferating clusters identified in multiple regions from $p53^{\Delta E5-6}$CKO mice (5 out of 14) was shown in (**a**). All the three clusters containing early glioma-like cells comprised of Olig2+p53$^{\Delta E5-6}$ cells (indicated by arrows) (**c**). **b, d** A representative image with BrdU+ proliferating cells located in SVZ-associated areas from $p53^{\Delta E5-6}Rictor^{\Delta/\Delta}$ CKO mice (eight out of nine) was shown in (**b**). BrdU+ cells with p53$^{\Delta E5-6}$ expression (indicated with arrows) were identified in the SVZ-associated areas in $p53^{\Delta E5-6}Rictor^{\Delta/\Delta}$ CKO brain (**d**). **e** The number of brains with early tumor cluster(s) characterized by BrdU+Olig2+ cells was compared in 5–7-month-old $p53^{\Delta E5-6}$CKO and $p53^{\Delta E5-6}Rictor^{\Delta/\Delta}$CKO models. Fisher's exact test was used to analyze the statistical difference. *$p < 0.05$. H&E staining shows representative brains with multifocal lesions and single lesion from the $p53^{\Delta E5-6}$CKO (**f**) and $p53^{\Delta E5-6}Rictor^{\Delta/\Delta}$CKO (**g**) mice at end stages, respectively. Analysis of brain histology demonstrated the percentage of brains with multifocal tumors (**h**) and brains with tumors directly associated with the SVZ only (**i**). Analysis was compared among three $p53$CKO models, $p53^{\Delta E5-6}Rictor^{\Delta/+}$CKO and $p53^{\Delta E5-6}Rictor^{\Delta/\Delta}$CKO models. Fisher's exact test was used to analyze the statistical difference. **$p < 0.01$, ***$p < 0.001$, ****$p < 0.0001$. **j** Model of a single-cell-derived early tumor cluster that sequentially accumulates GBM-relevant driver alterations in the RAS/MAPK and PI3K/AK/mTORC pathways, driving local expansion and distant dispersal during GBM evolution. SVZ subventricular zone, LV lateral ventricle, RMS rostral migratory stream. Scale bars: 50 μm (**c, d**); 1 mm (**a, b, f, g**).

*$p53$*_P8: 5′-TGTGATGATGGTAAGGA-3′
*Idh1*_S: 5′-TTATTGAAGTAAAAATGTCC-3′
*Idh1*_AS: 5′-TCTAAGCCCAGGTTTGAC-3′
*Idh2*_S: 5′-GCCTAGAGTCCCCACCGC-3′
*Idh2*_AS: 5′-CCCACCCTCTGCCATGTA-3′
*H3f3a*_S: 5′-ATGGCTCTTACAAAGCAG-3′
*H3f3a*_AS: 5′-TTAAGCACGTTCTCCG-3′

For $t(2;4)$ detection, targeted Sanger sequencing and real-time PCR. Based on the breakpoints analysis from WGS data, primers of *B4galt5*_S1; *B4galt5*_S2; *B4galt5*_S3; *B4galt5*_AS; *Dab1*_AS1; *Dab1*_AS2; *Dab1*_AS3 were designed and used for amplify the fragment of WT allele of Chromosome 2 and $t(2;4)$ from the four tumor cell lines in Mouse 2. The reference genome sequences that used to design the primers are: Chromosome 2 (NC_000068.7) and Chromosome 4 (NC_000070.6). Briefly, primer sets *B4galt5*_S2/*B4galt5*_AS were used to amplify the WT allele of Chromosome 2 and primer sets *B4galt5*_S2/*Dab1*_AS3; *B4galt5*_S1/*Dab1*_AS1; and *B4galt5*_S1/*Dab1*_AS2 were used to amplify the $t(2;4)$ fragments by using Phusion® High-Fidelity DNA Polymerase (M0530, NEB). Then the PCR products were sent out for sequencing at GENEWIZ, Inc by using the same primers used in PCRs. The primers used here are:

*B4galt5*_S1: 5′-CGGCAGGTCAGTGATAAGAAAC-3′
*B4galt5*_S2: 5′-GAAATTCCTCTAGTCAGAGGGTC-3′
*B4galt5*_S3: 5′-AGGACTTCAAGGAAAGGTAGCTCAG-3′
*B4galt5*_AS: 5′-TAGGACAGGCTAGTCCCAGAAC-3′
*Dab1*_AS1: 5′-CTCCTTGGAACTGCTCTGACT-3′
*Dab1*_AS2: 5′-TCCAGAGCAGTCACCAATAGAC-3′
*Dab1*_AS3: 5′-ATCCTTCAAATGCCCCTTGTCC-3′

For real-time PCR, depending on the sample concentration, various amounts of template genomic DNAs (average 100 ng/reaction) from early passage tumor cell lines of SVZ$^L$, SVZ$^R$ (from Mouse 2) as well as a tumor without $t(2;4)$ as control along with the SYBR Select Master Mix (Applied Biosystems) and the following primers sets were used in real-time PCR. See primers information in Supplementary Table 2.

$t(2;4)$ band: *B4galt5*_S1/*Dab1*_AS1 or *B4galt5*_S2/*Dab1*_AS3
WT Chromosome 2 band (wt): *B4galt5*_S3/*B4galt5*_AS
GAPDH band: *GAPDH*_Forward/*GAPDH*_Reverse
*GAPDH*_Forward: 5′-ACCCAGAAGACTGTGGATGG-3′
*GAPDH*_Reverse: 5′-CACATTGGGGGTAGGAACAC-3′

CT value (CT$_{testGene}$ − CT$_{reference}$) and then [1/CT$^2$] is calculated for each sample using GAPDH as the reference. The ratio of $t(2;4)$ band with WT Chromosome 2 band in SVZ$^L$ tumor was signed with "1" based on the homogeneous near-2N population observed in SKY analysis which has one copy of $t(2;4)$ and one copy of WT Chromosome 2, then the ratio of $t(2;4)$/wt in other samples was calculated.

**BrdU assay for the analysis of proliferation in adult brains**. To label the proliferating cells in adult brains, mice were injected with 50 μg of BrdU per gram of body weight. In all, 5–7-month-old mice were received five injections, with multiple injections applied at 2-h intervals. All of the mice were perfused with 4% paraformaldehyde (PFA) 2 h after the final BrdU injection. Then the samples were processed as described below for future analysis.

**Tissue preparation, histopathology, and tumor diagnosis**. Mice were aged until signs of distress appeared. Then, mice were perfused with 4% PFA and brains were dissected, followed by overnight post-fixation in 4% PFA at 4 °C. Brains were divided into two hemispheres along the midline and each hemisphere was then processed for either paraffin-embedded or cryostat sections. At some conditions, brains were also coronally cut and then processed. Serial sections from the side of brain which has the major tumor mass were sagittally or coronally prepared at

5 μm for paraffin sections. Typically, we began to collect the brain sections when the complete brain structure from middle line was shown as Section 1. For each slide, we collected four continuous brain sections. Total 31 slides were collected for each half brain. Then slides #1, #11, #21, and #31 were stained with H&E for tumor analysis. Stained sections were independently examined under a light microscope by Y.L. and C.Y.H. Tumor grading was first diagnosed by Y.L. by using the same criteria as previously described[21,22]. Then all the tumors were further diagnosed and confirmed by C.Y.H. based upon the WHO grading system for malignant astrocytic glioma, GBM, and medulloblastoma[2]. Adjacent sections were subjected to immunohistochemical analysis (see below).

**Immunohistochemistry and immunofluorescence**. Paraffin sections were deparaffinized and rehydrated. Adjacent sections were subjected to immunohistochemical and immunofluorescence analysis as previously described[21,22]. The visualization of primary antibodies was performed with a horseradish peroxidase (HRP) system using a diaminobenzidine-based (DAB) peroxidase substrate (Vectastain ABC kit, Vector). The visualization of the primary antibodies in immunofluorescence was performed with the use of the Alexa488, Alexa555, and Alexa647 conjugated secondary antibodies (A11004, A11034 for Alexa488; A21429, A21424 for Alexa555; A31571, A21236, A21247, A21245 for Alexa647; 1:400, Invitrogen, Life Technologies). DAPI was used as a counterstain to label individual cell nuclei. The primary antibodies used in this study were: rabbit anti-p53 (NCL-p53-CM5p, 1:1000, Leica Biosystems), mouse anti-Ki67 (550609, 1:500, BD Pharmingen), rabbit anti-Olig2 (AB9610, 1:2000, EMD Millipore), Guinea pig anti-Olig2 (1:10000, a kind gift of home-made antibody from Dr. B. Novitch), mouse anti-GFAP (556330, 1:2000, BD Pharmingen), rabbit anti-Pten (9559S, 1:1000, Cell Signaling), rabbit anti-p-S6 (5364S, 1:2000, Cell Signaling), rat anti-BrdU (ab6326, 1:500, Abcam), rabbit anti-p-Erk (9101S, 1:2000, Cell Signaling), mouse anti-Ascl1 (556604, 1:100-1:200, BD PharMingen), rabbit anti-Ezh2 (5246S, 1:1000–1:2000, Cell Signaling), and rabbit anti-BLBP (ab32423, 1:200, Abcam). Sections were examined under a light and fluorescent microscope (Olympus BX-63). Pten expression and pS6 expression were quantified based on the qualitative estimate of the ratio of Pten+ or p-S6+ cells.

High: most of the cells are Pten+ or p-S6+, the ratio of Pten+ or p-S6+ cells is estimated >80%;

Medium: mixed areas of Pten+ and Pten− cells or p-S6+ and p-S6− cells;

Low: most of the cells are Pten− or p-S6−, the ratio of Pten+ or p-S6+ cells is estimated <10%.

**Determination of Type 1 and Type 2 brains**. For MRI study, the location of each tumor was recorded and the number of tumor lesions was quantified for each of the detectable lesions based on MRI. After collection at the end stage, histological analysis was performed on the brains and the tumor location was further recorded and compared to MRI data.

For histological analysis, we excluded rare samples that have medulloblastoma in the hindbrain and focused on brains with glioma only. We analyzed these brain samples on four sagittal planes each ~200 μm apart, and documented the locations/size of each tumor foci on H&E stained slides. If we saw multiple tumor foci with clear margins throughout the slides, we then considered that brain as multifocal lesions. Brains with only one tumor mass throughout the slides analyzed were considered as single lesion. In some cases, tumors appeared multifocal lesions on one slide but were merged on other slide(s). These brains were considered as single lesion as well.

**Western blotting analysis**. Snap-frozen tissues from normal brains and tumors were homogenized in 1X SDS loading buffer [50 mM Tris-HCL (pH6.8), 2% SDS, 0.05% bromophenol blue, 10% glycerol, 100 mM β-mercaptoethanol]. Samples were analyzed by SDS-PAGE and transferred onto PVDF membranes (EMD

Millipore). The blots were then blocked in 5% nonfat milk in TBST, followed by incubation of primary antibodies at 4 °C overnight. After washing, the blots were incubated in HRP-conjugated secondary antibodies (1706515; 1706516, BioRad) at room temperature for 1 h. Signals were detected using ECL or ECL plus (GE healthcare) followed by film development. The primary antibodies used are as follows: rabbit anti-p53 (NCL-p53-CM5p, 1:1000, Leica Biosystems), rabbit anti-p-Akt(T308) (2965S, 1:1000, Cell Signaling), rabbit anti-p-Akt(S473) (4060L, 1:1000, Cell Signaling), rabbit anti-Akt (9272S, 1:2000, Cell Signaling), rabbit anti-Pten (9559S, 1:1000, Cell Signaling), rabbit anti-p-S6 (5364S, 1:2000, Cell Signaling), rabbit anti-S6 (2217S, 1:2000, Cell Signaling), mouse anti-p120 (Anti-Ras-GAP, 610040, 1:1000, BD Biosciences), rabbit anti-Nf1 (SC-67, 1:1000, Santa Cruz Bio-technology), and mouse anti-β-Actin (A5316, 1:20000, Sigma-Aldrich).

**Establishment of primary tumor and SVZ cell lines.** Tumor(s) and/or SVZ(s) were carefully dissected from the brains and placed into fresh Opti-MEM medium (Gibco by Life Technologies). Then the cells were centrifuged and dissociated with pre-warmed TrypLE select (1×) solution (Gibco by Life Technologies) for 20 min at 37 °C (a firm shake by hand every few minutes to break up clumps). NSC culture medium was a mixture of DMEM-low glucose: Neurobasal medium (Gibco by Life Technologies, 60.2 and 35.8 mL per 100 mL separately) supplemented with 20 ng/mL human bFGF (Sigma), 20 ng/mL EGF (Sigma), 1% N2 supplement (Gibco by Life Technologies), 2% B27 supplement (Gibco by Life Technologies), 50 μM 2-Mercaptoethanol (Sigma), and 1% penicillin Streptomycin (Gibco by Life Technologies). After incubation, the cells were centrifuged and then gently resuspended into NSC medium and filtered through a 70 μm nylon cell strainer. The cells were centrifuged again and resuspended with NSC medium. After counting, the cells were transferred into ultra-low binding six-well plates (Corning) with the NSC medium which has been prepared and equilibrated in a humidified incubator (5% CO2, balance air) for at least 1 h. In all, 2000–5000 cells were seeded per well and allowed for spheres formation. Fresh medium was added every 3 days during the sphere culture. Cell lines were passaged every 7–10 days.

**Cell dispersal, fixation, staining, and flow cytometry.** The procedure to detect the ploidy information from snap-frozen tissues by flow cytometry was based on a published protocol[41] with some modifications. Briefly, tissue samples, previously snap-frozen in −80 °C, were thawed in cold PBS. Then the samples were mechanically dispersed using pre-cleaned microscope slides (Fisher Scientific #12-550-343) and passed through the 70 μm nylon cell strainers (BD Falcon #352350). After cen-trifugation, the cell pellet was washed with cold PBS for two times and then suspended in 0.5 ml cold PBS. 5 ml of 70% ethanol stored at −20 °C was added dropwise under constant gentle vortexing. Samples were incubated for 30 min on ice and subsequently overnight at −20 °C before being subject to staining. Samples were centrifuged and washed with cold PBS two times before the cells were suspended in an appropriate volume (0.3–1 ml) of staining solution (PBS which contains 30 μg/ml propidium iodide and 0.3 mg/ml DNase-free RNase A) for 30 min at 37 °C. Then the samples were acquired using BD FACS Canto II and further analyzed by FlowJo software. Normal cerebral cortex was used as control for normal diploid DNA content.

**Metaphase spread, spectral karyotype (SKY) analysis.** To prepare metaphase spread, early passage cell lines from primary tumors or SVZ areas were cultured in NSC medium and incubated in 0.1 mg/ml KaryoMAX colcemid solution (Gibco by Life Technologies) for 2–3 h. Cells were collected, washed with PBS and then dissociated to single-cell suspension. The cells were centrifuged and resuspended in 75 mM potassium chloride solution (Gibco by Life Technologies, pre-warmed to 37 °C) and incubated at 37 °C for 18–20 min. Then the cells were fixed with ice-cold fixative solution (3:1 methanol (Fisher Chemical, A454-4): glacial acetic acid (Sigma-Aldrich, A6283-500ml)) and dropped onto glass slides. The slides were either stained with DAPI for further chromosome number counting or subjected to SKY analysis. For chromosome number counting, the metaphase images were captured using the fluorescence microscope (Olympus BX-63) and the chromo-some number was counted using ImageJ software. SKY analysis was performed by using a mouse SKY probe kit (Applied Spectral Imaging, Vista, CA) according to the manufacturer's protocol, followed by counterstaining with DAPI. Metaphase images were captured and analyzed by HiSKYV spectral imaging system from Applied Spectral Imaging. For each metaphase, the copy number of each chro-mosome and all the abnormal chromosomal structural events including translo-cations and fusions were recorded. Mouse Chromosome 19 fluorescence in situ hybridization (FISH) analysis was performed on dropped metaphase by using mouse Chr19 probe (XMP 19 orange, D-1419-050-OR, MetaSystems) according to the manufacture's protocol, followed by counterstaining with DAPI. Metaphase images were captured and analyzed by HiSKYV spectral imaging system from Applied Spectral Imaging. Chr19 copy number in each metaphase from Fish assay was counted for further analysis.

**Determination of whole-genome duplication (WGD).** Although focal amplifica-tions could in principle increase tumor ploidy to 2.5, computational analysis suggests that this is rare. Data from pan-cancer analysis showed tumors with versus without WGD were clearly separated at Ploidy 2.5 (Human, $n = 55$ chromosomes)[27]. Furthermore, we investigated the distribution of chromosome numbers in human

high-grade gliomas (including Grade III gliomas and Grade IV gliomas/GBMs) using cytogenetic data collected by the Mitelman database (Supplementary Fig. 2d). The data clearly showed the bimodal distribution for WGD and non-WGD tumors with chromosome counts around 55 (ploidy 2.5) as a cutoff. In our cytogenetic data, we confirmed that this bimodal distribution and the same ploidy 2.5 ($n = 50$) were indeed an excellent cutoff value to separate the two modes (Supplementary Fig. 2c). Thus, we used ploidy 2.5 (50 chromosomes in mouse cells) to separate WGD and non-WGD tumors from our GEM models.

**NRT sharing matrix.** The sample-wise NRT sharing matrix is calculated in the following manner: if two samples share NRT, the corresponding entry in the matrix is the positive number of shared NRTs; otherwise it is zero. In order to quantify the likelihood of inter-lineage NRT sharing, we calculate $p = $ P (metaphases A and B belong to different lineages A and B share NRT) using bootstrap. We randomly sample 100,000 pairs of metaphases from a total of 722 metaphases pooling 21 lineages analyzed by SKY. Let $n_r$ denote the number of pairs sharing NRT, $n_d$ denote the number of pares sharing NRT and belong to different animals (linea-ges), and $n_a$ denote the number of animals, then $p = n_d/n_r/(n_a - 1) = 0.000036$. The factor $1/(n_a - 1)$ is the normalization constant to ensure that selecting inter-lineage pair has the same probability of selecting intra-lineage pair.

**SKY-based single-cell phylogenetic tree building.** We used copy number of each individual chromosome, copy number of large-scale chromosomal alterations such as translocations/fusions and ploidy in each metaphase obtained from SKY analysis to build NJ trees for each sample, based on the pairwise distance between each two metaphases. We developed R codes and used RStudio (version 1.2.5033), R package "Analyses of Phylogenetics and Evolution" (APE, version 5.2), and MEGAX (version 0.1, for MAC) to generate SKY-based phylogenetic trees. Briefly, we used the following three steps to build SKY-based single-cell phylogenetic tree.

Step 1—feature selection: SKY method provides the chromosome profile for each individual cell: copy number of each individual chromosome, copy number of large-scale chromosomal alterations such as translocations/fusions and ploidy. All these chromosomal events will be used for the tree building.

Step 2—distance measure selection and distance matrix calculation: to build the tree, we first convert each of the chromosomal events into a distance measurement. We assigned every chromosomal event (e.g., gain or loss) at weight of "1" with two exceptions. First, since the phylogenetic tree is built on the lineage tracing of a FC harboring at least one cNRT^{2N}(s), we gave a higher weight of "5" for the de novo acquisition of an NRT from one cell without the NRT to become the NRT-bearing cell. The rationale for an NRT with the highest weight is based on the same breakpoint at the single-nucleotide level. This feature makes it highly unlikely to reproduce the exact NRT in different cell lineages, and thus is considered a unique/non-sharing identifier. Second, we considered WGD as a single genetic event and thus gave a weight of "1" instead of "40", due to duplication of all 40 chromosomes from one genetic process (see the "Methods" section for details). For example, a cell acquiring cNRT² (at 2N stage) and subsequently undergoing WGD with two copies of the cNRT^{2N} receives a weight of "6" (NRT + WGD = "5 + 1") instead of "10" (NRT + NRT = "5 + 5"). Then we developed the R code to calculate the distance between each pair of metaphases based on the above principle. See detailed information of distance calculation in the attached R code.

Step 3—NJ tree building: from Step 2, we generated the distance matrix of all the metaphases from each sample, then we developed the R code to reconstruct the NJ trees based on the distance matrix by using the R package "APE" (version 5.2) under the minimum evolution principle[42]. To identify the direction of evolution requires an outgroup, which is a normal progenitor cell without any chromosomal aberration. We add one normal euploid cell and assign it to be the ancestry (CONTROL). See detailed information of NJ tree building in the attached R code.

We built the NJ tree for tumor cells derived from each site of all Type 2 cases in which each dot in the tree represents a single proliferating cell/clone. The distance between any two cells on the tree is the consequence of minimal numbers of chromosomal events. The distance between every tumor cell and the normal 2N cell (germline) represents the number of somatic chromosomal events accumulated during tumor evolution. Thus, the cell carrying the fewest chromosomal event(s) is located at the distance closest to the normal 2N cell, representing the earliest evolutionary stage. Conversely, the cell carrying the most chromosomal events is most distant to the normal 2N cell, representing the latest stage of tumor evolution.

**Clonal evolution interpretation from SKY-based trees.** To illustrate the clonal evolution from different samples in each animal, we first illustrated the clonal evolution for each sample based on its single-cell phylogenetic tree built above. Briefly, we delineated tumor evolution for each sample by identifying a small number of clonally related cNRT^{2N}-bearing FCs, which gradually accumulated one or more chromosomal events. Once the genotypes of all FCs were identified for each tumor site, we used two criteria to determine the evolutionary direction of FCs carrying the same cNRT(s): (1) an irreversible transition by WGD followed by random loss of chromosomes (from a near-2N to sub-4N clone) and (2) the transition from simple to more complex genomes defined by accumulation of additional chromosomal events, including the universal loss of Pten/chr19 in parenchymal tumors in all Type 1 and Type 2 cases.

Lastly, we compared the genotypes of all the FCs from each sample in each animal, and illustrated their clonal relationship/evolution order based on the most parsimonious order of their occurrences.

**Gene expression analysis**. RNA extracted from 11 $p53^{null}$CKO glioma samples and 11 $p53^{R172H}$CKO primary tumors (of which each paired with a primary tumor-derived cell line), 2 early biopsies and 5 normal dorsal cortical tissues from control mice (3 from hGFAP-Cre+; $p53^{floxE2-10/+}$ mice and 2 from hGFAP-Cre−; $p53^{floxE5-6/floxE5-6}$ mice) were sent out for gene expression analysis in the DNA sequencing core at the University of Michigan. mRNA expression was profiled using Affymetrix GeneChip mouse genome 430 2.0 array, including 45K probes genome-wide. Dr. Johnson from University of Michigan DNA sequencing core, performed background correction, quantile normalization and gene expression robust multi-array average calculation for the 40 arrays using IVT Express kit and customized R codes.

To define the subtypes of the malignant gliomas and GBMs from out GEM models, we used the classification system established by Verhaak et al.[43], which defined four signatures with a total of 840 discriminative genes (210 genes for each signature), classifying GBMs into four subtypes, proneural, neural, classical, and mesenchymal. We used a Pearson correlation by comparing the expression pattern of these 840 feature genes between our GEM gliomas/glioma-derived cell lines and individual human GBMs with the known subtypes in the TCGA data sets. A positive correlation suggests that a given GEM tumor most resembles the GBM subtype based on its expression of the feature genes.

Alternatively, we further applied an enrichment-based method using the same feature genes. This method, called single sample gene set enrichment analysis, was used to predict GBM subtypes from >500 human GBMs in a recent TCGA study (Brennan et al.[1] cell). Primarily, this method serves to aggregate the expression of the feature genes into a score, which reflects the overall activity of the feature genes and thus the represents a specific subtype.

The human proneural subtype GBMs can be further divided into $IDH1/2$-WT non-G-CIMP GBMs and $IDH1/2$-mutant G-CIMP + GBMs. A recent study defined a set of best gene expression classifiers for these two groups in human GBMs[24]. We used the top 200 genes (89 found in mouse genes) to study the relationship between mouse and human GBM methylation subtypes. Similarity between samples was measured using Pearson correlation.

**Whole-genome sequencing (WGS) on bulk tumor samples**. Total 31 samples (including tails, paired tumor and SVZ cell lines, and/or primary tumors) from six Type 2 cases were performed whole-genome DNA sequencing analysis. Briefly, genomic DNA was extracted with Allprep DNA/RNA/miRNA Universal Kit (Qiagen) following the manufacturer's instructions. The library preparation and sequencing were done by the company. Among them, total of ten samples from two mice (Mice 2 and 3) were performed using Illumina HiSeq 2000 platform in the DNA sequencing core, University of Michigan. The sequencing coverage ranges from 8 to 13×, with an average of 10×. The remaining 21 samples from 4 mice (Mice 4, 5, 6 and 10) were performed using Illumina HiSeq X Ten platform from Novogene, with average of 30X sequencing coverage.

Raw sequencing reads are first filtered to remove contaminated and low-quality reads. Burrows–Wheeler Aligner[44] (version 0.7.8) was utilized to map the paired-end clean reads to the mouse reference genome (mm9). SAMtools[45] (version 1.0) is used for sorting the BAM file, and Picard (version 1.111) is utilized to mark duplicate reads. Single-nucleotide variant is then identified using GATK[46] (version 3.8), and is subsequently annotated using ANNOVAR[47] (version 2015Dec). CNV is detected by control-FREEC[48] (version 9.9).

**CNV analysis and comparison with SKY data**. Genomic CNV profiles of WGS data are identified by control-FREEC software, and then comparison of whole-chromosome CNV estimation (measured in log2 ratio) in Mice 1–4 between SKY and WGS is performed. For each chromosome in each mouse, the median log2 ratio is calculated for all 10-kb bins from WGS. For SKY, the log2 ratio is calculated as log2($ci$/median($c$)), where $ci$ is the average count for chromosome $i$, and $c$ is the median of $ci$ for all chromosomes ($i = 1–19;X$).

**Single-cell cloning preparation with subsequent WGS**. The early passage $SVZ^R$ tumor cell line (P1) from Mouse 2 was dissociated and single cells were seeded into 96-well plate. Total 31 single spheres were formed and then progressively expanded. The DNA content (ploidy) of each single clone was determined by Flow Cytometry. Then eight clones with near-2N genome and seven clones with sub-4N genome were further chosen based on $t(2;4)$ status. Among them, four clones with near-2N genome (two with $t(2;4)$ and two without $t(2;4)$) and three clones with sub-4N genome (two with $t(2;4)$ and one without $t(2;4)$) were subjected to subsequent WGS at Novogene as described above.

**CNV tree reconstruction from WGS data on single clones**. We built NJ trees from WGS data on single clones, based on the pairwise distance matrix estimated from CNV profiles. Briefly, control-FREEC provides the average copy number estimation (measured by log2 ratio) for every 10-kb sliding window in the chromosome, and the distance between two clones $a$ and $b$ is calculated as $\sum_i |r_a^i - r_b^i|$,

where $r_a^i$ (and $r_b^i$) is the log2 ratio for window $i$ in sample $a$ (and $b$), respectively. The NJ tree is then built similar with the tree built from SKY data (see the section "SKY-based single-cell phylogenetic tree building" above).

**Manual CNV tree reconstruction from WGS data on bulk cells**. The genomic CNV profiles of WGS data were used to reconstruct the phylogenetic tree. Briefly, we first identified specific CNV events, defined as events whose boundaries can be identified precisely within the resolution of Control-FREEC CNV detection (10 kb). Then, shared specific events were identified, defined as events with identical boundaries, in at least two samples. Whole-chromosome gain or loss was excluded as they were considered nonspecific events. We then determined the possible temporal order of these events. Known oncogenes or tumor suppressor genes[49] were also marked if these genes fall within the boundary of the CNV events. Finally, a manual tree was drawn, with the length of the trunk (or branch) proportional to the number of shared (or private) events between samples, respectively. The events used for reconstructing the tree were listed in the Source Data file.

**Image processing and figure assembling**. Adobe photoshop (version CS6) and ImageJ (version 1.52q) were used for image processing. Adobe illustrator (version CS6) was used for figure assembling.

**Quantification methods and statistical analyses**. We used GraphPad 6 and 8 for all the statistical analyses. Kaplan–Meier survival curves were used to compare the survivorship of the mice sacrificed due to brain tumors, soft tissue sarcomas, and/or other health concerns and the Mantel–Cox (Log-rank) test was used to statistically compare the overall survival (Fig. 1b). Unpaired, two-tailed Student's $t$ test was used to compare the statistical difference of $Pdgfr\alpha$ mRNA level from microarray data (Supplementary Fig. 1n); the NRT/fusion frequency (Fig. 2d and Supplementary Fig. 2i); the dominant NRT/fusion frequency (Fig. 2e and Supplementary Fig. 2h); the proliferation index (Fig. 3f); the frequency of non-cNRTs and NRTs in Type 1 and Type 2 tumors (Fig. 3g, h); the number of BrdU+ cells; the number of p-Erk+ cells and the ratio of p-Erk+BrdU+/p-Erk+ cells (Fig. 7d); and the number of p-Erk+Olig2+ cells (Fig. 7i). Fisher's exact test was used to compare the statistical difference of the frequency of human GBMs with WGD (Supplementary Fig. 2c); the frequency of brains with early tumor cluster(s) (Fig. 8e); the frequency of mouse brains with multifocal lesions (Fig. 8h) and the frequency of mouse brains with tumors directly associated with the SVZ only (Fig. 8i). Data were presented as standard error of the mean (SEM), and $p < 0.05$ was considered to be statistically significant in all of the statistical analyses.

**Reporting summary**. Further information on research design is available in the Nature Research Reporting Summary linked to this article.

## Data availability

The gene expression data have been deposited in the GEO database with Microarray data accession link: GSE152071. The whole-genome sequencing data reported in this paper have been deposited in the National Center for Biotechnology Information (NCBI) Sequence Read Archive (BioProject accession No. PRJNA638264).

All the other data supporting the findings of this study are available within the article and its supplementary information files and from the corresponding author upon reasonable request. A reporting summary for this article is available as a Supplementary Information file. Source data are provided with this paper.

## Code availability

The custom R codes to build neighbor-joining trees from SKY data have been deposited into the Github repository.

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

## Acknowledgements

We thank C. Li, P. Mateas, G. Tomasek, M. Best, T. Burns, L. Cregan, and E. Kim for technical assistance; members of the Zhu lab for support; Drs. T. Jacks, A. Berns, E. Lee, and A. Messing for providing $p53^{LSLR172H}$, $p53^{floxE2-10}$, $p53^{floxE5,6}$, and hGFAP-cre, respectively; and Drs. D. Wellik, S.J. Morrison, and E. Jecrois for critically reading the paper. This work is supported by grants from the NIH (2P01 CA085878-10A1, 1R01 NS053900, and R35CA197701).

## Author contributions

Y.L., B.L., W.L., J.Z.L., and Y.Z. conceived and designed the study. Y.L., Y.W., S.A., K.A.H., and B.H. performed the experiments. B.L., W.L., D.M.T., S.Z., and J.Z.L. performed computational and bioinformatics analysis of the data. K.A.H., B.H., A.R., and B.D.R. performed and analyzed serial-MRI and 3D-reconstruction data of glioma growth. C.Y.H. performed the pathological diagnoses of GEM gliomas. D.O.F. provided the expertise on SKY analysis. B.R.P. provided technical and reading assistance. Y.Z. and B.D.R. acquired the funding. Y.Z. and Y.L. drafted the paper and received contributions from all the authors.

## Competing interests

The authors declare no competing interests.
