## [Peer Review File · Nature Communications]

Reviewers' comments:

Reviewer #1 : Expert in Glioma

(Remarks to the Author):

In this manuscript the authors have used three different Tp53 mutant mouse models to investigate the spatiotemporal development of murine glioblastomas mimicking IDH wild type human glioblastomas. Using these models they have shown that tumors evolving from neural stem cells in the subventricular zone develop into highly malignant glioblastomas in spatially segregated regions via distinct patterns. As most malignant glioblastomas show activation of the PI3K/AKT pathway, both in humans and in their mouse models, the authors investigated the role of this pathway in the dissemination of the tumors. Interestingly, they show that when inhibiting this pathway early in tumorigenesis tumor dissemination was completely blocked and mice survived significantly longer. Data presented in this study provide important new insights about the initiation and manifestation of glioblastomas and may provide new strategies for treating these highly aggressive tumors especially when diagnosed early. This excellent manuscript is well written and figures are of high quality and clearly presenting the results discussed in the text.

Other comments:

The authors have annotated mouse tumors that occur at the end stage as a single mass as type 1 tumors and mouse tumors that appear at multiple sites as type 2 tumors. Genomic analyses show that tumor biopsies taken from the SVZ and from distant regions in type 1 tumors are highly similar while tumor biopsies taken from the SVZ and from distant regions in type 2 tumors are more distinct from each other. Instead of making this difference between type 1 and type 2 tumors, which for genomic and evolutionary analyses is interesting, could it be that it is actually all the same but that the so-called type 1 tumors are just tumors biopsied at a later stage when SVZ derived tumors and distal tumors have already grown together? This makes it more difficult if not impossible to find any differences between SVZ derived and distal tumors in type 1 tumors.

Unfortunately, I was unable to review the movies included.

Minor comments:

Figure 5a has a typo in manual

Figure 7a: Group 3 (space is missing between Group and 3)

Reviewer #2 : Expert in Glioma genetics

(Remarks to the Author):

The paper by Li et al postulates that, similar to IDH mutagenesis, IDH-wildtype GBM arises from a common ancestor with less aggressive phenotype. This common ancestor is at the origin of the wide genetic/genomic diversity observed in GBM and, while several works allude to this less aggressive ancestor, there are no GEM mouse models that recapitulate this possibility. To this effect, the authors generated p53-mutant mouse model systems with either deletion of a set of exons in p53 on both alleles, or with one allele carrying a common p53 mutation in human gliomas, p53R175H and the other allele containing the exon deletion or with complete removal of p53. They use these models to generate tumors in mice and follow these tumors at different time points and from different areas including the SVZ, which was recently described as a possible site of origin for these tumors. Their findings indicate that:

- the tumors resemble the proneural human tumors
- the kinetics of latency followed by rapid growth mimics human disease
- they are able to acquire metachronous tumors and synchronous tumor recapitulating genetic events identified in human disease (Ras/Mapk and Akt/Pi3k pathways) at dissemination.
- cells in SVZ harbor the cancer stem cell niche with quiescent abnormal cells potentially fueling disease.

This is a very dense manuscript on a potentially important model of glioblastoma. The data on tumor dissemination and the SVZ is well documented however there are several points that needs to be addressed to validate the relevance of the model presented.

1- P53 mutations are mainly associated with IDH mutagenesis or PDGFRA amplification/mutation in the human proneural subgroup. Other IDH-WT GBM do not have genetic TP53 alterations. As they have no PDGFRA alterations in their model how does this one fit human disease?

2- In addition why include the three p53 models if they showed equal results. It is confusing to this reviewer what is the relevance of introducing this at the beginning especially that there are no clear

differences between the tumors/genetic alterations/path of dissemination and only a minor point is made on the absence of differential effects of mutant p53 versus loss of tp53.

3- The authors state that tumors and cell lines exhibited a 'dominant expression signature' of the proneural type, but cell lines actually display an equally high enrichment for the 'classical' signature. Regarding gene expression analysis (Figure 1c-d): the analysis is not clearly explained. It is unclear what are the genes used for the four classes or how the correlation was converted to an enrichment score. Correlation of genes may simply be explained by tumor content, have the authors assessed this before preparing the expression arrays? The control mice show a high correlation for the proneural (and neural) type too, as the tumors; this suggests that the correlation observed for tumors may be due to tumor content and stroma in particular since the cell lines have a different pattern (higher for classical signatures).

4- They only do bulk WGS on 3 mice, for each they take 3-4 samples (tail, SVZ, tumor), call CNVs and build trees in a very simplistic way and the majority of their claims is based on these analyses. Indeed, to build 'single-cell phylogenetic trees', they use 'spectral karyotyping' to analyze chromosomal events. Translocations are called based on the images, and they use this information to construct trees. For the phylogenetic trees, their explanation of the method was impossible to follow. They constructed trees manually and their explanation is very convoluted. Alternatively, they used a computational algorithm (neighbor joining), but they defined so many cases inside the algorithm that it is almost a manual method. This needs to be explained in a much clearer way and as a lot of the paper hinges on this classification it would be important to have validations and/or references to other work where this technique was used.

5- Metachronous tumours had PI3K alterations in human disease (Wang et al Nature Genetics, cited by the authors). Did they identify increased alterations of this gene and of the pathway in their type II tumors compared to their type I?

6- There is a wealth of genetically engineered brain tumor mouse models showing this 'potential quiescence' before tumor arises. This raises the question as to whether what they are seeing is the natural length of time for tumors to arise in a mouse brain versus modelling glioma latency. Moreover, it is highly debatable whether IDH-WT glioblastoma arises from slower growing lesions and a p53 mutant mouse models is not the best model for these tumors as human tumors have wild-type TP53.

7- The phylogenetic reconstruction from WGS CNVs is quite rudimentary: trees are manually constructed from CNV calls; alternatively, a very simple tree-building algorithm (neighbor joining) was used. There are methods to infer clonal population structure from bulk samples (e.g. PyClone), which consider normal cell contamination and other factors that could be used.

8- the title is somehow misleading as they do not perform single cell analyses per say.

In all, this is an interesting paper, with a potentially important model of gliomagenesis. Addressing concerns related to the analyses performed and some of the data interpretation would strengthen the claims made.

Reviewer #3 : Expert in clonal evolution

(Remarks to the Author):

This is a difficult manuscript to review. There are lots of data piled together, making it difficult to follow and judge. When presenting data, authors should clearly lay out the rationale of key experiments, and then build other experiments around their key discoveries. Moreover, there are many gaps between data presented and conclusions made. I cannot support its publication.

Responses to Reviewers
NCOMMS-18-53272
Li et al.

We thank all three reviewers for their constructive and supportive comments on our manuscript. To address the critiques raised by the reviewers, we have revised the manuscript by performing additional experiments, analyzing the new data, and revising the figures/supplemental figures. The major experiments are summarized below and followed by a point-by-point response to each of the comments provided by the reviewers.

Major experiment 1: Determine two distinct growth patterns in vivo

Reviewer 1 raised an important question of whether the single-mass Type 1 pattern results from a more advanced stage of multifocal-mass Type 2 pattern in which spatially segregated tumors grow together (see Responses: 1-1). We performed serial MRI and 3D-reconstruction analysis on 43 tumor-bearing brains from all three *p53* conditional knockout (*p53CKO*) models, demonstrating the existence of both single-mass Type 1 and multifocal Type 2 growth patterns with frequencies of 30% and 70%, respectively. Of note, we also observed a small number of Type 2 tumors that merged completely, which would cause misclassification as a Type 1 case if only histological analysis was performed on end-stage tumors. More importantly, these results led to a previously unrecognized observation: the single-mass Type 1 versus multifocal-mass Type 2 pattern almost completely correlates with the acquisition pattern of clonal non-reciprocal translocations (cNRTs) in a founder cell/clone (FC) with a 4N versus 2N genome, respectively. These results provide critical insights into how the one-phase versus two-phase evolutionary pattern, respectively, impacts the single-mass Type 1 versus the multifocal-mass Type 2 growth patterns in vivo (see **Figure 2b-i and Page 6-8**).

Major experiment 2: Perform whole-genome sequencing (WGS) analysis of bulk tumor samples to complement the SKY-based single-cell/clone tree building

The major concerns raised by Reviewer 2 (See Responses: 2-4, 2-7 and 2-8) pertain to the single-cell analysis by spectral karyotyping (SKY) and WGS of bulk tumor samples in phylogenetic tree building. In the revised Results and Discussion, we have clarified the way in which we integrated these two methods to build a single-cell phylogenetic trees, revealing both clonal origin and dispersal from the SVZ stem-cell niche to distant brain regions in a genetically engineered mouse (GEM) model of *IDH*-WT GBMs (see **Page 10 and Pages 17-18**). Specifically, we performed WGS on bulk tumor samples along with their control tails on all 6 Type 2 case with a total of 31 samples. In summary, bulk-tumor-derived WGS data complemented the SKY-based single-cell/clone tree building, confirming (1) that the sample from SKY represented the bulk of the tumor samples (no sampling bias), (2) that cNRTs featured identical breakpoints, and (3) identification of alterations of cancer drivers, including loss of *Nf1* and *Pten*, during early and late phase of tumor evolution. Together, we integrated SKY-based single-cell analysis with WGS-based analysis of bulk tumor cells and single-cell derived bulk tumor cells to investigate tumor evolution (see **Figure 4 and Pages 17-18**).

Major experiment 3: Perform whole-genome sequencing (WGS) analysis of single-cell-derived bulk tumor sample to validate the SKY-based single-cell/clone tree building

In a proof-of-principle experiment, we designed a method that used WGS data to build a single-cell phylogenetic tree. Specifically, we cloned single cells into a 96-well platform, amplified single clones and performed WGS on single-cell-derived bulk tumor samples. The genome-wide profiles of copy number variations (CNVs) called by WGS data of single-cell-derived tumors were used to generate a phylogenetic tree by the neighbor-joining (NJ) method. These results not only validate the evolutionary direction of founder cell/clones (FCs) established by the SKY-based NJ tree, but also identified a precursor (SVZ^R-FC0) of the cNRT^{2N-1X}-bearing CA, SVZ^R-FC1, which harbors a cancer driver alteration: a specific homozygous deletion at *Nfi* tumor suppressor gene (see **Figure 5d and Supplementary Figure 7 and Page 13**).

Major revisions: Clarification of the single-cell tree building strategy

An important point raised by Reviewer 2 is a lack of clarity of building a single-cell phylogenetic tree. In the revised manuscript, we have specifically stated that all NJ trees were built by SKY-based single-cell/clone analysis, thus reflecting the phylogeny of tumor evolution. A new figure was dedicated to demonstrating how we used the WGS data of bulk tumor samples to complement SKY-based single-cell analysis, particularly by determining the unique breakpoint sequences of each of the cNRTs used for tree building (see **Figure 4 and Pages 10-11**). The strengths and weaknesses of SKY-based single-cell analysis versus WGS analysis of bulk tumor samples were also discussed and included in the revised manuscript (**Page 10 in Results and Pages 17-18 in Discussion**). Furthermore, we included detailed descriptions of building and interpreting the single-cell phylogenetic trees in the Methods section of the Supplementary Information:

1. SKY-based single cell phylogenetic tree building using neighbor-joining algorithm with customized feature and distance measure selection (**Pages 16-17**).
2. Clonal evolution interpretation from SKY-based single cell phylogenetic trees (**Pages 17-18**).

Given that there was no concern over the molecular mechanisms underlying local expansion and distant dispersal in Figures 6 and 7, we have made only minimal revisions to the new Figures 7 and 8 in the revised manuscript. The detailed flow of figures is included in “Response to Reviewer 3” (see below).

Below, we offer a detailed point-by-point rebuttal to the issues raised by the reviewers:

Reviewer #1 writes: “...Data presented in this study provide important new insights about the initiation and manifestation of glioblastomas and may provide new strategies for treating these highly aggressive tumors especially when diagnosed early.

This excellent manuscript is well written and figures are of high quality and clearly presenting the results discussed in the text...”

Response: We thank this reviewer for generous comments and the excellent summary of the major findings of the work.

Reviewer #1 writes: "...In this manuscript the authors have used three different Tp53 mutant mouse models to investigate the spatiotemporal development of murine glioblastomas mimicking IDH wild type human glioblastomas. Using these models, they have shown that tumors evolving from neural stem cells in the subventricular zone develop into highly malignant glioblastomas in spatially segregated regions via distinct patterns. As most malignant glioblastomas show activation of the PI3K/AKT pathway, both in humans and in their mouse models, the authors investigated the role of this pathway in the dissemination of the tumors. Interestingly, they show that when inhibiting this pathway early in tumorigenesis tumor dissemination was completely blocked and mice survived significantly longer..."

Response: We thank this reviewer for precisely and more concisely summarizing the major results of the work. We have revised the manuscript in the organization of "Results and Discussion" with the subtitles according to these comments.

Main points:

1. *"The authors have annotated mouse tumors that occur at the end stage as a single mass as type 1 tumors and mouse tumors that appear at multiple sites as type 2 tumors. Genomic analyses show that tumor biopsies taken from the SVZ and from distant regions in type 1 tumors are highly similar while tumor biopsies taken from the SVZ and from distant regions in type 2 tumors are more distinct from each other. Instead of making this difference between type 1 and type 2 tumors, which for genomic and evolutionary analyses is interesting, could it be that it is actually all the same but that the so-called type 1 tumors are just tumors biopsied at a later stage when SVZ derived tumors and distal tumors have already grown together? This makes it more difficult if not impossible to find any differences between SVZ derived and distal tumors in type 1 tumors."*

Response: We greatly appreciate this reviewer's comment on the classification between Type 1 and Type 2 tumors described in our manuscript. The reviewer suggests a possibility that the Type 1 tumors may actually represent more advanced stage of Type 2 tumors in which multifocal tumors, particularly SVZ-derived tumors, merge into a single mass. Experimentally, we performed serial MRI on a total of 43 cases with brain tumors from all three *p53* conditional knockout (*p53CKO*) models. We indeed found the evidence of the merge of spatially segregated tumors in a subset of Type 2 cases. However, both imaging and genetic studies demonstrate the existence of two distinct growth patterns, Type 1 and Type 2, which are characterized by single and multifocal tumor masses, respectively. The serial MRI revealed that single lesions developed throughout the entire screening process in approximately 30% of brains analyzed, demonstrating the existence of the single-mass Type 1 growth pattern. Therefore, Type 1 tumors do not necessarily represent a later stage of Type 2 tumors. As the reviewer rightfully pointed out, we also observed the merge of spatially segregated tumors in 43% (13 of 30) with the Type 2 growth pattern. Among those 13 Type 2 cases, 5 exhibited a partial merge of some of the multifocal tumors, but remained multifocal at the end stage. Notably, the 8 other cases presented as a single mass at the end stage were the result of the merge of two or more spatially segregated lesions.

Thus, a small number of Type 2 cases could potentially be misclassified as Type 1 cases, if only based on histological analysis of tumors at the end stage. In summary, we found that while it is possible for a small subset of Type 2 tumors to merge at the end stage, the majority of Type 2 tumors remain multifocal at the end stage (see **Figure 2b-c, Supplementary Movies 1-4 and Pages 6-7**).

More importantly, one of the major findings from the manuscript revision is the realization that the Type 1 versus Type 2 pattern is highly correlated with the ploidy of a tumor founder cell/clone (FC) with a tetraploid (4N) versus diploid (2N) genome, respectively. We analyzed 6 cases with both serial MRI and spectral karyotyping (SKY) data. Tumors from all 3 single-mass Type 1 cases carried a single copy of one or more clonal non-reciprocal translocation(s) (cNRT), accompanied by three copies of the two translocated chromosomes. These observations suggest that Type 1 tumors arise from a FC with a sub-4N genome, which has already undergone whole-genome duplication (WGD) and subsequently acquired a cNRT(s) (termed as cNRT^{4N}). As a result, the acquisition of a cNRT^{4N} led to loss of one copy of each of the two translocated chromosomes and 3 copies in a sub-4N genome. Strikingly, all tumors from the 3 multifocal-mass Type 2 cases, despite also having sub-4N genomes, carried two copies of one or more cNRT(s) accompanying with only two copies of the translocated chromosomes. The pattern of NRT acquisition in Type 2 tumors is consistent with a model wherein a cNRT(s) is acquired in a FC cell with a 2N genome before undergoing WGD. Thus, this type of cNRTs is termed as cNRT^{2N}, which is accompanied by two copies of the two translocated chromosomes. In total, we confirmed the acquisition pattern of cNRT^{4N} and cNRT^{2N} in a total of 7 Type 1 cases and 6 Type 2 cases, respectively. Of note, the presence of a cNRT^{2N} validated one of the Type 2 cases (Mouse 6), which was initially misclassified as a Type 1 case by histological analysis due to the merge of two spatially segregated tumors, but subsequently reclassified by the serial MRI and 3D-reconstruction analysis (see **Figure 2d-i, Supplementary Movies 1-4 and Pages 7-8**).

Despite both exhibiting sub-4N genomes and loss of Pten/chr19 at the end stage, distinct patterns of cNRT^{4N} and cNRT^{2N} in Type 1 and Type 2 tumors suggests distinct mechanisms at the initiating stage for the generation of single mass and multifocal masses during tumor evolution. After having performed analysis of multiple segments of the same tumors (including the SVZ) and early biopsied samples, we observed no evidence for the presence of tumor precursor cells with 2N genomes and/or normal Pten/chr19 in any of Type 1 cases analyzed. In contrast, we observed tumor precursor populations with near-2N genomes and/or normal Pten/chr19 in 5 of the 6 Type 2 cases analyzed. Specifically:

1. Three Type 2 cases contained near-2N/normal Pten/chr19 tumor precursor cells were predominant, if not exclusive, in the spatially segregated SVZ. More importantly, these near-2N/normal Pten/chr19 tumor precursor cells in the SVZ exhibited a less proliferative phenotype compared to clonally related highly proliferative sub-4N/Pten/chr19-loss tumor cells in the distant brain parenchyma (see **Figures 3 and 5, Supplementary Figures 3-5, and Pages 8-10**).
2. Two Type 2 cases contained tumor precursor cells with near-2N genomes, but already underwent loss of Pten/chr19 (see **Figure 6a-b and Pages 13-14**).

3. All 4 tumors with two copies of a cNRT^{2N} from one Type 2 case exhibited no evidence of near-2N tumor cells (see **Figure 6c and Pages 13-14**).

Despite potentially being analyzed at different evolutionary stages, SVZ-derived tumor cells from all 6 Type 2 cases consistently harbored fewer cytogenetic events than their autologous parenchyma-derived tumor cells. Thus, we propose a two-phase evolutionary model in the Type 2 pattern, with spatial segregation of tumor origin and manifestation and with slow and rapid proliferative phases spatially segregated in the SVZ and distant brain parenchyma, respectively. Importantly, the Type 2 pattern recapitulates a unique evolutionary pattern proposed by a recent study that revealed that low-level cancer driver mutations were observed in the tumor-free SVZ in a subset of human *IDH*-WT GBMs, but not other brain tumors including *IDH*-mutant GBMs. Of note, the Type 1 pattern possesses the rapid proliferative phase in the brain parenchyma, but lacks the less proliferative phase in the SVZ observed in the Type 2 pattern. The Type 1 pattern resembles the rapid progression course observed in human primary/de novo GBMs (that have wild-type *IDH1* or *IDH*-WT).

2. “Unfortunately, I was unable to review the movies included.”

Response: To better visualize the Type 1 versus Type 2 pattern of tumor evolution, we performed three-dimensional (3D) reconstruction of serial MRI images of the tumors. Movie 1 shows a representative case of the Type 1 pattern with one single-lesion growth. Movies 2~4 show examples of the Type 2 pattern, in which multiple lesions initiated at distinct locations, but eventually merged together and formed one single lesion. These movies are prepared as Supplementary Movies with QuickTime movie (.mov) format (see **Figure 2b and Supplementary Movies 1-4**).

Minor comments:

1. “Figure 5a has a typo in manual”

Response: We have removed this part of the Results that used bulk tumor samples to build the Neighbor-Joining tree (see more details in “Response to Reviewer 2 Critique #4 below”).

2. “Figure 7a: Group 3 (space is missing between Group and 3)”

Response: We have reorganized the figures and have corrected the layout (see **Supplementary Figure 3a and 4**).

We sincerely thank this reviewer for pointing out the significance of distinguishing the two growth patterns observed in Type 1 and Type 2 cases. This important distinction led to the elucidation of two patterns of tumor evolution in our mouse models, which mimic the two patterns recently identified in locally and distally recurred human *IDH*-WT GBMs. Most importantly, the identification of cNRT^{4N} and cNRT^{2N} allows us to infer the presence of two different types of FCs, driving one-phase rapid tumor evolution in Type 1 tumors versus spatially segregated two-phase evolution from the SVZ to distant dissemination in Type 2 tumors. Based

on these new results, we have revised the order of the figures (see details in “Response to Reviewer 3”).

Reviewer #2 writes: “...In all, this is an interesting paper, with a potentially important model of gliomagenesis. Addressing concerns related to the analyses performed and some of the data interpretation would strengthen the claims made...”

Response: We thank this reviewer for the positive comments and specific critiques for improving our manuscript.

Reviewer #2 writes: “...The paper by Li et al postulates that, similar to IDH mutagenesis, IDH-wildtype GBM arises from a common ancestor with less aggressive phenotype. This common ancestor is at the origin of the wide genetic/genomic diversity observed in GBM and, while several works allude to this less aggressive ancestor, there are no GEM mouse models that recapitulate this possibility. To this effect, the authors generated p53-mutant mouse model systems with either deletion of a set of exons in p53 on both alleles, or with one allele carrying a common p53 mutation in human gliomas, p53R175H and the other allele containing the exon deletion or with complete removal of p53. They use these models to generate tumors in mice and follow these tumors at different time points and from different areas including the SVZ, which was recently described as a possible site of origin for these tumors. Their findings indicate that:

- the tumors resemble the proneural human tumors
- the kinetics of latency followed by rapid growth mimics human disease
- they are able to acquire metachronous tumors and synchronous tumor recapitulating genetic events identified in human disease (Ras/Mapk and Akt/Pi3k pathways) at dissemination.
- cells in SVZ harbor the cancer stem cell niche with quiescent abnormal cells potentially fueling disease...”

Reviewer #2 also writes: “...This is a very dense manuscript on a potentially important model of glioblastoma. The data on tumor dissemination and the SVZ is well documented however there are several points that needs to be addressed to validate the relevance of the model presented...”

Response: We thank this reviewer for providing an excellent outline of the major findings of the work as well as specific issues to be addressed for the revised manuscript. However, we regret that we did not clearly explain whether, similar to *IDH*-mutant GBMs, *IDH*-WT GBMs arise from a common ancestor with less aggressive phenotype. Indeed, our results reveal two distinct growth patterns in GEM models of *IDH*-WT GBMs, which are both fundamentally different from the progressive nature of *IDH*-mutant GBMs. The tumors with the Type 1 pattern arise from a cNRT^{4N}-bearing FC that has already undergone WGD with a sub-4N genome and loss of *Pten*/chr19, driving a rapidly proliferative single mass. We observed no evidence of the presence of less proliferative precursor cells in Type 1 cases, despite performing multi-segmental and early biopsy analysis. In contrast, the acquisition pattern of cNRT^{2N} suggests that tumors with the Type 2 pattern arise from a FC with near-2N genome. Indeed, cNRT^{2N}-bearing less proliferative tumor precursor cells with near-2N genomes and normal *Pten*/chr19 were predominantly, if not exclusively, observed in the SVZ of a subset of Type 2 cases analyzed. The most critical aspect of the Type 2 pattern is the spatial segregation of tumor origin in the SVZ and tumor manifestation in the distant brain parenchyma (see **Discussion in Pages 18-19**).

Main points:

1. “*P53 mutations are mainly associated with IDH mutagenesis or PDGFRA amplification/mutation in the human proneural subgroup. Other IDH-WT GBM do not have genetic TP53 alterations. As they have no PDGFRA alterations in their model how does this one fit human disease?*”

Response: We agree with the reviewer that “*P53 mutations are mainly associated with IDH mutagenesis or PDGFRA amplification/mutation in the human proneural subgroup*”. According to the TCGA datasets ¹, 100% of *IDH*-mutant GBMs have *TP53* mutations based on the exome sequencing data. However, our analysis of the TCGA datasets revealed a high frequency of *TP53* mutations in *IDH*-WT Proneural GBMs (43.9%; 18 out of 41). It should be noted that *TP53* mutations are also present in the other three subtypes of *IDH*-WT GBMs, albeit with less frequency [Neural: 15.8% (6/38); Classical: 14.7% (10/68) and Mesenchymal: 30.1% (22/73)] (see **Supplementary Figure 1m and the attached Figure panel a for reviewers**). Thus, our *p53CKO* models with the Proneural signature are relevant to *IDH*-WT GBMs in humans, particularly in line with a recently proposed hypothesis that all *IDH*-WT GBMs arise from the stem glioma with the Proneural signature ².

Despite an association between *TP53* mutation and *PDGFRA* amplification/mutation in the human Proneural subgroup, nearly half of human *TP53*-mutant *IDH*-WT Proneural GBMs (8 out of 18 in the TCGA datasets) do not have any genetic alterations of *PDGFRA* (**Supplementary Figure 1m and the attached Figure panel a for reviewers**). Thus, our *p53CKO* models potentially recapitulate a subset of *TP53*-mutant *IDH*-WT Proneural GBMs without genetic alterations in *PDGFRA*. However, approximately 50% of the tumors analyzed from our *p53CKO* models exhibited high levels of both mRNA and protein expression of *PDGFRA*/*PDGFR*, suggesting a non-genetic mechanism of activating *PDGFR* signaling in these cases. In summary, malignant gliomas/GBMs from the *p53CKO* models recapitulate human *IDH*-WT/*TP53*-mutant Proneural GBMs with both high and low levels of *PDGFRA* signaling (see **Figure 1d and Supplementary Figure 1n**).

2. “*In addition why include the three p53 models if they showed equal results. It is confusing to this reviewer what is the relevance of introducing this at the beginning especially that there are no clear differences between the tumors/genetic alterations/path of dissemination and only a minor point is made on the absence of differential effects of mutant p53 versus loss of tp53.*”

Response: The reviewer is correct in pointing out a lack of clarity in the rationale of using *p53CKO* models with three distinct mutations or deletions. While the majority of human cancers, including GBMs, harbor missense point mutations in *TP53* that often accumulate high levels of mutant p53 protein, most GEM models employ loss of function of *p53* alleles instead. One recent study found no evidence of gain-of-functions of mutant *TP53* alleles in human myeloid cancers, challenging this well-established concept in the p53 field. Importantly, our study provides a complex picture regarding the existence of oncogenic activity of mutant *p53* protein beyond loss of function. On the one hand, the two *p53* mutant alleles, encoded by a hot-spot missense point mutation (*p53*^{R172H}) or an in-frame deletion (*p53*^{ΔE5-6}), despite both accumulating high levels of

mutant p53 proteins in tumors, did not produce more severe tumor-promoting phenotypes (e.g. tumor latency and GBM penetrance) than those induced by a conditional p53-null allele ($p53^{\Delta E2-10}$) (**Figure 1b-f and Supplementary 1c-d, j**). On the other hand, malignant gliomas and GBMs driven by these two p53 mutant alleles, but not the p53 null allele, exhibited high levels of genomic instability with frequent accumulation of clonal non-reciprocal translocations (cNRTs), a phenomenon observed in many human cancers such as prostate cancer, lung adenocarcinoma and GBMs (**Figure 2d-e**). The TCGA study demonstrates that up to 32 distinct genomic structural rearrangements were observed in over 80% (34 out of 42) of the human GBMs¹. These results not only reveal additional oncogenic activities of mutant p53 alleles in GEM GBM models, but also demonstrate that p53-mutant driven GBM models are more similar to human GBMs than those induced by p53-null alleles.

More importantly and paramount to our manuscript is the accumulation of unique/non-sharing cNRTs in malignant gliomas and GBMs driven by mutant p53 alleles, but not the p53-null allele. The unique cNRTs, particularly those acquired in a FC with a 2N genome (cNRT^{2Ns}), were used as a neutral marker to build a single-cell phylogenetic tree, tracing the common ancestor (CA) from the SVZ stem-cell niche to distant regions of brain parenchyma during tumor evolution in Type 2 cases. Thus, our p53-mutant driven models with a lineage-tracing marker provide a unique platform to study the different signaling pathways that contribute to the tumor initiation, progression and recurrence at the clonal level. The rationale of using three p53CKO models was presented at the start of the Results section (**see Pages 4-5**).

3a. *“The authors state that tumors and cell lines exhibited a ‘dominant expression signature’ of the proneural type, but cell lines actually display an equally high enrichment for the ‘classical’ signature. Regarding gene expression analysis (Figure 1c-d): the analysis is not clearly explained. It is unclear what are the genes used for the four classes or how the correlation was converted to an enrichment score. Correlation of genes may simply be explained by tumor content, have the authors assessed this before preparing the expression arrays? The control mice show a high correlation for the proneural (and neural) type too, as the tumors; this suggests that the correlation observed for tumors may be due to tumor content and stroma in particular since the cell lines have a different pattern (higher for classical signatures).”*

Response: We thank this reviewer for pointing out a lack of clarity in our methods and analysis. In the revision, we revised the “Methods” section under the subtitle of “Gene expression analysis” to provide details of this analysis (**see Pages 18-19**). Specifically, we used the classification system established by Verhaak et al.³, which defined four signatures with a total of 840 discriminative genes (210 genes for each signature), classifying GBMs into four subtypes, Proneural, Neural, Classical and Mesenchymal. We used a Pearson correlation by comparing the expression pattern of these 840 feature genes between our GEM gliomas and individual human GBMs with the known subtypes in the TCGA datasets (**Figure 1e**). A positive correlation suggests that a given GEM tumor most resembles the GBM subtype based on its expression of the feature genes. Based on this analysis, malignant gliomas and GBMs from our GEM models have a high correlation with human Proneural subtype GBMs. It should be noted that we mistakenly labeled the Pearson correlation value as an enrichment score in the previous Figure 1c and 1d.

Despite our confidence in this method, we agree with the reviewer's concern that this correlation-based prediction can be confounded by factors such as tumor purity. To address potential contamination of normal brain tissues during tumor dissection, we not only validated each tumor sample by histopathological analysis (Dr. Chen-Ying Ho) to ensure tumor purity of the samples and any samples with significant contamination of brain tissues were labeled in the Figures with an "asterisk". We further established early-passage self-renewing neurosphere cultures (cell lines) from original tumors under serum-free conditions. This culture condition is widely used to enrich glioma stem cells known to maintain the phenotype and genotype of primary brain tumors, and more importantly, would exclude contamination of normal brain cells⁴. Of note, two tumor samples, including one revealed by histological analysis with significant contamination of normal brain tissues after dissection, exhibited an additional Neural signature (**marked with "*" in Figure 1f**). We performed side-by-side analysis of 11 matched tumors and tumor-derived primary cell lines. Importantly, these early-passage tumor-derived cell lines retained the strong Proneural signatures as of their parental tumors, but eliminated the Neural signature, likely due to eliminating normal tissue contamination (**Figure 1f**). Cell-lines acquired a moderate level of the Classical signature, likely a result of the presence of high concentration of EGF in neurosphere culture media activates EGFR signaling, as Classical signature GBMs are associated with EGF signaling. This observation further supports the conclusion that malignant gliomas and GBMs from our GEM models strongly resemble the human Proneural subtype GBMs.

Third, we applied an enrichment-based method using the same feature genes. This method, called single sample gene set enrichment analysis (ssGSEA), was used to predict GBM subtypes from >500 human GBMs in a recent TCGA study¹. Primarily, this method serves to aggregate the expression of the feature genes into a score, which reflects the overall activity of the feature genes and thus represents a specific subtype. Consistent with the correlation-based approach, we found that malignant gliomas and GBMs from our GEM models strongly express the Proneural signature (**Supplementary Figure 1j**). Furthermore, the ssGSEA analysis demonstrates that the tumor-derived cell lines retained strong Proneural signatures with some degree of activation of the Classical signature (**Supplementary Figure 1j**). For comparison, we have retained the original correlation-based data (**Figure 1e**) and included the ssGSEA results (**Supplementary Figure 1j**). Together, both correlation- and enrichment-based methods demonstrate that malignant gliomas and GBMs and tumor-derived cell lines from the *p53CKO* models most resemble Proneural subtype of human GBMs.

3b. The reviewer rightfully pointed out "...*The control mice show a high correlation for the proneural (and neural) type too, as the tumors...*"

Response: This is not due to stroma contamination as suggested ("*this suggests that the correlation observed for tumors may be due to tumor content and stroma in particular since the cell lines have a different pattern (higher for classical signatures)*"). Indeed, the Proneural and Neural signatures represent two normal brain cell populations, oligodendrocyte precursor cells (OPCs) and neuronal lineages, respectively³. Thus, the control tissues used in the study (cerebral cortical tissues from the control mice) are expected to express high levels of both Proneural and Neural signatures, which are validated by both the Pearson Correlation and ssGSEA methods. To validate this observation with human tissues, we analyzed 10 non-neoplasia brain samples from

the TCGA datasets, which showed strong Neural and Proneural signatures (**as did mouse brains, see the attached Figure panel b for reviewers**).

4. *“They only do bulk WGS on 3 mice, for each they take 3-4 samples (tail, SVZ, tumor), call CNVs and build trees in a very simplistic way and the majority of their claims is based on these analyses. Indeed, to build 'single-cell phylogenetic trees', they use 'spectral karyotyping' to analyze chromosomal events. Translocations are called based on the images, and they use this information to construct trees. For the phylogenetic trees, their explanation of the method was impossible to follow. They constructed trees manually and their explanation is very convoluted. Alternatively, they used a computational algorithm (neighbor joining), but they defined so many cases inside the algorithm that it is almost a manual method. This needs to be explained in a much clearer way and as a lot of the paper hinges on this classification it would be important to have validations and/or references to other work where this technique was used”.*

Response: We thank the reviewer for pointing out a lack of clarity on the NJ trees built by bulk tumor-based WGS data versus single-cell SKY data. In the revised version, we provided the rationale to use of SKY-based single cell/clone analysis complemented by WGS of bulk tumor samples (see **Pages 10-11 and Page 18**):

1. The SKY technology is a chromosome analysis at the true single-cell level, as it provides the structural and numerical chromosomal alternations in each single metaphase of cells/clones. This cytogenetic analysis has been widely used to detect the complex chromosomal aberrations in human disease diagnosis, and has been extended to mouse and rat chromosomes. Thus, we have clarified that the SKY-based tree is a true genealogy, but not the one built by WGS data of bulk tumor samples. Specifically, the presence of cNRTs and WGD in p53-mutant driven malignant gliomas and GBMs allowed us to use the SKY method to perform single-cell/clonal analysis of tumor evolution. Both NRT and WGD can be readily determined by the SKY method, but are relatively challenging to identify using the current single-cell WGS technology. This is because detection of an NRT requires the identification of chromosomal fusion, which relies on split reads mapping. However, single-cell DNA sequencing usually have sparse coverage on most genome locations, rendering low probability to have reads covering the fusion junction. Thus, SKY still has an advantage to detect large-scale chromosomal translocation events.
2. Despite providing single-cell resolution, the SKY method only reveals large, chromosome-level changes assayed from a small number of proliferating cells. In contrast, WGS of bulk samples profiles the average genome of a large population of cells and, importantly, has the potential to uncover additional genetic alterations down to single-nucleotide resolution. Accordingly, we performed WGS on bulk tumor cells along with their control tails on all 6 Type 2 case with a total of 32 samples (**Figure 6f-k**). We also used bulk-tumor-derived WGS data to validate and complement the SKY-based single-cell/clone tree building (**Figure 5d**). First, we showed that CNVs called from the WGS data were highly consistent with the chromosomal gains and losses determined by the SKY data (**Figure 4a and Pearson correlation coefficient =0.85; Figure 4b**). These results demonstrate that the small number of tumor cells analyzed by the SKY method are representative to the whole tumor cell populations, confirming no significant sampling bias from the SKY method. Second, we used

WGS data to “genotype” the individual breakpoints of NRTs in each sample. In total, 20 out of 26 NRT events were identified by WGS confidently with high quality reads. We confirmed that cells from the same brain shared the same breakpoints, thus establishing the validity of defining NRT as a single event (not recurrent) when building a tree (**Figure 4c-f**). That is, WGS data confirmed that all cells with the same NRT(s) arise from the same genetic event, sharing the CA. Third, WGS data identified cancer driver alterations, including loss of *Nf1* and of *Pten*, acquired in SVZ-derived and parenchymal tumors, respectively (**Figure 5d and Figure 6f-k**).

3. We have revised the Methods section that describes the use of the NJ algorithm to build a phylogenetic tree by SKY data. It should be noted that no manual method was used. In summary, we specified three critical steps: (1) feature selection, or feature construction, (2) selection of a distance measure, and (3) reconstruction and interpretation of the phylogenetic trees (see **Methods under the subtitle “SKY-based single cell phylogenetic tree building using neighbor-joining algorithm with customized feature and distance measure selection”, Pages 16-17**):

Step 1 - the feature selection: we compiled all chromosomal events, including NRT, WGD, chromosomal fusion, gain and loss, from SKY data of tumor cells of each site of all Type 2 cases.

Step 2 - selection of a distance measure: we applied a computational algorithm (neighbor-joining or NJ) to build a phylogenetic tree at the single-cell/clone level by converting chromosomal events into a distance matrix under the minimum evolution principle. We assigned every chromosomal event (e.g. gain or loss) at weight of “1” with two exceptions. Given that the phylogenetic tree is built on the lineage tracing of a cNRT^{2N}-bearing FC, we first gave a higher weight of “5” for the de novo acquisition of an NRT from one cell without the NRT to the NRT-bearing cell. An NRT is given higher weight than chromosomal copy-number change because it shares the same breakpoint at the single nucleotide level. This feature makes it highly unlikely to reproduce an exact NRT in different cell lineages and thus can be considered as a unique identifier or neutral marker. Second, we considered WGD as a single genetic event and gave a weight of “1” instead of “40”, due to duplication of all 40 chromosomes. For example, a cell acquiring cNRT^{2N-1X} (at 2N stage) and subsequently undergoing WGD with 2 copies of the cNRT^{2N-2X} receives a weight of “6” (NRT+WGD = “5+1”) instead of “10” (NRT+NRT = “5+5”).

Step 3 - reconstruction and interpretation of the phylogenetic trees: The algorithm design is detailed in the revised Methods section. Using this NJ algorithm, we built a phylogenetic tree for tumor cells derived from each site of all 6 Type 2 cases. Each dot on the tree represents a single proliferating cell/clone (metaphase cell), while the distance between every cell and normal 2N cell represents the number of somatic chromosomal events accumulated during tumor evolution. Based on the NJ tree, we delineated tumor evolution at each site in a given Type 2 case by identifying a small number of clonally related cNRT^{2N}-bearing FCs, which gradually accumulated one or more chromosomal events. Once the genotypes of all FCs were identified for each tumor site, we used two criteria to determine the evolutionary direction of FCs carrying the same cNRT(s): (1) an irreversible transition by WGD followed by random loss of chromosomes

(from a near-2N to sub-4N clone) and (2) the transition from simple to more complex genomes defined by accumulation of additional chromosomal events, including the universal loss of *Pten*/chr19 in parenchymal tumors in all Type 1 and Type 2 cases. One exception was observed in a Type 2 case that parenchymal tumors acquired a *Pik3ca*^{H1047R} point mutation and activated PI3K/Akt signaling independent of *Pten* loss. Consequently, we revealed that tumor evolution in Type 2 cases arises from a single cNRT^{2N}-bearing FC with a near-2N genome and normal *Pten*/chr19 in the SVZ stem-cell niche, whose clonally related progeny evolve locally with some of the progeny to undergo WGD and become sub-4N cells. Importantly, local evolution in the SVZ tends to generate relatively quiescent tumor cells with normal *Pten*/chr19. In contrast, some progeny of the cNRT^{2N}-bearing FC appear to acquire additional chromosomal events with more complex genomes, seeding into one or more regions in the distant brain parenchyma and becoming highly proliferative high-grade glioma and GBM cells with sub-4N genomes and *Pten*/chr19 loss (**Figures 5, 6a-c and Supplementary Figure 6**).

We regret that we did not make it clear that WGS data of bulk tumor samples profiles the average genome of a large population of cells, and thus the NJ trees built from CNVs of the bulk WGS data cannot be interpreted as the lineage relationship of the two samples. Thus, we have removed this data in the revised manuscript.

5. “*Metachronous tumours had PI3K alterations in human disease (Wang et al Nature Genetics, cited by the authors). Did they identify increased alterations of this gene and of the pathway in their type II tumors compared to their type I?*”

Response: As detailed in “Response to Reviewer 1, Critique #1” and “Response to Reviewer 2, General Critique” above, no difference in genetic alterations of the PI3K/AKT signaling pathway was observed between Type 1 and Type 2 tumors at the end stage, which universally exhibited loss of *Pten*/chr19 with activation of PI3K/AKT signaling pathway (**Figure 1d and Supplementary Figure 5a, b**). One key difference between these two types of growth pattern is that Type 1 tumors arise from a FC with loss of *Pten*/chr19 and a sub-4N genome while, in contrast, Type 2 tumors arise from a FC with normal *Pten*/chr19 and a near-2N genome, which is predominantly, if not exclusively, observed in the SVZ and spatially segregated from highly proliferative autologous sub-4N tumors with *Pten*/chr19 loss in the brain parenchyma (**Figure 3 and Supplementary Figure 5c-f**).

6a. “*There is a wealth of genetically engineered brain tumor mouse models showing this ‘potential quiescence’ before tumor arises. This raises the question as to whether what they are seeing is the natural length of time for tumors to arise in a mouse brain versus modelling glioma latency.*”

Response: We regret that we did not make this distinction clear at this critical point of our study. Indeed, we did not observe a “potential quiescence” stage in the brain parenchyma. Specifically, approximately 30% of malignant gliomas and GBMs exhibited the Type 1 pattern, which arise from a rapid proliferative FC with no evidence of the “potential quiescence” phase. To our knowledge, we were the first to use a combination of serial MRI and MRI-guided biopsy analysis to demonstrate the rapid growth pattern of malignant gliomas and GBMs in a model of *IDH*-WT GBM (**Supplementary Figure 3**). More importantly, the Type 2 pattern recapitulates

the spatial segregation of tumor initiation in the SVZ and manifestation in the distant brain parenchyma – a pattern recently identified in a subset of human *IDH*-WT GBMs, but not in other brain tumors⁵. Thus, the unique aspect of the Type 2 pattern observed in 70% of malignant gliomas and GBMs in *p53*CKO models is spatial segregation of the “quiescence” phase in the SVZ and the rapid tumor progression phase in the distant brain parenchyma. Together, our *p53*CKO models exhibited two unique patterns observed in human *IDH*-WT GBMs, which have not investigated or modelled in the previously published models (**see the Discussion in Pages 18-19**).

6b. *“Moreover, it is highly debatable whether IDH-WT glioblastoma arises from slower growing lesions and a p53 mutant mouse models is not the best model for these tumors as human tumors have wild-type TP53.”*

Response: A recent study showed that in half of *IDH*-WT GBMs, but not other types of brain tumors, including *IDH*-mutant GBMs, the SVZ niche contained low-level driver mutations compared with spatially segregated GBMs in distant regions⁵. This observation suggests that *IDH*-WT GBMs could arise from a slow growing precursor cell, located in the spatially segregated tumor-free SVZ. Among the 9 *IDH*-WT GBM cases reported, two of them had *TP53* driver mutations⁵. Furthermore, as described in “Response to Reviewer 2, Critique #1”, *TP53* mutations are one of the most frequently mutated genes in *IDH*-WT GBMs, enriched in nearly 44% of the Proneural *IDH*-WT subtype GBMs¹. Thus, our *p53*CKO models recapitulates human *IDH*-WT/*TP53*-mutant Proneural tumors and, more importantly, these models exhibited two unique features observed in human *IDH*-WT GBMs as described above. Of note, a recent study based on comprehensive characterization of human cancer genomes reconstructed the life history and evolution of mutational processes and driver mutation sequences of 38 types of cancer. This study revealed that bi-allelic *TP53* inactivation by a *TP53* mutation with accompanying 17p deletion (harboring *TP53*) is one of the most frequent early mutations in a variety of cancers, including GBM⁶.

7. *“The phylogenetic reconstruction from WGS CNVs is quite rudimentary: trees are manually constructed from CNV calls; alternatively, a very simple tree-building algorithm (neighbor joining) was used. There are methods to infer clonal population structure from bulk samples (e.g. PyClone), which consider normal cell contamination and other factors that could be used.”*

Response: As discussed in details in “Response to Reviewer 2 Critique #4 above, we used the SKY data to build single-cell/clone phylogenetic trees, delineating the lineage relationship between tumor cells from multiple spatially segregated sites from the same brains in all 6 Type 2 cases. As WGS of bulk samples profiles the average genome of a large population of cells, the NJ trees built from CNVs of these data does not reflect genealogy and thus have been removed from the revised manuscript. Accordingly, we did not perform PyClone analysis to infer clonal population structure from bulk tumor samples. Of note, our WGS analysis was performed on early-passage tumor cell lines, which have largely eliminated normal cell contamination. Importantly, we performed a proof-of-principle experiment to validate the SKY-based NJ tree using WGS analysis of single-cell-derived bulk tumor samples (**Figure 5d**). Specifically, we (1) cloned individual cells from one tumor of a Type 2 case into a 96-well platform, (2) amplified single cells into neurospheres, (3) screened them for the absence and presence of the cNRT and

WGD used for SKY-based tree building, (4) performed WGS on single-cell-derived bulk tumor samples, (5) called CNVs from these WGS data and built the CNV-based NJ tree (**See Methods under the subtitle “CNV tree reconstruction from WGS data on single clones” for details in Page 20**). The whole-genome analysis not only validates the evolutionary direction from a clone without the cNRT to a clone with the cNRT initially determined by the SKY method, but also identifies a clonal relationship between cells without and with the cNRT. Importantly, the WGS data revealed that these clones shared a homozygous deletion of the tumor suppressor gene *Nf1*.

8. “the title is somehow misleading as they do not perform single cell analyses per say.”

Response: As discussed in details in “Response to Reviewer 2 Critique #4 and #7 above, our study used SKY-derived chromosomal events to build a phylogenetic tree to reveal tumor evolution at the single-cell/clone level. We also validated the SKY-based single-cell/clone trees by using WGS analysis of bulk tumor samples and single-cell-derived bulk tumor samples. However, the title “Single-cell derived murine models of *IDH*-wild type glioblastoma exhibit a unique evolutionary pattern with spatial segregation of tumor initiation and manifestation” in the previous version was intended to demonstrate that malignant gliomas/GBMs from the *p53*^{R172H}CKO and *p53*^{ΔE5-6}CKO models arise from a single NRT^{2N}-bearing precursor cell with near-2N genome in the SVZ. To avoid any potential confusion, we have removed the term of “single-cell” with a new title: “Murine models of *IDH*-wild type glioblastoma exhibit a unique evolutionary pattern with spatial segregation of tumor initiation and manifestation”.

We would like to thank the reviewer for their constructive and thorough comments that greatly helped improve our manuscript.

Reviewer #3 writes: “...This is a difficult manuscript to review. There are lots of data piled together, making it difficult to follow and judge. When presenting data, authors should clearly lay out the rationale of key experiments, and then build other experiments around their key discoveries...”

Response: In this revised manuscript, we have made specific efforts to simplify the main messages and flow of the study, particularly based on the excellent summary of both Reviewers 1 and 2 on our manuscript. We have divided the revised manuscript into three parts:

Part 1. Development of GEM models for *IDH*-wild type glioblastoma (*IDH*-WT GBM)

Figure 1 and Supplementary Figure 1. *p53*-mutant and *p53*-null alleles are equally sufficient to induce malignant gliomas similar to human primary *IDH*-WT/non-G-CIMP GBM.

Key findings: All three *p53* conditional knockout (*p53*CKO) models developed highly penetrant malignant gliomas and GBMs similar to human *IDH*-WT GBM, including histopathology, Proneural subtype, non-GIMP phenotype.

Figure 2 and Supplementary Figure 2. Two distinct growth patterns are associated with different acquisition patterns of cNRTs in *p53*-mutant model.

Key findings:

1. High levels of chromosomal abnormalities not only demonstrate additional oncogenic activities of mutant *p53* alleles beyond loss of functions, but also clonal NRTs or cNRTs provide a neutral marker to trace lineage progression during tumor evolution.
2. Similar to human *IDH*-WT GBMs, two distinct growth patterns were observed: single-mass Type 1 and multifocal-mass Type 2 pattern, which are highly correlated with the ploidy of a founder cell/clone (FC) with a tetraploid (4N, cNRT^{4N}) and diploid (2N, cNRT^{2N}) genome, respectively.

In summary, the presence of unique one or more cNRT(s), particularly cNRT^{2N}, in *p53*-mutant driven malignant gliomas and GBMs, provides the basis of tracing clonal origin and clonal dispersal in multifocal-mass Type 2 tumors.

Part 2. Delineation of tumor evolution from the SVZ stem-cell niche to distant brain parenchyma in GEM malignant gliomas and GBMs at the single-cell/clone level.**Figure 3 and Supplementary Figures 3-5. A two-phase evolutionary pattern spatially segregated in the SVZ and distant brain regions.****Key findings:**

1. Multi-segment and early biopsy tissue analysis revealed no evidence of tumor precursor populations with near-2N genomes or normal *Pten*/chr19 in radiographically detectable Type 1 tumors, suggesting a one-phase rapid growth pattern.
2. The Type 2 pattern exhibits a two-phase pattern characterized by near-2N/normal *Pten*/chr19 tumor precursor populations maintained in the SVZ and spatially segregated sub-4N/*Pten*/chr19-loss malignant gliomas and GBMs in the distant brain parenchyma.

Figure 4. Validation of SKY-based cytogenetic markers used for building single-cell phylogenetic trees.**Figure 5 and Supplementary Figures 5-7. Single-cell phylogenetic trees reveal the two-phase evolutionary pattern in Type 2 cases.****Figure 6. Distant and local dispersal from the SVZ to distant brain regions in Type 2 cases.****Key findings:**

1. WGS of bulk tumor samples validates the rationale of using SKY-based chromosomal events (e.g. cNRTs and whole-genome duplication, WGD) to build a phylogenetic tree for tumor evolution.

2. Using a Type 2 case as an example (Mouse 2) reveals the two-phase evolutionary pattern at a single-cell/clone level: (1) local evolution of a cNRT^{2N}-bearing common ancestor (CA) in the SVZ and (2) distant dispersal of multiple clonally related progeny of CA.
3. Reconstruction of a single-cell/clone phylogenetic trees for all Type 2 cases.
4. Summary of one-phase and two-phase evolutionary pattern for single-mass Type 1 and multifocal Type 2 tumors.

Part 3. Molecular mechanisms underlying local evolution and distant dispersal in a subset of Type 2 cases.

Figure 7 and Supplementary Figure 8. Clonal expansion in the SVZ was caused by increased Olig2⁺ transit-amplifying progenitors, associated with *Nf1* loss and Erk/MAPK activation.

Key findings:

1. Homozygous loss of *Nf1* tumor suppressor gene accompanying with activation of Erk/MAPK signaling occurs in SVZ-derived tumors in Type 2 cases.
2. Local expansion of tumor precursor populations is driven by increased numbers of Olig2⁺ transit-amplifying progenitors.

Figure 8 and Supplementary Figure 8. Rictor/mTORC2 deletion inhibits Akt and blocks distant dispersal.

Key findings:

1. PI3K/Akt inhibition by Rictor/mTORC2 deletion blocks distant glioma dispersal, generating less proliferative tumors associated with the SVZ.

In summary, using a combination of serial MRI and 3D-reconstruction, we reveal two distinct growth patterns of tumor evolution in these *p53*CKO models: Type 1 with locally rapid one-phase evolution of single masses and Type 2 with two-phase of spatially segregated tumor initiation in the SVZ and rapid tumor manifestation in distant regions of the brain parenchyma. Unique genetic features of *p53*-mutant driven malignant gliomas and GBMs, particularly the acquisition of one or more cNRT^{2N}(s) and WGD, allow us to build SKY-based single-cell phylogenetic trees and reveal tumor evolution of spatial segregation of tumor origin in the SVZ stem-cell niche and manifestation in distant brain regions. Importantly, constant correlation between GBM formation in the parenchyma and *Pten*/chr19 loss led to a proof-of-principle experiment, in which we show that PI3K/Akt inhibition by Rictor/mTORC2 deletion blocks distant glioma dissemination, generating less proliferative tumors associated with the SVZ.

References

1. Brennan CW, *et al.* The somatic genomic landscape of glioblastoma. *Cell* **155**, 462-477 (2013).
2. Ozawa T, *et al.* Most human non-GCIMP glioblastoma subtypes evolve from a common proneural-like precursor glioma. *Cancer Cell* **26**, 288-300 (2014).
3. Verhaak RG, *et al.* Integrated genomic analysis identifies clinically relevant subtypes of glioblastoma characterized by abnormalities in PDGFRA, IDH1, EGFR, and NF1. *Cancer Cell* **17**, 98-110 (2010).
4. Lee J, *et al.* Tumor stem cells derived from glioblastomas cultured in bFGF and EGF more closely mirror the phenotype and genotype of primary tumors than do serum-cultured cell lines. *Cancer Cell* **9**, 391-403 (2006).
5. Lee JH, *et al.* Human glioblastoma arises from subventricular zone cells with low-level driver mutations. *Nature* **560**, 243-247 (2018).
6. Gerstung M, *et al.* The evolutionary history of 2,658 cancers. *Nature* **578**, 122-128 (2020).

a

b

Sample	Cluster	Paper	Core	Proneural	Neural	Classical	Mesenchymal
TCGA-06-0673-11	Neural	FALSE	NA	4121.54441	7086.537525	-4637.608616	-5006.261712
TCGA-06-0675-11	Neural	FALSE	NA	3519.469485	6968.540789	-4619.407555	-3709.474913
TCGA-06-0676-11	Neural	FALSE	NA	4338.059868	4967.625556	-3767.26825	-4336.207548
TCGA-06-0678-11	Neural	FALSE	NA	3702.028887	3932.532762	-3924.89627	-3311.516977
TCGA-06-0680-11	Neural	FALSE	NA	3427.597811	6356.095015	-4042.694164	-4014.291553
TCGA-06-0681-11	Neural	FALSE	NA	4181.295311	5160.916157	-4273.372974	-4022.33338
TCGA-08-0623-11	Neural	FALSE	NA	4453.265046	4700.582994	-4097.438961	-4225.678422
TCGA-08-0625-11	Neural	FALSE	NA	4077.594747	4156.380237	-3548.519395	-3870.246771
TCGA-08-0626-11	Proneural	FALSE	NA	4233.889343	3941.272895	-3373.769556	-4791.681929
TCGA-08-0627-11	Proneural	FALSE	NA	4702.281832	4122.714502	-3418.366764	-4800.476024

a. Genomic alternations in five molecular subtypes of human GBMs (Brennan et al, 2013 Cell). *TP53* and *PDGFRA* genomic alternations in these five molecular subtypes are highlighted with red dashed box. Detailed analysis was performed and the result was shown in Supplementary figure 1m.

b. The expression signature of 10 non-neoplasia brain samples from TCGA was examed.

REVIEWER COMMENTS

Reviewer #1 (Remarks to the Author):

The authors have sufficiently addressed my previous comments and I am especially happy to see that they have clarified the possibility that Type I Tumors may represent an advanced stage of merged Type II Tumors, showing that this indeed can happen in a subset of cases. I have no further comments.

Reviewer #2 (Remarks to the Author):

The authors have significantly improved their manuscript through the incorporation of additional experiments, key clarifications of rationale and data analysis methods. They convincingly state their case for segregating Type 1 from Type 2 tumors based on serial MRI and SKY findings, as well as provide evidence for a subset of IDH-WT, TP53-mutant proneural gliomas that may be modeled by their GEM. Moreover, they adequately clarify the rationale for employing a single-cell SKY method in conjunction with WGS to construct phylogenetic trees. In all, Li et al. have appropriately addressed reviewer comments and present a thorough investigation of a relevant GEM model.

There remain a few aspects that could be clarified in the manuscript:

- The authors could better define the criteria for classifying Type 1 and 2 lesions. It is unclear in Fig. 2C why there are Type 1 tumors with multifocal lesions (contrary to the stated definition of Type 1 tumors), or why only 13 of 30 Type 2 tumors meet the definition of “spatially segregated tumors” when this is the stated definition of all Type 2 tumors.
- The authors claim (page 11) that clonal non-reciprocal translocations (cNRTs) serve as a “neutral lineage-specific marker”. It is unclear from the data presented that cNRTs serve a neutral function. The authors could leverage transcriptomic data to understand the impact of cNRTs on transcription, if they wish to assert this claim of neutrality.
- The authors could touch on a few points in their discussion:
 - o How accurately does their GEM model gliomas when all GEM tumors are whole-genome duplicated but only ~20% of TP53-mutant IDH-WT GBMs are whole-genome duplicated?
 - o Since p53 Δ E2-10null mice don't show clonal non-reciprocal translocations, how do the authors explain the mechanism by which Type 1 and 2 tumors are formed in these mice? The authors should

also clarify whether the Type 1 and 2 tumors that they extensively studied by SKY and WGS were selected from one or more p53-mutant or -null strains.

o Since the authors claim to model the subset of proneural GBMs that are TP53-mutant and IDH-WT, what are other genetic alterations that are enriched in this specific subset of GBMs that also correspond to their findings in GEM tumors (e.g. NF1 or PTEN loss)?

- IHC images are often unclear and should be accompanied by quantification supporting the statements made.

o Fig. S5C: p-S6 is seen in areas of the mouse brain where there is no Pten loss, but this is not commented on.

o Fig. 7C: the absence of pERK+ cells in the SVZ of control mice is not evident.

o What is the frequency of double positive pERK+Olig2+ cells among all p53-mutant cells in regions of the SVZ vs. tumor?

o The authors claim that p53-mutant cells do not express markers of mature OPCs (Ascl1, Ezh2) when the arrows in Fig. 7G highlight these double-positive cells, contradicting this statement.

- To improve the clarity of the manuscript, a few modifications could be made:

o Omitting extraneous data (e.g. Fig. 3B on Qki) where the relevance is unclear.

o Fig. 4 as validation of integrating SKY and WGS data could be moved to supplementary files.

o Clearly defining appropriate terminology – e.g. what is the difference between “founder clones” and “common ancestors”?

Minor

- Fig. 2A suggests that there may be a bimodal pattern of tumor formation in p53 Δ E5-6CKO mice. Have the authors examined whether Type 1 or 2 lesions form with different latencies?

- Page 3, first paragraph, sentence starting with “Primary GBMs are typically wild-type” is incomplete.

- Page 9: “Consistently, Pten was expressed at significantly higher level in SVZ-derived cells than those derived from autologous parenchymal tumors”. Sentence should be revised since it is only tumors that have Pten/chr19 loss where this trend is apparent.

Reviewer #3 (Remarks to the Author):

With additional experiments and clearer presentation, this revised MS has significantly improved its presentation. I thus support its publication. To further strengthen its main point that there is a unique evolutionary pattern in this murine model, some minor but essential explanations/discussions are needed.

One of the big advantages of this report is its use of a combination of SKY data with GWS data to study the genotype during cancer evolution. A bit more information about the SKY analyses will be helpful. For example, a table of the SKY data can help readers understand the dynamics at the karyotype level during evolution. It is known that both clonal chromosomal aberrations (CCAs) and non-clonal chromosome aberrations (NCCAs) are essential to indicate genome instability. With such data, authors can answer the following important questions: what is the overall instability for type I and type II tumors? What about sensitivity to drug treatment? Do type II tumors have higher chromosomal level diversity? Is any genome chaos involved (especially in model with p53 mutations)? Is there a punctuated cancer evolution phase detected from this model (either type I or II)?

Another concept missing is the importance of the chromosomal translocation (not just neutral specific marker, but different system coding). Karyotype coding needs to be mentioned for readers to understand why karyotype change is rather important for cancer evolution, as karyotype change (including polyploidy) mediated macrocellular evolution is key. By the way, authors need to let readers know that the use of the term whole genome duplication based on chromosome number > 50 is a bit too simplified. Again, a table of this data will be useful for future research.

Last, authors should clearly emphasize if some of the interesting questions raised in the introduction can be answered by this report. It is useful to highlight the similarities and differences from this model and human cancer data.

Responses to Reviewers
NCOMMS-18-53272A
Li et al.

We thank all three reviewers for their constructive and supportive comments on our manuscript. **No additional experiments were requested by the reviewers.**

While Reviewer 1 had no further concerns, Reviewer 2 suggested “...*There remain a few aspects that could be clarified in the manuscript...*” and Reviewer 3 asked for “...*some minor but essential explanations/discussions are needed...*”.

Accordingly, we have made no change in the “Abstract”, have corrected one minor grammatical error in the “Introduction”, and have made a few minor revisions in the “Results”, including some revised figure panels, two additional figure panels, and a new Supplementary Table as requested. **It should be noted that none of these revisions altered the conclusions of the manuscript.** Here is a list of the revisions that we have made in the figures:

Main Figures:

1. Fig. 2c: A minor revision that better illustrates the classification of Type 1 versus Type 2 tumors (see details in “Response 1a to Reviewer 2”).
2. Fig. 3b: A minor revision that removes the Western blot data of Qki as suggested by Reviewer 2 (see details in “Response 10 to Reviewer 2”).
3. Fig. 3g,h: The two newly added panels that compare the frequency of different types of NRTs between SVZ- and parenchyma-derived tumors in Type 2 cases as well as Type 1 tumors (see details in “Response 1a to Reviewer 3”).
4. Fig. 7c,d,i: Minor revisions to include high-magnification images and the quantification of the data to improve the clarity of the data presentation (see details in “Response 7 and 8 to Reviewer 2”).

Supplementary Figures and Tables:

1. Supplementary fig. 2c,d: Minor revisions that illustrate the rationale of using Ploidy 2.5 as a cutoff to distinguish tumors without whole-genome duplications (WGD) from those with WGD (see details in “Response 2b to Reviewer 3”).
2. Supplementary Table 1: An additional table that lists all clonal and non-clonal NRTs observed in Type 1 and Type 2 tumors (see details in “Response 1a to Reviewer 3”).

Following reviewers’ suggestions, we have substantially revised the “Discussion”, which has significantly improved the clarity and presentation of our manuscript. Here we provide a point-by-point response to the comments provided by the reviewers.

Reviewer #1 writes: “...*The authors have sufficiently addressed my previous comments and I am especially happy to see that they have clarified the possibility that Type I Tumors may represent an advanced stage of merged Type II Tumors, showing that this indeed can happen in a subset of cases. I have no further comments...*”.

Response: We would like to thank this reviewer again for raising this excellent question from our initial submission. This allowed us to uncover two distinct acquisition patterns of clonal non-reciprocal translations, cNRT^{4N} versus cNRT^{2N}, in the common-ancestor (CA) cell, associated with the evolution of Type 1 versus Type 2 tumors, respectively. **No further revision is requested.**

Reviewer #2 writes: “...*The authors have significantly improved their manuscript through the incorporation of additional experiments, key clarifications of rationale and data analysis methods. They convincingly state their case for segregating Type 1 from Type 2 tumors based on serial MRI and SKY findings, as well as provide evidence for a subset of IDH-WT, TP53-mutant proneural gliomas that may be modeled by their GEM. Moreover, they adequately clarify the rationale for employing a single-cell SKY method in conjunction with WGS to construct phylogenetic trees. In all, Li et al. have appropriately addressed reviewer comments and present a thorough investigation of a relevant GEM model...*”.

Response: We thank this reviewer for the positive comments for our manuscript.

There remain a few aspects that could be clarified in the manuscript:

- 1a. “...*The authors could better define the criteria for classifying Type 1 and 2 lesions. It is unclear in Fig. 2C why there are Type 1 tumors with multifocal lesions (contrary to the stated definition of Type 1 tumors) ...*”.

Response: We thank the reviewer for pointing out a lack of clarity on this important issue. Type 1 and Type 2 tumors are determined by serial MRI and/or histological analysis, developing as single-mass and multifocal-mass lesions, respectively. In the old version of **Fig. 2c**, we placed a subset of Type 2 cases with a complete merge of multiple spatially segregated tumors into the Type 1 category. As this reviewer pointed out, this presentation of the data led to unwanted confusion. In the revised **Fig. 2c**, we simplified the presentation, showing two separate bars for Type 1 and Type 2 cases from each of the three GEM models. Two distinct patterns were used to label the Type 2 cases with either partial (diamonds) or complete (diagonal lines) merge of multiple spatially segregated tumors during evolution. We have also attached a Figure attached to this response in which the old and new version of Fig. 2c were placed side-by-side for comparison (Additional Figure 1 for Reviewers).

- 1b. “...*or why only 13 of 30 Type 2 tumors meet the definition of “spatially segregated tumors” when this is the stated definition of all Type 2 tumors...*”.

Response: We thank the reviewer for pointing out a lack of clarity. Approximately 70% (30 out of 43) of the mice developed multiple segregated tumors and thus were diagnosed as Type 2 tumors. However, serial MRI/3D-reconstruction analysis showed that 13 of these 30 Type 2 cases exhibited either a partial (5/13) or complete merge (8/13) of multiple segregated tumors. These 13 Type 2 cases were placed on the top of each bar of Type 2 category labelled with

colored dashed lines in the revised Fig. 2c (see above). We have clarified this issue in the revised Results section (Page 7).

- 2. “...The authors claim (page 11) that clonal non-reciprocal translocations (cNRTs) serve as a “neutral lineage-specific marker”. It is unclear from the data presented that cNRTs serve a neutral function. The authors could leverage transcriptomic data to understand the impact of cNRTs on transcription, if they wish to assert this claim of neutrality...”

Response: Regarding functional neutrality, we had two important lines of the evidence supporting the notion that the clonal non-reciprocal translocations (cNRTs) in malignant gliomas and GBMs of both GEM models were unlikely to function as the major cancer drivers. First, the cNRTs were unique to the tumors from individual mice, i.e. never recurrent in different mice (**Fig. 4f**). Second, the cNRTs were absent in *p53*-null driven tumors, further suggesting the notion that they are not major drivers for glioma formation. However, functional studies using in vitro and/or in vivo assays are required to make a strong claim of neutrality. Accordingly, we have removed the term of “neutral” from the revised manuscript (Page 12). We would like to thank the reviewer for pointing out a potential overstatement of our results.

• **The authors could touch on a few points in their discussion:**

- o 3. “...How accurately does their GEM model gliomas when all GEM tumors are whole-genome duplicated but only ~20% of *TP53*-mutant *IDH*-WT GBMs are whole-genome duplicated...?”

Response: We have revised the Results as “Consistently, 18% of *TP53*-mutant *IDH*-WT GBMs in the TCGA dataset underwent WGD compared to only 7% of *TP53*-wild-type tumors, suggesting that these *p53*CKO models resemble the subset of *TP53*-mutant *IDH*-WT GBMs with WGD in humans (Page 7).

- o 4a. “...Since *p53*ΔE2-10null mice don’t show clonal non-reciprocal translocations, how do the authors explain the mechanism by which Type 1 and 2 tumors are formed in these mice?”

Response: Similar frequency of Type 1 versus Type 2 tumors between *p53*-mutant and *p53*-null driven GEM models suggests a potentially shared mechanism underlying glioma formation in the two types of the genetic perturbations. However, the absence of cNRTs in *p53*-null driven GEM tumors precluded us from using the SKY method to establish the lineage relationship, tracing the origin of sub-4N malignant gliomas and GBMs in the brain parenchyma back to the near-2N lesions in the SVZ. However, single nucleotide variations (SNVs) identified by single-cell WGS analysis will allow us to build a phylogenetic tree for spatially segregated Type 2 tumors in the *p53*-null driven GEM model. We have included this revision in the Discussion section (Page 20).

- o 4b. “...The authors should also clarify whether the Type 1 and 2 tumors that they extensively studied by SKY and WGS were selected from one or more *p53*-mutant or -null strains...”.

Response: Our extensive analysis of tumor evolution for Type 1 and Type 2 tumors using SKY and WGS technologies has focused on the *p53*^{R172H} model, which genetically mimic human

GBMs with the hotspot $TP53^{R175H}$ mutation. This clarification has been included in the revised Results section “We therefore focused on the $p53^{R172H}$ model, which was characterized by the high frequency of cNRTs and carried the equivalent $TP53^{R175H}$ hotspot missense mutation in human cancers, for further investigation” (Page 7,8).

o 5. “...*Since the authors claim to model the subset of proneural GBMs that are TP53-mutant and IDH-WT, what are other genetic alterations that are enriched in this specific subset of GBMs that also correspond to their findings in GEM tumors (e.g. NF1 or PTEN loss)?...*”.

Response: We have added this following to the Discussion section (Page 18).

In brief, we found molecular or genetic alterations of cancer drivers, including *Pdgfra*, *Pten* and *Nf1*, which are also altered in human $TP53$ -mutant IDH -WT Proneural GBMs. Similar to human $TP53$ -mutant IDH -WT Proneural GBMs, $p53$ -mutant driven malignant gliomas in our GEM models were IDH -WT with non-G-CIMP and a Proneural signature, with ~50% of which overexpressed *Pdgfra*/*Pdgfra*¹. Similar to IDH -WT GBMs with almost a universal loss of *PTEN*/Chr10, we observed 100% penetrance of *Pten*/chr19 loss in our GEM gliomas. Further, the bi-allelic inactivation of *Pten* observed in almost all GEM gliomas from our models is enriched in $TP53$ -mutant IDH -WT Proneural GBMs (60%) compared to other subtypes (30%-40%)¹. These observations validate the relevance of our GEM models to human $TP53$ -mutant IDH -WT Proneural GBMs. It should be noted that approximately 50% of our GEM gliomas exhibited *Nf1* loss, which, compared to $TP53$ -WT GBMs (20%), were enriched in $TP53$ -mutant GBMs (50%), including $TP53$ -mutant Proneural GBMs (40%)^{1,2}.

• IHC images are often unclear and should be accompanied by quantification supporting the statements made.

o 6. “...*Fig. S5C: p-S6 is seen in areas of the mouse brain where there is no Pten loss, but this is not commented on...*”.

Response: We have revised the figure legend for clarity, by adding the following: “Certain *Pten* wild-type brain areas, including neurons in the hippocampus and cerebellum, exhibited high levels of p-S6 expression. This was in contrast to malignant gliomas in the corresponding SVZ and brain parenchyma, thus suggesting *Pten*-independent regulation of PI3K/Akt/mTORC1 signaling pathway in neurons, but not in GBMs” (see the revised Figure legend for Supplementary Fig. 5c).

o 7. “...*Fig. 7C: the absence of pERK+ cells in the SVZ of control mice is not evident...*”.

Response: We have revised Figure 7 to address the reviewer’s concerns. Specifically, we have added insets in Fig. 7c showing high-magnification areas containing p-Erk⁺BrdU⁺ cells (yellow) in the mutant SVZ. The control SVZ only contained p-Erk⁻BrdU⁺ cells (green), with pERK⁺ cells (red, yellow) found only outside the SVZ. Additionally, the number of BrdU⁺ cells, the number of p-Erk⁺ cells and the ratio of p-Erk⁺BrdU⁺/p-Erk⁺ cells were quantified in control and $p53$ -mutant SVZs. The quantification results were shown in the revised Fig. 7d.

o 8. “...*What is the frequency of double positive pERK+Olig2+ cells among all p53-mutant cells in regions of the SVZ vs. tumor?...*”.

Response: We thank the reviewer for this suggestion. We have quantified pErk+Olig2+ cells in the control normal SVZ, mutant-SVZ, and also the glioma-like clusters. The pErk+Olig2+ cells were increased 4-fold in the glioma-like clusters compared with the mutant-SVZ (none in the control SVZ). These results are shown in the revised Fig. 7, as Fig. 7i.

o 9. “...*The authors claim that p53-mutant cells do not express markers of mature OPCs (Ascl1, Ezh2) when the arrows in Fig. 7G highlight these double-positive cells, contradicting this statement...*”.

Response: We regret using a long sentence that led to mis-presentation of the data. We have revised the sentence as the following: “Consistently, p-Erk⁺Olig2⁺p53^{Mutant}-expressing glioma precursor cells expressed the markers for neural stem cells and transit-amplifying progenitors (TAPs), including Ascl1, Ezh2 and BLBP (brain lipid binding protein), but not more differentiated oligodendrocyte precursor cells (OPCs) (Fig. 7j)” (Page 17).

• **To improve the clarity of the manuscript, a few modifications could be made:**

o 10. “...*Omitting extraneous data (e.g. Fig. 3B on Qki) where the relevance is unclear...*”.

Response: According to reviewer’s suggestion, we have removed the Western blotting data of Qki expression in Fig. 3b and the description of Qki data in the revised Results section (Page 10).

o 11. “...*Fig. 4 as validation of integrating SKY and WGS data could be moved to supplementary files...*”.

Response: Figure 4 demonstrates how bulk-tumor-derived WGS data validate and complement the SKY-based single-cell/clone tree building. First, we showed that CNVs called from the WGS data were highly consistent with the chromosomal gains and losses determined by the SKY data (**Fig. 4a and Pearson correlation coefficient =0.85; Fig. 4b**). These results demonstrate that the small number of tumor cells analyzed by the SKY method are representative to the whole tumor cell populations, confirming no significant sampling bias from the SKY method. Second, we used WGS data to “genotype” the individual breakpoints of NRTs in each sample. In total, 20 out of 26 NRT events had their breakpoints identified confidently with high quality reads. We confirmed that cells from the same brain shared the same breakpoints, thus establishing the validity of defining NRT as a single event (not recurrent) when building a tree (**Fig. 4c-f**). WGS data thus confirmed that all cells with the same NRT(s) arise from the same genetic event. Third, the non-recurrent nature of cNRTs among tumors from different mice was further validated by Monte Carlo simulation. Together, these results provide the methodological basis for building single-cell phylogenetic trees, revealing the unique two-phase pattern of tumor evolution in our GEM GBM model. Together, Figure 4 goes far beyond basic technical verification, which was

the major revised figure to address this reviewer's critique from the last submission. We believe that it should be kept as the main figure.

o 12. "...Clearly defining appropriate terminology – e.g. what is the difference between “founder clones” and “common ancestors”?...”.

Response: We have revised the manuscript to more specifically define these two terms. The common ancestor (CA) is defined as the oldest founder clone (FC0 or FC1) either directly observed or inferred by single-cell phylogenetic tree building. The FCs are clonally related progeny directly or indirectly derived from the CA cell during cancer evolution. Additionally, we have removed references to CA and the use of CA/FC from the results section.

Minor

• 13. "...*Fig. 2A suggests that there may be a bimodal pattern of tumor formation in p53 Δ E5-6CKO mice. Have the authors examined whether Type 1 or 2 lesions form with different latencies?...*".

Response: This is an interesting observation that we did not recognize in the last submission. We have now analyzed survival curves and tumor growth curves of p53 Δ E5-6CKO mice with Type 1 and Type 2 tumors and observed no difference (Additional Figure 2 for Reviewers). The “*bimodal pattern*” in the figure is likely resulted from a relatively small number of p53 Δ E5-6CKO mice analyzed in the study. Indeed, a significant variation in the onset of tumor formation (ranging from 6 months to 10 months determined by the timing of the earliest detectable tumors) was observed in all three GEM models (**Fig. 2a**). Of note, rapid progression of these adult-onset malignant gliomas and GBMs was similarly observed among these three models.

• 14. "...Page 3, first paragraph, sentence starting with “Primary GBMs are typically wild-type” is incomplete...”.

Response: We thank the reviewer for pointing out this grammatical error. This sentence has been revised as “Primary GBMs are typically wild-type (WT) for Isocitrate Dehydrogenase 1/2 (*IDH*), with no *IDH*-mutant associated hypermethylated phenotype (termed Glioma CpG Island Methylated Phenotype or G-CIMP), while secondary (progressive) GBMs often arise from *IDH*-mutant lower-grade gliomas^{1,3,4,5,6,7} (Page 3).

• 15. "...Page 9: “*Consistently, Pten was expressed at significantly higher level in SVZ-derived cells than those derived from autologous parenchymal tumors*”. Sentence should be revised since it is only tumors that have *Pten/chr19* loss where this trend is apparent...”.

Response: We have revised this sentence as “Consistent with intact *Pten/chr19*, the expression of Pten protein was significantly higher level in SVZ-derived cells than those derived from autologous parenchymal tumors” (Page 9).

We would like to thank the reviewer for his/her rigorous and thorough comments that greatly helped improve our manuscript.

Reviewer #3 writes: “...*With additional experiments and clearer presentation, this revised MS has significantly improved its presentation. I thus support its publication. To further strengthen its main point that there is a unique evolutionary pattern in this murine model, some minor but essential explanations/discussions are needed...*”.

1a. “...*One of the big advantages of this report is its use of a combination of SKY data with GWS data to study the genotype during cancer evolution. A bit more information about the SKY analyses will be helpful. For example, a table of the SKY data can help readers understand the dynamics at the karyotype level during evolution. It is known that both clonal chromosomal aberrations (CCAs) and non-clonal chromosome aberrations (NCCAs) are essential to indicate genome instability. With such data, authors can answer the following important questions: what is the overall instability for type I and type II tumors? What about sensitivity to drug treatment? Has type II tumors with higher chromosomal level diversity?*”

Response: We thank this reviewer for supporting the publication of our manuscript. Following this reviewer’s suggestion, we have included a table that lists each of cNRTs and non-cNRTs identified in individual tumors from Type 1 (n = 7) and Type 2 (n = 6) cases. We determined the frequency of different types of NRTs, including clonal and non-clonal (Supplementary Table 1). Despite no significant difference in cNRTs observed among all tumor samples analyzed, both Type 1 and Type 2 tumors observed in the brain parenchyma showed a significantly more diverse population of non-cNRTs and consequently total NRTs than those from SVZ-derived tumors with near-2N genomes of Type 2 cases, exhibiting high levels of genome instability (Fig. 3g,h and Supplementary Table 1). Of note, the sub-4N tumor cells, including those observed in the SVZ of Type 2 tumors, all exhibited high levels of genome instability (Fig. 3g,h). Thus, the Type 2 pattern exhibits a two-phase tumor evolution – the “early” phase with normal *Pten*/chr19 and relative stable genomes in the SVZ and the “late” phase with *Pten*/chr19 loss and highly unstable genomes in the spatially segregated distant brain parenchyma, while the Type 1 pattern only has a one “late” phase during tumor evolution (Fig. 3i). This revision has been included in the Results section (Page 10). In addition, we have revised in the Discussion section “Importantly, low versus high levels of genomic instability observed in the SVZ- versus autologous parenchyma-derived tumor cells in Type 2 cases could confer different sensitivity to drug treatment as previously described in human GBMs⁸” (Page 20).

1b. “...*Is any genome chaos involved (especially in model with p53 mutations)?...*”.

Response: Our analysis did not observe a dense jumble of copy-number variations clustered in the same chromosomal location, which is called “genome chaos” in the literature. However, the presence of cNRTs exclusively observed in *p53*-mutant driven malignant gliomas and GBMs demonstrates higher levels of genomic instability than those driven by *p53*-null mutations.

1c. “...*Is there a punctuated cancer evolution phase detected from this model (either type I or II)?...*”.

Response: We thank the reviewer for raising the potential significance of the two-phase cancer evolutionary pattern observed in Type 2 tumors of our GEM models. We observed the existence of the genome stable and unstable phases in autologous SVZ- and parenchyma-derived tumors in Type 2 cases, respectively (also see “Response 1a to Reviewer 3 above). However, further studies using the high-resolution WGS technology are required to delineate whether the transition between these two phases is sudden or not (a punctuated cancer evolution).

2a. *“...Another concept missing is the importance of the chromosomal translocation (not just neutral specific marker, but different system coding). Karyotype coding needs to be mentioned for readers to understand why karyotype change is rather important for cancer evolution, as karyotype change (including polyploidy) mediated macrocellular evolution is key...”*.

Response: We thank this reviewer for raising a potential role of NRTs in cancer evolution. As discussed in “Response 4 to Reviewer 2” above, we used cNRT as a lineage-specific tracing marker to build the phylogenetic trees. However, our results suggest that cNRTs are unlikely to play a major driver role in cancer evolution in our GEM models, particularly in the case of the *p53*-null driven malignant gliomas and GBMs that lack cNRT. However, WGD is highly associated with high levels of genome instability of sub-4N malignant gliomas and GBMs, suggesting a key role in mediating macrocellular evolution (see the details in “Response to 1a-c”).

2b. *“...By the way, authors need to let readers know that the use of the term whole genome duplication based on chromosome number > 50 is a bit too simplified. Again, a table of this data will be useful for future research...”*.

Response: We thank the reviewer for pointing out a lack of clarity. We have provided the rationale on how we used the increase of tumor ploidy to 2.5 (Ploidy 2.5) as a cutoff to define whole-genome duplication (WGD) in our study (see details below in the revised Methods, Page 16).

Although focal amplifications could in principle increase tumor ploidy to 2.5, computational analysis suggests that this is rare. Data from pan-cancer analysis showed tumors with versus without WGD were clearly separated at Ploidy 2.5 (Human, $n = 55$ chromosomes)⁹. Furthermore, we investigated the distribution of chromosome numbers in human high-grade gliomas (including Grade III gliomas and Grade IV gliomas/GBMs) using cytogenetic data collected by the Mitelman database (Supplementary Fig. 2d). The data clearly showed the bimodal distribution for WGD and non-WGD tumors with chromosome counts around 55 (ploidy 2.5) as a cutoff. In our cytogenetic data, we confirmed that this bimodal distribution and the same Ploidy 2.5 ($n = 50$) was indeed an excellent cutoff value to separate the two modes (Supplementary Fig. 2c). Thus, we used ploidy 2.5 (50 chromosomes in mouse cells) to separate WGD and non-WGD tumors from our GEM models.

3. *“...Last, authors should clearly emphasize if some of the interesting questions raised in the introduction can be answered by this report. It is useful to highlight the similarities and differences from this model and human cancer data...”*.

Response: We thank the reviewer for this suggestion. Accordingly, we have substantially revised the Discussion (Page 18-20). First, we discuss how the p53-mutant driven malignant gliomas resemble a subset of human *TP53*-mutant *IDH*-WT GBMs with a Proneural signature and WGD. Second, we discuss the one- and two-phase evolutionary patterns elucidated in Type 1 and Type 2 tumors, which model the growth patterns of single- and multifocal-mass in human *IDH*-WT GBMs, respectively. These studies provide important insights on the clinical challenges to identify early precursor cells for primary/de novo GBMs. Third and more specifically, spatial segregation of tumor initiation in the SVZ and tumor manifestation in the distant brain parenchyma provides a great challenge to investigate and develop treatment for GBMs with such a unique evolutionary pattern. Finally, we provide a proof-of-principle experiment to convert highly migratory GBMs into a localized lesion with inhibition of PI3K/Akt/mTORC signaling.

We would like to thank this reviewer for the excellent suggestions.

References

1. Brennan CW, *et al.* The somatic genomic landscape of glioblastoma. *Cell* **155**, 462-477 (2013).
2. Verhaak RG, *et al.* Integrated genomic analysis identifies clinically relevant subtypes of glioblastoma characterized by abnormalities in PDGFRA, IDH1, EGFR, and NF1. *Cancer Cell* **17**, 98-110 (2010).
3. Ceccarelli M, *et al.* Molecular Profiling Reveals Biologically Discrete Subsets and Pathways of Progression in Diffuse Glioma. *Cell* **164**, 550-563 (2016).
4. Noshmehr H, *et al.* Identification of a CpG island methylator phenotype that defines a distinct subgroup of glioma. *Cancer Cell* **17**, 510-522 (2010).
5. Ohgaki H, Kleihues P. The definition of primary and secondary glioblastoma. *Clin Cancer Res* **19**, 764-772 (2013).
6. Ozawa T, *et al.* Most human non-GCIMP glioblastoma subtypes evolve from a common proneural-like precursor glioma. *Cancer Cell* **26**, 288-300 (2014).
7. Sturm D, *et al.* Paediatric and adult glioblastoma: multifactorial (epi)genomic culprits emerge. *Nat Rev Cancer* **14**, 92-107 (2014).
8. Piccirillo SG, *et al.* Contributions to drug resistance in glioblastoma derived from malignant cells in the sub-ependymal zone. *Cancer Res* **75**, 194-202 (2015).
9. Zack TI, *et al.* Pan-cancer patterns of somatic copy number alteration. *Nat Genet* **45**, 1134-1140 (2013).
10. Bielski CM, *et al.* Genome doubling shapes the evolution and prognosis of advanced cancers. *Nat Genet* **50**, 1189-1195 (2018).

(c) Frequency of Type 1 and Type 2 growth patterns of three p53CKO models were determined by serial MRI. The single-mass Type 1 pattern was marked by black. The multifocal-mass Type 2 pattern was presented by three types of end-stage multifocal masses: (1) no merge (red), (2) partial merge (red with diamonds) and (3) complete merge (could be misdiagnosed as Type 1, red with stripes).

Figure 2 for "Response 13 to Reviewer 2"